# Strong aerosol indirect radiative effect from dynamic-driven diurnal variations of cloud water adjustments

Jiayi Li[1+], Yang Wang[1+], Jiming Li[1*], Weiyuan Zhang[1], Lijie Zhang[1], Yuan Wang[1]

[1] Collaborative Innovation Center for Western Ecological Safety, College of Atmospheric Sciences, Lanzhou University, Lanzhou 730000, China.

*Correspondence to*: Jiming Li (lijiming@lzu.edu.cn)

[+] These authors contributed equally to this work.

**Abstract.** Aerosol-cloud interaction (ACI) remains a key uncertainty in climate projections. A major challenge is that the sign and magnitude of cloud liquid water path (LWP) response to aerosol perturbations (represented by cloud droplet number concentration, $N_d$) at different temporal and spatial scales are highly variable, but potential microphysical-dynamical mechanisms are still unclear, especially at a diurnal scale. Here, geostationary observations were conducted in two distinct cloud regions: the stratocumulus region off the western Australia and clouds over the East China Sea characterized by a transition from stratocumulus to cumulus under strong anthropogenic influences. In contrast to the commonly observed inverted-V $N_d$-LWP relationship, LWP increases at high $N_d$ (> ~300 cm$^{-3}$) in the ECS, exhibiting a V shape. Our analysis indicates this unique V shape arises from large-scale meteorological covariations (e.g. cold air advection), which lead to increases in both LWP and $N_d$. Furthermore, the diurnal variation of LWP adjustments is driven primarily by diurnal-related boundary layer decoupling and cloud-top entrainment. The diurnal LWP adjustments exhibit a distinct regional pattern associated with cloud regimes. The results indicate that neglecting diurnal variations of LWP adjustments leads to an underestimation (up to 89%) of the cooling effect induced by changes in cloud albedo due to aerosol perturbations in the AUW. The bias spans from a 24% overestimation to a 40% underestimation in the ECS. Our findings highlight the key role of diurnal variations of ACI in reducing the uncertainty in climate projections.

## 1 Introduction

Marine low-level clouds (MLCs), which cover one-third of the global ocean (Klein and Hartmann, 1993), exert a strong cooling effect by reflecting the incoming solar radiation back into space (Jiang et al., 2023). Cloud reflectivity to solar radiation is highly sensitive to atmospheric aerosol concentrations. Because aerosols can serve as the cloud condensation nuclei (CCN), which modify key microphysical variables such as cloud droplet number concentrations ($N_d$) and droplet effective radius ($r_e$). For a given cloud liquid water content, aerosol-induced increases in CCN can enhance $N_d$ and hence reduce $r_e$, boosting cloud albedo (Twomey, 1977), which is known as cloud albedo effect, being an important component of aerosol-cloud interactions (ACI). Additional alterations in cloud microphysics may arise from changes in the quantity of liquid water or cloud cover that

are induced by aerosol variations. These changes can lead to rapid adjustments within the cloud in response to aerosol perturbations, which is another crucial component of ACI (Bellouin et al., 2020). For example, it has been documented that liquid water path (LWP) can either increase due to precipitation suppression (positive LWP adjustments) (Albrecht, 1989) or decrease due to entrainment feedbacks (negative LWP adjustments) (Ackerman et al., 2004; Bretherton et al., 2007; Small et al., 2009). While the Twomey effect is well-recognized, however, LWP adjustments are highly uncertain as the least

understood and most poorly quantified in all climate forcing (IPCC, 2023).

These large uncertainties in LWP adjustments are generally attributed to the complex interplay of microphysical-dynamical conditions and aerosol loading (represented by $N_d$) that vary with different temporal and spatial scales (Bender et al., 2019; Chen et al., 2014; Glassmeier et al., 2021; Gryspeerdt et al., 2022a). Numerous observational studies have been carried out to understand the extent of this variability and uncertainties of LWP adjustments, with the aim of constraining

model simulations (Gryspeerdt et al., 2019, 2021; Rosenfeld et al., 2019; Trofimov et al., 2020; Wilcox, 2010). These investigations have spanned various regions and targets, revealing diverse cloud responses attributable to the varied mechanisms of LWP adjustments. In addition, it has been confirmed that analysis methods, sampling strategies, and meteorological covariations could be another considerable source of uncertainty in LWP adjustments (Chen et al., 2014; Gryspeerdt et al., 2022b; Rosenfeld et al., 2019, 2023). Here, we focus on the time-dependence of LWP adjustments (i.e.,

diurnal variations) as it is associated with both sampling strategies and meteorological covariations. It has been established that marine cloud properties and the cloud-topped marine boundary layer exhibit prominent diurnal variations in response to solar radiation, which are closely related to their regional dependence (Duynkerke and Hignett, 1993; Wood et al., 2002). The microphysical-dynamical boundary layer feedback, which generally covaries with the regional diurnal cycle, could augment or weaken the LWP adjustments and thus lead to the diurnal variation of LWP adjustments with broad spreads and even

different signs. This means that a one-size-fits-all approach to global-mean LWP adjustments may not provide a robust constraint, given the regional and temporal mechanisms at play (Michibata et al., 2016). Additionally, the microphysical-dynamical mechanisms behind are complex and still poorly understood (Feingold et al., 2024). This drives the speculation that the diurnal variations of LWP adjustments could be one of the most significant yet overlooked sources of uncertainty of ACI.

However, to date, a majority of studies have relied on observations from polar-orbiting satellites to investigate the spatial

distribution and long-term variations of $N_d$ (Bennartz and Rausch, 2017; Li et al., 2018; McCoy et al., 2018), which are insufficient to depict the time-dependent nature of LWP adjustments. Based on Himawari-8 geostationary satellite, the diurnal variations of cloud microphysical properties and LWP adjustments in two typical regions, and the associated influencing factors and mechanisms are presented in this study. Our research aims to expand our understanding of the influence of meteorological factors, initial aerosol states (especially $N_d$), and the covariance between meteorology and aerosols on cloud

LWP, gaining a comprehensive understanding of the diurnal variations in LWP adjustments, which is a highly time-dependent variable lacking quantification, in conjunction with shifts in regional meteorological conditions.

## 2 Data and Methods

Our analysis focuses on $1° \times 1°$ marine low-level cloud samples, aggregated from filtered pixel-level satellite data. Within the sight of Himawari-8, we selected two cloud regions with significantly different environmental backgrounds (see Fig. S1 in Supplementary Materials). One is a remote stratocumulus region located in the west of Australia (AUW: 25°-35°S, 95°-105°E) (Klein and Hartmann, 1993). The other is in the East China Sea (ECS: 20°-30°N, 120°-130°E), which is significantly impacted by anthropogenic aerosols and characterized by Sc to Cu transition (Long et al., 2020). The comparison between the two regions allows us to explore the regional differences of LWP adjustments and their potential driving mechanisms. In total, we collected 480189 cloud samples in the AUW and 173181 cloud samples in the ECS using a 4-year (2016-2019) hourly record from SatCORPS Himawari-8.

### 2.1 $N_d$ retrieval based on geostationary satellite product

In this study, 4 years (2016-2019) of hourly cloud microphysical properties data from the Satellite Cloud and Radiation Property retrieval System (SatCORPS) Clouds and the Earth's Radiant Energy System (CERES) Geostationary Satellite (GEO) Edition 4 Himawari-8 over the Northern Hemisphere (NH) (Southern Hemisphere (SH)) Version 1.2 data product (CER_GEO_ED4_HIM08_NH_V01.2, CER_GEO_ED4_HIM08_SH_V01.2) were collected (NASA/LARC/SD/ASDC, 2018b, a). The datasets are derived from the Advanced Himawari Imagers (AHI) on Himawari-8 geostationary satellite, using the Langley Research Center (LARC)s SatCORPS algorithms in support of the CERES project (Minnis et al., 2021; Trepte et al., 2019). The retrievals are at 2-km resolution (at nadir) and are sub-sampled to 6 km. The sub-sampled resolution meets the needs of the CERES project without having a data implosion. The cloud optical thickness (CLOT), cloud effective radius ($r_e$) and cloud-top temperature (CLTT) from the SatCORPS product during the daytime were used to calculate $N_d$ in our study. Other cloud properties such as cloud-top height (CLTH), cloud base height (CLBH) and cloud thickness (H) were used in further analysis. The SatCORPS product is based on the CERES Ed4 cloud retrieval algorithm (Minnis et al., 2021), which provides more accurate parameterizations of CLTH and H than the CERES Edition 2 retrieval algorithm (Minnis et al., 2011). Briefly, for boundary layer clouds, CLTH is retrieved using a lapse rate method: $\Gamma_b = (CET - T_0)/(CLTH - Z_0)$ (Sun-Mack et al., 2014). Cloud effective temperature (CET) was estimated from the Infrared Window (IRW) channel. $Z_0$ denotes the surface elevation and $T_0$ is the sea surface temperature. H is computed using empirical formulas with $\tau$: $H = 0.39 \ln \tau - 0.01$ for liquid clouds. CLBH is directly obtained by subtracting H from CLTH.

SatCORPS cloud droplet effective radius ($r_e$) is primarily estimated from the 3.9 μm near-infrared band (Kang et al., 2021), which is closest to the cloud top with less bias in further calculation of $N_d$ (Grosvenor et al., 2018). $N_d$ is estimated as (Bennartz, 2007):

$$N_d = \frac{\sqrt{5}}{2\pi k} \left( \frac{f_{ad} c_\omega \tau}{Q \rho_w r_e{}^5} \right)^{\frac{1}{2}} \tag{1}$$

where $\tau$ represents cloud optical depth and $\rho_w$ is liquid water density. The extinction efficiency factor $Q \approx 2$. $k$, related to droplet

size distribution, is set as 0.8 for maritime cloud (Martin et al., 1994; Painemal and Zuidema, 2011). $c_w$ represents the condensation rate determined by temperature in cloud (here is the cloud-top temperature from SatCORPS). A constant adiabatic value ($f_{ad}$) of 0.8 is used to represent the deviation from the adiabatic profile (Bennartz, 2007). This is the most common method to derive $N_d$ from passive satellite observations (Bennartz, 2007; Bennartz and Rausch, 2017; Li et al., 2018; McCoy et al., 2018) and has been validated as a reliable technique for observing changes in long-term variations of $N_d$ (Boers et al., 2006). Li et al. (2018) demonstrated that passive satellite $N_d$ retrievals exhibit strong consistency with active satellite retrievals. The SatCORPS Himawari-8 retrievals agree well with in-situ observations according to Kang et al. (2021). In this study, the LWP from SatCORPS is calculated as $\frac{5}{9}\rho_w \tau r_e$ in sub-adiabatic conditions, following the method by Wood and Hartmann (2006). The combination of these two retrieval methods of $N_d$ and LWP has been widely used in the satellite investigations of LWP adjustments (Fons et al., 2023; Gryspeerdt et al., 2019; Qiu et al., 2023; Smalley et al., 2024).

Several sampling strategies were adopted in this study to select cloud pixels to reduce uncertainties (Grosvenor et al., 2018; Gryspeerdt et al., 2019; Li et al., 2018). Only pixels in the liquid phase with cloud-top temperature warmer than 268 K under 3.2 km were included. To maintain consistency with previous studies (Bennartz and Rausch, 2017; Li et al., 2018), we adopted 268 K as the threshold of CLTT for liquid clouds, rather than 273 K. In fact, 96% (97%) of the samples exhibited CLTT above 273 K in the AUW (ECS) region. Therefore, the threshold has a negligible impact on the overall results. The lower bounds of $r_e$ ($\tau$) were set as 4 μm (4) to reduce uncertainties. Moreover, pixels with solar zenith angles larger than 65° were excluded. Filtered data were used to calculate $N_d$ and then aggregated to a 1° × 1° grid. Each grid containing at least 30 pixels is considered a cloud sample. On average, each grid contains 83 (87) pixels in the AUW (ECS) region.

We followed the above methods to filter cloud pixels, which only limit cloud top properties and cloud phase, inevitably including different cloud regimes, such as low-level cumulus clouds. This might introduce uncertainties as cumulus clouds and stratocumulus clouds have different adiabatic properties, but we have set $f_{ad}$ as a constant value in $N_d$ calculations. Small et al. (2013) found that the $f_{ad}$ of cumulus clouds showed no significant variation with height, whereas Wood (2005) observed that the adiabaticity in stratocumulus clouds decreased from cloud base to cloud top. The difference in departures from adiabaticity between cumulus and stratocumulus stems from their different entrainment processes. Stratocumulus clouds are primarily influenced by the entrainment of dry air at cloud top (Mellado, 2017). In contrast, cumulus clouds are dominated by lateral entrainment (Heus et al., 2008). Uncertainties may also occur as $f_{ad}$ varies with cloud depth (Grosvenor et al., 2018; Min et al., 2012; Wang et al., 2021). As acquiring hourly $f_{ad}$ on a global scale is rather difficult, to date, studies investigating diurnal variations of LWP adjustments based on geostationary satellites continue to employ a constant $f_{ad}$ value (Fons et al., 2023; Qiu et al., 2024; Smalley et al., 2024). Also, the choices of a constant $k$ might introduce bias into the retrieval of $N_d$ (Grosvenor et al., 2018). Studies have found that $k$ parameter varied with the height within cloud and cloud types (Brenguier et al., 2011; Martin et al., 1994; Painemal and Zuidema, 2011). This indicates that the presence of diurnal variations in $k$ and $f_{ad}$ (e.g., hourly changes in entrainment rate can modify $f_{ad}$) introduces further bias. The resulting uncertainties warrant further in situ observation to improve the accuracy.

To minimize the influence of precipitation on $N_d$ and LWP retrievals, GPM IMERG Final Precipitation L3 Half Hourly 0.1 degree x 0.1 degree V07 (GPM_3IMERGHH) was used (Huffman et al., 2020). Cloud samples were included in the analysis only if the GPM_3IMERGHH precipitation rate equals 0 mm/hr in a $1° \times 1°$ grid. To align these two satellite products, SatCORPS cloud pixels within each 0.1° grid of GPM_3IMERGHH are assigned the same precipitation value. Considering the limited ability of GPM to detect light precipitation and drizzle, we additionally applied a $r_e = 14\,\mu m$ threshold to distinguish between drizzle scenes and non-drizzle scenes (black lines in Fig. 1).

## 2.2 Quantification of LWP adjustments

To quantify LWP response, two methods have been used in previous studies. The logarithmic relationship between $N_d$ and LWP ($\frac{\partial \ln LWP}{\partial \ln N_d}$) is the standard way to quantify LWP sensitivity to aerosol from satellite data, where $N_d$ is considered a proxy of CCN. Another way of describing the changes of cloud water due to aerosols ($-\frac{\Delta \ln \tau}{\Delta \ln r_e}$) is deduced from the contributions of changes in LWP and $r_e$ to the changes in cloud optical depth ($\frac{\Delta \tau}{\tau} = \frac{\Delta LWP}{LWP} - \frac{\Delta r_e}{r_e}$) (Christensen and Stephens, 2011; Coakley and Walsh, 2002). Whereas the latter method is put forward with a default condition that $\Delta r_e$ is always negative, it is only applicable to small-scale pollution tracks like industry tracks, volcano tracks or ship tracks, etc. (Rahu et al., 2022; Toll et al., 2019). Therefore, the former method is applied in this study, which has been commonly used in research on aerosol-cloud interactions based on large-scale satellite observations (e.g. Glassmeier et al., 2021; Gryspeerdt et al., 2019).

LWP adjustment at any given moment is the result of all available data at that moment. The regression slope of $N_d$ and LWP in log-log space ($\frac{\partial \ln LWP}{\partial \ln N_d}$) is calculated on 1° grid scale. We employed equal-width binning, using the median LWP within each $N_d$ bin to regress the slope. To reduce noise from sparse samples, only bins with more than 50 samples were used to calculate LWP adjustments. Additionally, we tested the equal-sample binning method. The patterns of the $N_d$-LWP relationship and diurnal variations of LWP adjustments remained robust across different binning methods. The main reason for choosing equal-width binning was to preserve the original physical scale of the samples, avoiding the excessive smoothing of samples with diverse meteorological conditions gathered in a single bin using equal-sample binning (Towers, 2014).

## 2.3 Reanalysis datasets

Aerosol property is represented by the total column extinction optical depth (AOD) at 550 nm from hourly time-averaged 2-dimensional data collection in Modern-Era Retrospective analysis for Research and Applications version 2 (MERRA-2), with a spatial resolution of $0.5° \times 0.625°$ (Buchard et al., 2017). It is interpolated onto a $1° \times 1°$ grid using the bilinear interpolation method.

Meteorological indicators related to cloud microphysical process are obtained from ERA5 reanalysis data (Hersbach et al., 2020), including sea surface temperature (SST), lower-tropospheric stability (LTS), relative humidity on 700 hPa and 1000 hPa (RH700 and RH1000), vertical velocity on 700 hPa (omega700), horizontal wind field on 700 hPa and horizontal

temperature advection at the surface ($SST_{adv}$). The ERA5 is the fifth-generation atmospheric reanalysis of global climate and is produced using the ECMWF's Integrated Forecast System cycle 41r2 with a 4-dimensional variation assimilation system. Compared to the ERA-Interim, the ERA5 has higher spatial (0.25° × 0.25°) and temporal resolutions (hourly), and the representation of atmospheric processes has been further improved. In this study, the ERA5 reanalysis data is matched to SatCORPS data in the same way as GPM_3IMERGHH.

The LTS is expressed as the difference of potential temperature between 700 hPa and the surface (Klein and Hartmann, 1993). For the horizontal temperature advection at the surface ($SST_{adv}$), it is expressed in spherical coordinates as Jian et al. (2021) and Qu et al. (2015):

$$SST_{adv} = -\frac{u}{R_E \cos\phi}\frac{\partial SST}{\partial\lambda} + \frac{v}{R_E}\frac{\partial SST}{\partial\phi} \tag{2}$$

where $R_E$ is the mean Earth radius, SST is the sea surface temperature, u and v are the eastward and northward horizontal 10 m wind components, respectively. $\Phi$ and $\lambda$ represent the radians of latitude and longitude. A positive/negative $SST_{adv}$ indicates warm/cold advection, which influences the surface latent and sensible heat fluxes then the moisture transport within the cloud layer and the cloud thickness (George and Wood, 2010) and, consequently, influences the cloud liquid water.

## 3 Results

### 3.1 LWP adjustments vary alongside microphysical-dynamical conditions

Figure 1 shows the normalized joint histograms of $N_d$ and LWP in log-log space for all samples in the AUW and ECS regions. The complete pictures of all available daytime are presented in Fig. S2. The $N_d$-LWP relationships show similar patterns during daytime in each region, but different results in the two regions. The overall LWP adjustments are –0.31 in the AUW and 0.02 in the ECS region. For $N_d < \sim300$ cm$^{-3}$, LWP decreases with increased $N_d$, which is typically attributed to sedimentation-entrainment feedback (Ackerman et al., 2004) and evaporation-entrainment feedback (Small et al., 2009), leading to negative LWP adjustments in both regions. However, LWP begins to rise at high $N_d$ (> ~300 cm$^{-3}$), exhibiting an overall V shape, particularly in the ECS region, where clouds are downwind of the major emission sources of China. 18% of the samples in the ECS region exhibited $N_d$ values exceeding 300 cm$^{-3}$. To investigate whether the positive $N_d$-LWP relationship is influenced by broken scenes, we assessed the sensitivity of our results to CF. As shown in Fig. S3, the rise in LWP at high $N_d$ coincides with an increase in CF. The average CF for samples with $N_d >$ 300 cm$^{-3}$ is 86%. Additionally, the positive $N_d$-LWP relationship persists in both overcast (CF > 80%) and broken (CF < 80%) cloud scenes. This consistency indicates that the observed LWP increase at high $N_d$ is unlikely to be an artifact of broken-cloud scenes.

The V shape observed in our results differs from the inverted-V shape reported in previous studies (Glassmeier et al., 2021; Gryspeerdt et al., 2019). Specifically, it is characterized by the absence of an ascending branch at low $N_d$ and the emergence of an ascending branch at high $N_d$. The inverted-V shape is typically associated with positive LWP adjustments at low $N_d$, which have been linked to precipitation suppression (Albrecht et al., 1995). That is, as increasing $N_d$, the reduced $r_e$

may enhance the stability against coalescence and suppress the precipitation and loss of LWP (Albrecht, 1989; Glassmeier et al., 2021). The positive slopes are often observed in very pristine environments (Gryspeerdt et al., 2023), especially when $N_d$ is below approximately 10 cm$^{-3}$ (Fons et al., 2023; Goren et al., 2025). In contrast, in this study, 98% of the AUW samples exhibit $N_d$ values exceeding 15 cm$^{-3}$, and 99% of the ECS samples have $N_d$ greater than 30 cm$^{-3}$. Therefore, we did not find this positive slope of the inverted-V shape. Nevertheless, the LWP increasing signal resulting from precipitation suppression is still detectable in our study. For instance, samples with $r_e > 14$ μm—conditions more likely to contain drizzle (Rosenfeld et al., 2012)—still exhibit a weaker negative LWP adjustment than those with $r_e < 14$ μm (Fig. 1, –0.22 vs. –0.47 in the AUW and -0.13 vs. -0.23 in the ECS), consistent with the results of Zhou and Feingold (2023) in the northwestern Atlantic. It suggests that in drizzle-like samples, the precipitation suppression partially offsets the dominant LWP reduction caused by the entrainment effect, resulting in a weak decrease in LWP with increasing $N_d$ compared to non-drizzle samples.

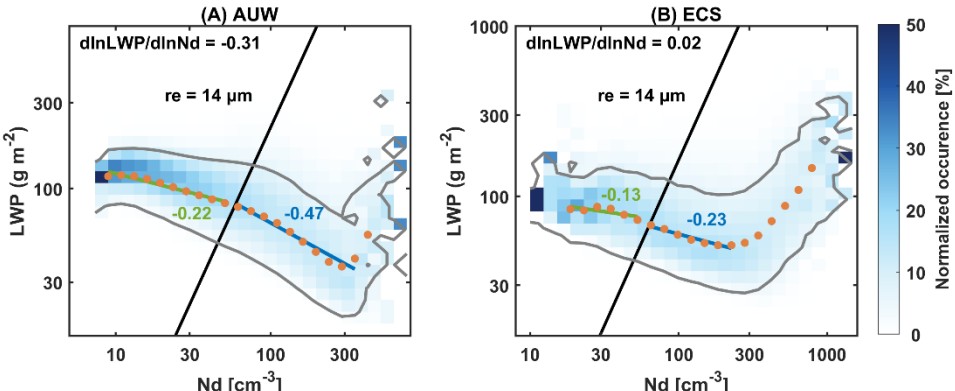

**Figure 1: Joint histograms of $N_d$ and LWP in log-log space in the AUW and ECS regions.** The column of each $N_d$ bin is normalized. The black lines are fitted based on the bins in the joint histogram with the effective radius ($r_e$) closest to 14 μm. The gray lines represent the contour of 5% occurrence. Orange dots represent the median LWP in each $N_d$ bins with a sample size greater than 50. The green and blue lines are regression slopes for the orange points with $r_e$ above and below 14 μm, respectively.

In this study, the ascending branch of the V shape at high $N_d$ condition (> ~300 cm$^{-3}$) is the main reason for the overall positive LWP adjustments in the ECS region. Positive sensitivity of LWP to $N_d$ perturbations over the ECS has been reported but not fully understood (Bender et al., 2019; Gryspeerdt et al., 2019; Michibata et al., 2016; Zhang et al., 2021). Here, our results indicate a strong transition in meteorological conditions across the turning point of V shape (Fig. 2), suggesting large-scale meteorology as a possible driver.

Meteorological conditions significantly modulate cloud microphysical processes (e.g., cloud droplet activation, condensation, entrainment, collision-coalescence, and precipitation) (Feingold et al., 2025), which in turn alter both the sign and magnitude of LWP adjustments, particularly within the sharp environmental transition from coastal to offshore areas in

the ECS region. Kuroshio Current produces a sharp SST gradient in the ECS region (shown in Fig. S4A), leading to a distinct transition in boundary layer thermodynamic structure and cloud properties from the coast to offshore areas (Liu et al., 2016). Following Rosenfeld et al. (2019), we categorize the clouds into three regimes, i.e., Sc (LTS > 18 K), Sc to Cu transition (14

K ⩽ LTS ⩽ 18 K), and Cu (LTS < 14 K) (Fig. S4, B, C, and D). Sc presents over a cooler sea surface along the coast (Fig. S4, A and B). The coastal distribution suggests that most of Sc may be advected from the Sc region in the southeast Chinese plain (Klein and Hartmann, 1993). According to the cloud advection scheme by Miller et al. (2018), cloud advection can be approximated as a translation of the cloud field with the wind field. The advection height is assumed to correspond to the height of the cloud top. Based on the 700 hPa wind field (Fig. S4A), it is plausible that Sc in the ECS region is possibly

advected from the southeast Chinese plain. As air moves offshore, the cloud layer decouples with the surface mixed layer over the warmer sea surface—a process known as the "deepening-warming mechanism" (Albrecht et al., 1995). In this decoupled boundary layer, Cu forms in the moist and unstable subcloud layer and rises to the upper cloud layer, resulting in a locally cumulus-coupled MBL.

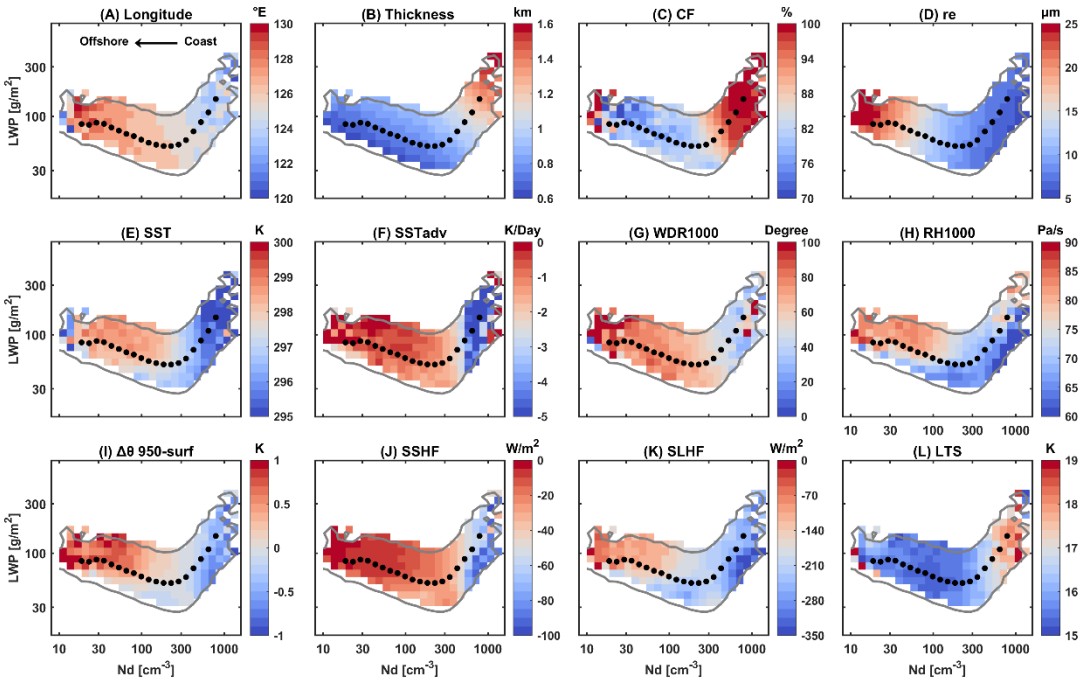

**Figure 2: Distributions of meteorological conditions in $N_d$-LWP log-log space in the ECS region.** The color scale represents the median values in each bin. Only bins with an occurrence of at least 5% are shown, bounded by the gray lines. (A) Longitude. (B) Cloud thickness. (C) Cloud fraction (CF). (D) Cloud effective radius ($r_e$). (E) Sea surface temperature (SST). (F) Horizontal temperature advection at the surface (SST$_{adv}$). (G) Wind direction on 1000 hPa. 0° indicates a northerly wind. (H) Relative humidity on 1000 hPa (RH1000). (I) The potential temperature difference between 950 hPa and 2 m above

the sea surface ($\Delta\theta_{950\text{-surf}}$), a proxy of the sub-cloud layer stability. (J) Surface sensible heat flux (SSHF). (K) Surface latent

heat flux (SLHF). For the vertical fluxes, the negative is upwards. (L) Lower-tropospheric stability (LTS). Black dots represent the median LWP in each $N_d$ bins with a sample size greater than 50.

The cloud samples in the ascending branch are concentrated west of 125°E and dominated by continental air masses (Fig. 2A), which are characterized by strong northerly cold air advection at the surface that destabilizes the air-sea interface (Fig. 2, F and G). The potential temperature difference between 950 hPa and 2 m above the sea surface ($\Delta\theta_{950\text{-surf}}$) is calculated as an indicator of sub-cloud layer stability, revealing an extremely unstable sub-cloud layer in the ascending branch (Fig. 2I). Northerly winds transport relatively dry, cold, aerosol-rich air across the warm ocean (Fig. 2, F, G, and H). This destabilizes the sub-cloud layer and intensifies the upward fluxes of sensible and latent heat from sea surface into the atmosphere (Fig. 2, I, J and K) (Long et al., 2020), raising saturation water vapor pressure and facilitating cloud droplet activation. Additionally, high LTS along the coast (Fig. 2L) suppresses vertical mixing at cloud top (Scott et al., 2020), allowing activated droplets to accumulate more liquid water with thicker clouds (Fig. 2B) and higher CF (Fig. 2C). These conditions jointly elevate both $N_d$ and LWP, forming the ascending branch of the V shape pattern.

While cold air outbreaks (CAOs) also contribute to the observed increases in both $N_d$ and LWP, our analysis suggests that cold air advection is a more consistent and seasonally pervasive driver and CAOs represent a strong form of cold air advection. Following Papritz et al. (2015), the Cold Air Outbreak Index (CAOI) was calculated as the difference in potential temperature between the surface skin and 850 hPa. CAO events are identified when CAOI > 0. Our results indicate that CAOs are most pronounced in autumn and winter, with no significant occurrence in spring (Fig. S5). Results of summer are statistically insignificant due to the limited samples (3%), particularly after excluding cases with strong precipitation (GPM = 0 mm hr$^{-1}$). The seasonal variations are consistent with the East Asian monsoon, where strong northerly winds prevail in winter but weaken in spring (Liu et al., 2016), leading to reduced CAOI. In contrast, the impacts of cold air advection are prevalent throughout the seasons (Fig. S6), making it a more plausible reason for the observed sub-cloud destabilization and subsequent increases in $N_d$ and LWP.

In the AUW region, LWP also increases when $N_d$ exceeds 300 cm$^{-3}$. However, the region is relatively clean with only 0.02% of all samples exhibiting $N_d$ above ~300 cm$^{-3}$. Given the limited sample size, these results are not statistically representative, and only a brief discussion is provided here. Samples with $N_d$ > ~300 cm$^{-3}$ still demonstrate distinct meteorological conditions compared to samples with $N_d$ < ~300 cm$^{-3}$ (Fig. S7). In contrast to the ECS region, pollution sources in the AUW region originate from lower latitudes (Fig. S7A). This may be attributed to the influence of warm and moist environment over the warm ocean with weak large-scale subsidence (Fig. S7, E, H, and L), which promote cloud droplet activation and consequently lead to positive LWP adjustments at high $N_d$.

The above results suggest that the impact of large-scale meteorology on cloud microphysical processes ultimately determines the pattern of LWP adjustment. Previous studies employed various methods to exclude environmental confounding factors, such as opportunistic experiments from ship-track or volcano eruptions (Chen et al., 2022; Toll et al., 2019), where an overall weak LWP adjustment was observed. For satellite studies, Rosenfeld et al. (2019) pointed out that cloud thickness (H)

constrained most of the meteorological impacts, and $N_d$ explained nearly half of the LWP variability for a given H. They demonstrated an overall positive LWP adjustment when separating H. However, we find that LWP adjustments become negative after constraining H in the intervals of Fig. 3 (B and E), indicating the dominant effect of entrainment-feedbacks. The discrepancy may arise from their focus on samples in convective cores (top 10% of cloud optical thickness), which are closer to adiabatic, whereas our samples suggest more exchange with the free atmosphere.

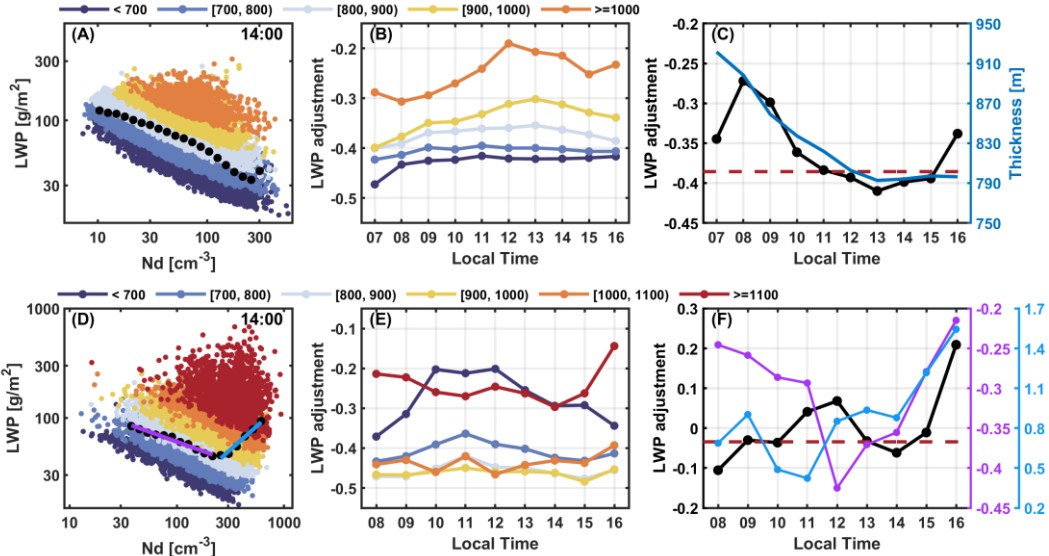

**Figure 3: LWP adjustments in log-log spaces and their diurnal patterns in two typical regions (the west of Australia, AUW and the East China Sea, ECS).** Cloud samples are scattered in $N_d$-LWP log space at 1400 LT in the (A) AUW and (D) ECS region. The complete pictures of all available daytime are presented in Fig. S11. Colored dots are samples in different cloud thickness (H) bins (unit: m). Black dots represent the median LWP in each $N_d$ bin. The colored lines are the fits of black
dots at different stages in the ECS region. Diurnal variations of LWP adjustments binned by H in the (B) AUW and (E) ECS regions are shown. Colored lines in (F) are diurnal variations of different stages in (D), while black lines in (C) and (F) are the overall diurnal variations of LWP adjustments in two regions, respectively. The blue line in (C) represents the diurnal variation of H. Red dashed lines represent the average LWP adjustments during MODIS Terra (1030 LT) and Aqua (1330 LT) overpasses, –0.39 for the AUW region (C) and –0.03 for the ECS region (F).

Here, our results indicate the physical significance of constraining H. In the AUW region, negative LWP adjustments become weaker as H increases (Fig. 3B). H alters LWP adjustments by influencing cloud microphysical processes, such as promoting condensation growth (Fons et al., 2023). Thicker clouds with higher cloud-top $r_e$ are less sensitive to entrainment-feedbacks with increasing $N_d$ compared to thinner clouds. In other words, LWP in different H intervals responds differently to
285 $N_d$, so it is necessary to restrict H to exclude the effects of covariations. However, in the ECS region, negative LWP adjustments for clouds with H < 900 m become stronger with increasing H, while for clouds with H > 900 m, quite the contrary: it weakens

with increasing H (Fig. 3E). The bidirectional sensitivity of LWP adjustments to H is likely attributed to distinct mixing characteristics among different cloud regimes in the ECS region. Constraining H in the ECS region restricts a majority of mechanisms influencing cloud vertical development. Cloud thickness typically serves as a mediator for large-scale meteorology (such as cold air advection, LTS, and surface heat fluxes) to influence LWP. These processes are particularly evident in the ECS region, where the increase in LWP at high $N_d$ corresponds with an increase in cloud thickness (Fig. 2B). Therefore, the stratification of cloud thickness can isolate a significant portion of covariations, highlighting the impact of $N_d$ on LWP.

In summary, the above results reveal that LWP adjustments strongly depend on microphysical-dynamical processes (e.g., precipitation suppression, and entrainment feedbacks) and large-scale meteorology (e.g., cold air advection and the stability of MBL). Given that some of these factors display diurnal variations in response to the solar radiation cycle, LWP adjustments would also exhibit diurnal patterns (black lines in Fig. 3, C and F). We surmise that the prevailing dynamic conditions at any given time are responsible for the observed diurnal variations of LWP adjustments. To verify this hypothesis, we investigated the diurnal variations in LWP adjustments and their potential influencing factors.

## 3.2 How LWP adjustments change over the diurnal scale and associated mechanisms

LWP adjustments exhibit pronounced diurnal variations with distinct regional contrasts. In the AUW region, the negative LWP adjustments strengthen from around 0800 LT to 1300 LT, reaching their strongest value at –0.41, and then weaken to –0.34 (black line in Fig. 3C). In the ECS region, the positive LWP adjustments exhibit two local peaks during the observation period, occurring at 1200 LT and 1600 LT, with peak values of 0.07 and 0.21, respectively (black line in Fig. 3F). And two local minima LWP adjustments are observed at 0800 LT and 1400 LT, with values of –0.11 and –0.06, respectively. The results highlight the limitations of using the sparse polar-orbiting satellite observations to represent LWP adjustment at specific times. For example, MODIS overpass averages (red dashed line in Fig. 3C) overestimate the intensity of negative LWP adjustment in the AUW region by 44% at 0800 LT. In the ECS region, the intensity of negative LWP adjustments are underestimated by 73% at 0800 LT, while the intensity of positive adjustments at 1600 LT are underestimated by 114% (Fig. 3F). Such biases can lead to substantial errors in estimation of ACI (see Section 4 for details).

We first analyze the role of meteorological factors in driving the diurnal variations of LWP adjustments (Fig. S8). Overall, the covariance of a single meteorological factor affects only the magnitude of LWP adjustments. In the AUW region, the lower LTS corresponds to weaker negative LWP adjustments. Samples with relatively low LTS are characterized by larger $r_e$ in the $N_d$-LWP space (Fig. S9), leading to stronger precipitation suppression by increasing $N_d$ and thus a weaker negative LWP adjustment. In the ECS region, stronger cold air advection corresponds to greater sensible and latent heat fluxes, resulting in more positive LWP adjustments, which is consistent with the findings presented in the previous section. The diurnal variations of LWP adjustments cannot be explained by a single meteorological factor. Therefore, it is necessary to start with the diurnal variations of cloud properties to analyze the mechanisms behind the diurnal LWP adjustment patterns.

The AUW region is one of the subtropical Sc regions over the eastern part of the ocean away from continents (Klein and

Hartmann, 1993), characterized by large LTS and strong large-scale subsidence (Fig. S10), which are favorable for the formation of Sc. Figure 4 depicts the diurnal variations of cloud properties in the Sc-like AUW region. The diurnal variation of LWP shows a typical pattern with a peak in the morning and a gradual reduction until early afternoon. According to previous studies, this pattern is subject to the diurnal cycle of solar insolation (Bretherton et al., 2004; Mechoso et al., 2014; Wood et al., 2002). Specifically, during the daytime, solar radiation absorption within the cloud layer and long-wave cooling at the

cloud top drive the turbulent mixing within the cloud layer and inhibit turbulence to the sea surface, thus leading to the decoupling of the cloud-topped marine boundary layer (MBL) (Duynkerke and Hignett, 1993; Ghosh et al., 2005; Slingo et al., 1982). As decoupling cuts off the moisture source from the sea surface, the imbalance between entrainment drying and upward moisture flux may thin the cloud layer. The decrease of LWP before 1300 LT is primarily attributed to the lifting of the cloud base, indicating that entrainment drying originates from evaporation at the cloud base, which is in line with an early

modeling study for typical Sc cloud regimes (Bougeault, 1985). After 1300 LT, the gradual reduction of solar heating hinders the intensification of decoupling and helps rebuild the turbulence between the cloud and subcloud layer. Therefore, LWP increases after 1300 LT likely due to the reconstruction of turbulence.

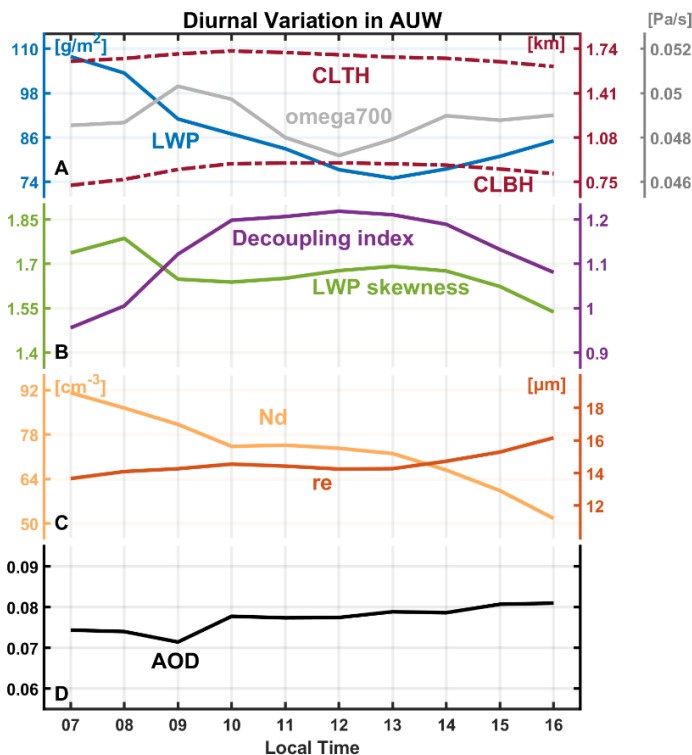

**Figure 4: Diurnal patterns in the AUW region.** (A) Cloud liquid water path (LWP), cloud-top height (CLTH), cloud base

height (CLBH), and vertical velocity on 700 hPa (omega700, positive values indicate downdraft) from ERA5 reanalysis. (B) LWP skewness and decoupling index in the AUW region. (C) Cloud droplet number concentration ($N_d$) and effective radius ($r_e$). (D) Aerosol optical depth (AOD).

Following the quantification method of Zheng et al. (2018) and Kazil et al. (2017), this study presents auxiliary verifications of the decoupling process. First, according to Zheng et al. (2018), decoupling of the subtropical Sc decks during cold advection is often unstable (negative temperature advection). The formation of Cu beneath the Sc will render local coupling by feeding moisture into the upper cloud layer, thus causing a positive skewness of the probability density function (PDF) of LWP. Therefore, the skewness of the LWP PDF can be used to estimate the degree of decoupling for each cloud sample:

$$\text{skewness} = \frac{E(x - u)^3}{\sigma^3} \tag{3}$$

where E is the expected value, μ and σ is the mean and standard deviation of x, respectively. Positive skewness indicates more data tends to be distributed to the right, and vice versa. Larger LWP skewness indicates a larger decoupling degree.

As shown in Fig. 4, LWP skewness increases before 1300 LT and then decreases, illustrating the decoupling process discussed above. Note that while the cumulus penetration alters LWP, small variations in LWP skewness suggest that it cannot be directly compared with the reduction of LWP caused by decoupling, thus having no evident effect on the diurnal variation of LWP over the AUW region. Additionally, due to the fluctuation of LWP skewness before 0900 LT, another decoupling index defined by Kazil et al. (2017) is used for further indication, quantifying the relative position between the CLBH and the lifting condensation level (LCL). A larger index implies a stronger degree of decoupling:

$$\text{decoupling index} = \frac{CLBH - LCL}{LCL} \tag{4}$$

LCL is derived from ERA5 reanalysis following Wood and Bretherton (2006). The two indices support each other and confirm the decoupling process.

$N_d$ continually declines from 0700 LT to 1600 LT and $r_e$ does not change significantly before 1200 LT and then rises. In contrast, there is no evident diurnal variation of AOD in the AUW, which is reasonable in the remote ocean area but insufficient to explain the diurnal variations of $N_d$ and $r_e$. This suggests that other factors rather than aerosols may be responsible for the diurnal variations of $N_d$ and $r_e$ over the AUW region. Combining the nature of the decoupling process and diurnal patterns of cloud properties in Fig. 4, we discuss the possible mechanisms for the diurnal variation of $N_d$ and $r_e$ based on earlier cloud microphysics studies. According to Verlinden (2018), the shortwave heating counteracts longwave cooling during daytime, resulting in weakening of cloud-top entrainment. Meanwhile, the decoupling that cuts off moisture transport suppresses condensational growth. The combination of these two processes may lead to little variations in $r_e$ before 1200 LT. Additionally, the decoupling process leads to the suppression of both surface moisture transport and cloud base updrafts, which may in turn reduce the supersaturation and hence the number of activated cloud droplets. This may explain the continuous decrease in $N_d$ before 1300 LT. Furthermore, according to the relationship between CLTH, $w_s$ (always negative), and entrainment rate ($w_e$) ($\frac{dCLTH}{dt} = w_s + w_e$) in the mixed-layer model framework (Painemal et al., 2013), we explain the variations after 1200 LT. CLTH begins to decrease after 1200 LT, suggesting an intensification of large-scale subsidence ($w_s$, always negative in Sc

region) and/or a weakening of entrainment rate ($w_e$). Large-scale subsidence on 700 hPa from ERA5 reanalysis becomes stronger (gray line in Fig. 4A). It may enhance the temperature-inversion jump, which will in turn decrease the entrainment rate (Painemal et al., 2013). During this period, the condensational growth by the reconstructed water vapor supply will enhance $r_e$. Meanwhile, the coalescence process, enhanced by an increase in $r_e$ leads to a decrease in $N_d$. This process could be more dominant than the increase in activated cloud droplets caused by water vapor reestablishment for an increase in $N_d$ to be

observed in this study.

Based on the diurnal mechanisms of MBL discussed above, the diurnal LWP adjustment pattern is primarily a consequence of the influence of these diurnal-related mechanisms on the relationship between $N_d$ and LWP. In the AUW region, the diurnal variations of the overall LWP adjustments (black line in Fig. 3C) and cloud thickness (blue line in Fig. 3C) demonstrate a strong consistency with a turning point at 1300 LT. The variation of LWP adjustment here is mainly attributed

to the gradual thinning of clouds, which reflects the differential LWP responses to $N_d$ with varying H. LWP adjustment becomes more negative with the thinning of clouds, which is consistent with the results in Fig. 3B. After 1300 LT, cloud thickness remains almost unchanged. The variation in LWP adjustments is mainly governed by the weakening of entrainment due to the intensification of large-scale subsidence (Fig. 4A). During this time, the weakening of the entrainment process leads to a weakening of the negative LWP adjustments over the AUW region.

In contrast, conditions of MBL in the ECS region are more complicated. As mentioned in the last section, the ECS is a Sc-Cu transition region due to the "deepening-warming" process. Under this condition, MBL is seldom fully decoupled but exhibits local cumulus coupling. Apparently, LWP skewness is a more appropriate indicator to reflect cumulus coupling in this region. For diurnal variations in the ECS in Fig. 5, there is a general decrease in LWP before 1300 LT, followed by an increase. This is in contrast to the pronounced cloud thinning observed in the AUW region due to the decoupling of MBL by

solar heating. In the ECS region, the overall change of LWP is not significant (less than 10 g/m²). Since MBL is never fully coupled, these minor observed changes are mainly caused by local cumulus coupling. The variations of LWP and LWP skewness exhibit a strong consistency. We also calculate the coefficient of variation ($c_v$) of CLOT to represent the uniformity of each cloud sample. $c_v$ is defined as the standard deviation ($\sigma$) divided by the mean($\mu$):

$$c_v = \frac{\sigma}{\mu} \tag{5}$$

The smaller the $c_v$ is, the less dispersion there is among the cloud pixels in the cloud sample, resulting in a more uniform sample. It turns out that the cloud layer is influenced primarily by the strength of cumulus coupling, rather than other factors.

In the ECS region, the weakest cumulus activity occurs at 1300 LT (the lowest LWP skewness in Fig. 5B), which may be attributed to solar insolation. In the Sc to Cu transition region, the decoupled cloud layer and subcloud layer are often separated by a stable transition layer, which has been widely observed by the Atlantic Stratocumulus Transition Experiment

(ASTEX) conducted over the northeast Atlantic Ocean. Based on ASTEX, Rogers et al. (1995) suggested that the shortwave radiation would hinder convection during daytime by increasing the stability of the transition layer. Miller et al. (1998) extended this theory to the diurnal variations and believed that the diurnal variation of Cu development was regulated by the

stability of the transition layer.

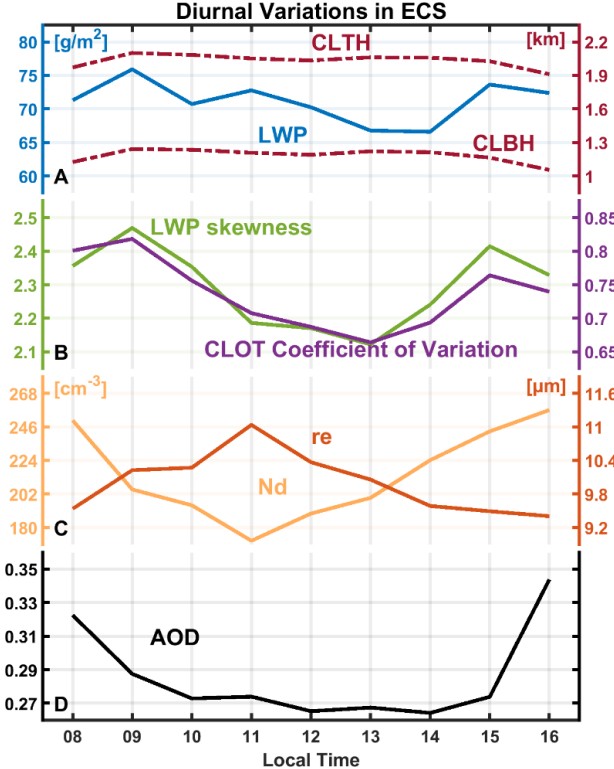

**Figure 5: Diurnal patterns in the ECS region.** (A) Cloud liquid water path (LWP), cloud-top height (CLTH) and cloud base height (CLBH). (B) LWP skewness and coefficient of variation ($c_v$) of cloud optical depth (CLOT) in the AUW region. (C) Cloud droplet number concentration ($N_d$) and effective radius ($r_e$). (D) Aerosol optical depth (AOD).

In terms of microphysical properties, $N_d$ in the ECS decreases before 1100 LT and then increases. Variations of $r_e$ are just the opposite except insignificant change since 1400 LT. The crucial mechanism leading to such changes may be attributed to the weakest entrainment drying at 1100 LT, resulting in the highest values of $r_e$ and the lowest values of $N_d$. Such diurnal variations in entrainment have also been observed in other coastal areas. Caldwell et al. (2005) reported the weakest entrainment rate at 1100 LT during the East Pacific Investigation of Climate (EPIC) stratocumulus cruise in 2001. Painemal et al. (2017) found the minimum of entrainment occurred between 0900-1100 LT over the northeast Pacific region, attributing the diurnal pattern to the turbulence caused by long-wave radiative cooling. Additionally, other factors may also contribute to the diurnal variations of $N_d$ and $r_e$. For example, the changes before 1100 LT may include the impacts of reducing aerosol loadings. Subsidence from both cloud top and bottom occurred after 1400 LT may limit the entrainment and the continuous decline of $r_e$. Cumulus coupling may also contribute to the increase of $N_d$, and Martin et al. (1995) found a local increase in $N_d$ induced by the intrusion of cumulus clouds during ASTEX.

Based on the above mechanisms, the diurnal variation of LWP in the ECS region is relatively small, yet $N_d$ exhibits a distinct diurnal pattern. Changes in $N_d$ determine the slope of LWP adjustments at the ascending and descending branches of the V shape that correspond to different meteorological conditions. The $N_d$ turning point between the two stages exhibits the same diurnal variation as the average $N_d$ (Fig. S12). Before noon, a decrease in $N_d$ weakens the positive branch (blue line in Fig. 3F), while the negative branch intensifies (purple line in Fig. 3F). Collectively, the two branches determine the diurnal

variation of the overall LWP adjustments.

Given that the samples span four years across all seasons, a sensitivity analysis was conducted to assess the impact of seasonal variations. Overall, the diurnal LWP adjustment pattern is not sensitive to seasonal changes in the AUW region (black lines in Figs. S13F-S16F compared to Fig. 3F). Since the AUW region is a persistent stratocumulus area, the diurnal variations of cloud thickness remain consistent across all seasons, with the thickest clouds in the morning and the thinnest in the early

afternoon, followed by a slow increase. It suggests that the decoupling process in the persistent Sc region is insensitive to the seasonal changes, leading to similar patterns of LWP adjustments. The ECS region exhibits seasonal differences (Figs. S13-S16). Among the total samples (173181), spring, summer, autumn, and winter account for 31%, 3%, 22%, and 44%, respectively. Due to the limited summer samples (3%), their results are statistically insignificant, especially after eliminating the samples with precipitation by applying the threshold (GPM = 0 mm hr$^{-1}$). The LWP adjustments in other seasons exhibit

similar diurnal patterns and magnitudes, peaking at noon (black lines in Figs. S13F, S15-S16F). This similarity may be due to the weak seasonal variations in the diurnal patterns of LWP and $N_d$ (not shown). The diurnal patterns of the ascending branch of the V shape during spring and winter align with the overall results (blue lines in Figs. S13F and S16F compared to Fig. 3F). The $N_d$ minimum occurring at 1100 LT coincides with the weakest positive LWP adjustments in the ascending branch. Among all seasons, autumn exhibits the lowest $N_d$, corresponding to the weakest positive LWP adjustments in the ascending branch

(~50%/31% lower than spring/winter) and the largest diurnal fluctuations (Fig. S15F). This may be attributed to the weakest cold air advection during autumn (Fig. S6). The diurnal pattern of the descending branch in spring differs from other seasons (purple line in Fig. S13F), possibly due to the diurnal variation of entrainment rate which can be illustrated by the variation of CLTH. Here, based on the relationship between CLTH, $w_s$ (always negative) and entrainment rate ($w_e$) ($\frac{dCLTH}{dt} = w_s + w_e$) (Painemal et al., 2017), the diurnal variations of $w_e$ (entrainment rate) can be qualitatively analyzed with the diurnal variations

of CLTH and large-scale subsidence ($w_s$) (Fig. S17). Before 1400 LT, the variation of large-scale subsidence is unrelated to CLTH, thus the change in CLTH can only be attributed to the entrainment rate. The entrainment rate weakens before 1200 LT, leading to a weakening of the negative LWP adjustments. It then strengthens until 1400 LT, which enhances the negative LWP adjustments. After 1400 LT, the observed decrease in CLTH is mainly attributed to an increase in large-scale subsidence. The enhanced subsidence further suppresses the entrainment rate, thereby weakening the negative LWP adjustments.

To summarize, Figure 6 depicts schematics of the dominant mechanisms in the two regions. In the AUW region, the primary mechanism behind the diurnal variation of LWP adjustments is the cloud thinning driven by MBL decoupling before 1300 LT. After 1300 LT, the gradual weakening of cloud-top entrainment mitigates the negative LWP adjustments. The diurnal

variation of LWP adjustments in the ECS region is jointly determined by the ascending and descending branches of the V shape, which is linked to the microphysical processes responsible for the diurnal variations of $N_d$ (e.g., entrainment drying).

Failure to accurately capture these diurnal variations in LWP adjustments and the underlying physical processes in observational studies may result in substantial inaccuracies in the quantification of regional and global LWP adjustments, and the associated radiative forcing.

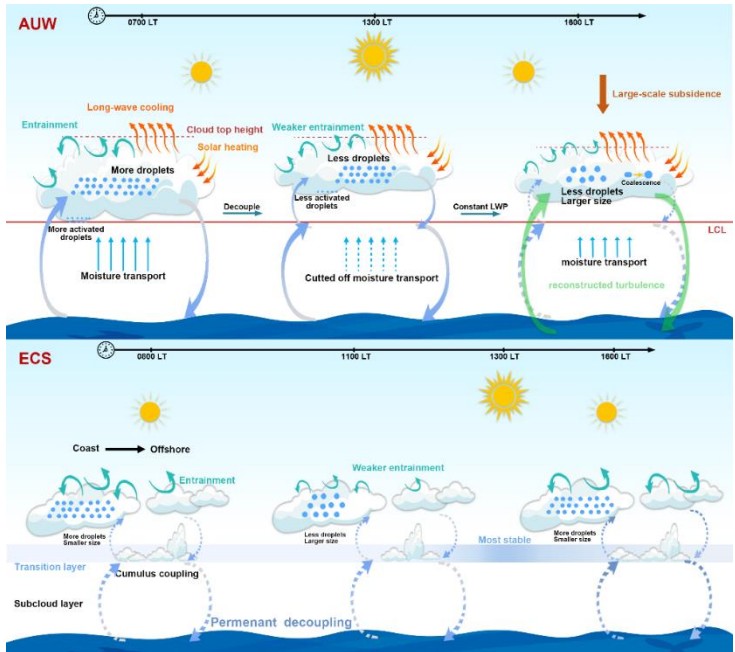

**Figure 6: Schematics of diurnal dominant mechanisms observed in the AUW and ECS. See text for details.** Only the
primary mechanisms are presented, while the relatively unimportant ones are omitted. Note that we represent the lifting condensation level (LCL) and transition layer at the same altitude for intuition. However, this depiction does not imply that their heights remain constant throughout the diurnal variation.

## 4 Discussion

As discussed above, regional geostationary observations reveal the significant impact of regional diurnal dynamic
processes on LWP adjustments, ranging from –0.41 to –0.27 in the AUW and from –0.11 to 0.21 in the ECS. Assuming a constant LWP adjustment based on polar-orbiting snapshots, rather than considering its diurnal variations will ultimately affect the estimation of the aerosol indirect effect. The cloud albedo ($A_c$) susceptibility to aerosols perturbations is estimated as (Bellouin et al. 2020):

$$S = \frac{dA_C}{d\ln N_d} = \frac{A_C(1 - A_C)}{3}\left(1 + \frac{5}{2}\frac{d\ln LWP}{d\ln N_d}\right) \tag{6}$$

where S is the sensitivity of cloud albedo to $N_d$. $A_c$ is calculated from $\tau$ based on a general expression for two-stream

approximation solution (Glenn et al., 2020):

$$A_c = \frac{\tau}{13.33 + \tau} \qquad (7)$$

The first term of Eq. (6) refers to the changes in albedo due to the changes in $N_d$, while holding the LWP (i.e. Twomey effect). The second term, which accounts for LWP adjustment, can regulate the Twomey effect. The Twomey effect is completely offset when $\frac{d \ln \mathrm{LWP}}{d \ln N_d}$ equals –2/5. Figure 7 shows the diurnal variations of S, calculated with Eq. (6) using the diurnal variations of both $A_c$ and LWP adjustments. To isolate their individual influence, S was calculated using MODIS-averaged value for either $A_c$ or LWP adjustments while retaining the diurnal variation for the other (Fig. S18). Given the minimal diurnal fluctuation in $\frac{A_C(1-A_C)}{3}$, the diurnal variations of S are mainly controlled by LWP adjustments. According to Fig. 7, if S is evaluated only at fixed moments (e.g. the average value during MODIS overpasses for Terra at 1030 LT and Aqua at 1330 LT), the cooling effect of S is consistently underestimated before 1100 LT, with a maximum bias of 89% at 0800 LT. At 1300 LT, S even turns negative, suggesting that albedo decreases with increasing $N_d$, which has been reported in previous studies (Zhang et al., 2022). The negative S is possibly linked to strong decoupling over the AUW region at 1300 LT as discussed in Section 3.2. In the ECS region, the associated bias spans from a 24% overestimation at 0800 LT to a 40% underestimation at 1600 LT. The results highlight the critical need to account for diurnal variations of LWP adjustments when assessing the aerosol indirect effect. Future studies should incorporate geostationary observations or high-resolution simulations to better constrain the diurnal effects of LWP adjustments.

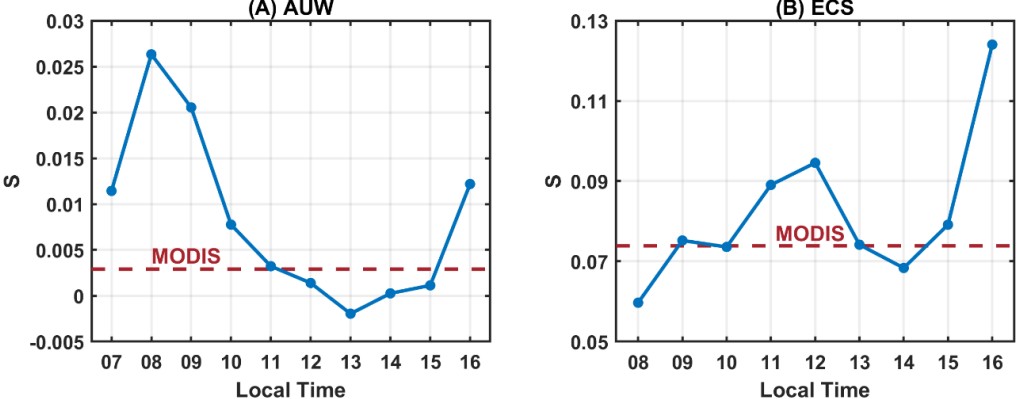

**Figure 7: The diurnal variations of S calculated by Eq. (6) in the (A) AUW and (B) ECS (blue lines).** The red dashed lines represent the average values during MODIS Terra (1030 LT) and Aqua (1330 LT) overpasses.

Our observed diurnal LWP adjustment pattern in the AUW region is consistent with Qiu et al. (2024)'s findings in the eastern North Atlantic, where thick-thin cloud transitions dominated daytime variability. However, the main drivers emphasized in the two studies are different. Qiu et al. (2024) calculated LWP adjustment within each 1° grid box to minimize the meteorological covariations and highlighted cloud-intrinsic evolution, whereas we retain these covariations and then

disentangle their influence by cloud thickness stratification analyses following Rosenfeld et al. (2019). Consequently, we
attribute the diurnal variations in LWP adjustments mainly to temporal changes in meteorological and dynamical conditions.
Additionally, after 1300 LT, cloud thickness remains relatively stable; the weakening of negative LWP adjustments is linked
to reduced entrainment as large-scale subsidence strengthens (Fig. 4A). Furthermore, we conduct the same analyses in the ECS
region with a completely different environmental background and obtain entirely different results. The $N_d$-LWP relationship
exhibits a V shape pattern, contrasting with the inverted-V shape reported in previous studies. The discrepancy likely results
from the covariations induced by the geographical dependence of samples. This demonstrates that the significant regional
differences in the diurnal variations of LWP adjustments, depending on aerosol loadings, cloud regimes and meteorological
conditions.

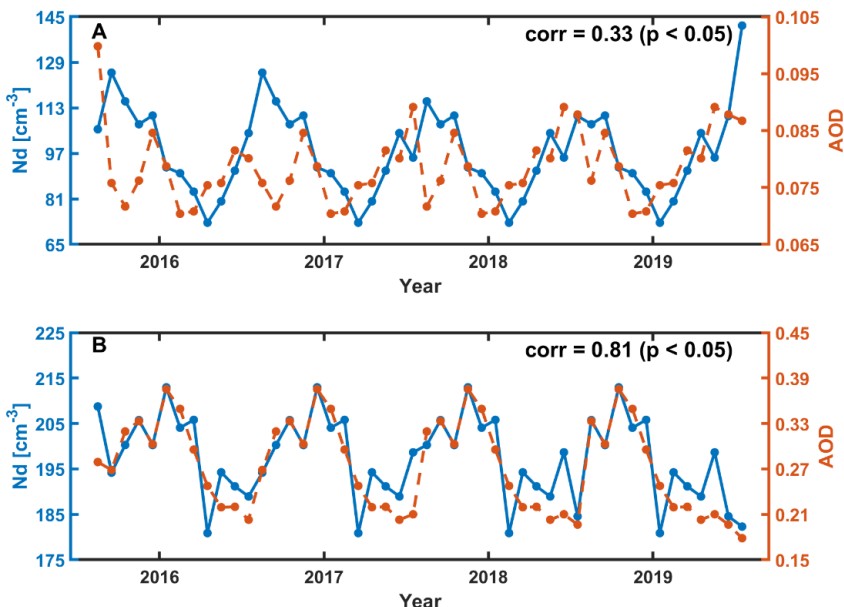

**Figure 8: 4-year long-term variations of $N_d$ and aerosol optical depth (AOD) from MERRA-2 at 1200 LT in the AUW
(A) and ECS (B) region.** The correlation coefficients (corr) between $N_d$ and AOD are 0.33 and 0.81 (significant at the 95%
confidence level), respectively.

It is worth noting that our results also reveal diurnal variations of $N_d$, a core indicator in ACI, which are also attributed to
the MBL diurnal processes. While previous studies have analyzed the long-term variations of $N_d$, highlighting the key role of
aerosols (Hu et al., 2021; Li et al., 2018; McCoy et al., 2015, 2018; Quaas et al., 2006), there is no good consistency between
them in diurnal variations. This discrepancy may stem from previous polar-orbiting satellite observations at fixed times have
overlooked the crucial role played by other physical mechanisms at different times. Figure 8 shows significant correlations
observed between the 4-year long-term variations of AOD and $N_d$ at 1200 LT in both regions, particularly in the ECS with a

correlation of 0.81. Meanwhile, both regions show the similar distribution patterns, with higher $N_d$ and smaller $r_e$ near the continental coastal area, aligning with the average AOD spatial distribution (spatial correlation coefficients of 0.84 in the AUW and 0.91 in the ECS) (Fig. S1), suggesting a pronounced impact of anthropogenic activities on cloud microphysical properties on a long-term scale. Note that the correlations between AOD and $N_d$ at certain fixed times are not statistically significant (not shown). This may be due to the relatively insignificant impact of aerosol effects at these moments, while other processes may exert a more pronounced influence. For example, strong boundary layer decoupling inhibits cloud droplet activations (Zeider et al., 2025). Mesoscale cloud organization can also introduce spatial heterogeneity in $N_d$ independent of aerosol loading (Zhou and Feingold, 2023). Future research should broaden its scope to investigate the effects of other influencing factors on $N_d$ at specific times, in addition to the role of aerosols. Moreover, in the context of global warming, whether these physical processes will be affected and consequently contribute to variations of $N_d$ deserves further investigation.

Several limitations should be acknowledged in this study. First, the time-dependence of LWP adjustments we discussed differs from the cloud evolution process, emphasizing diurnal variations caused by changes in dominant mechanisms at different times rather than tracking the evolution of individual clouds. This approach may introduce uncertainties into our results since the full cloud life cycle and evolution are not the same with diurnal variations. The full cloud lifetime evolution associated with LWP adjustments is not within the scope of this study and warrants further exploration. Additionally, given the scarcity of observational data at fine scales, certain mechanisms are indirectly inferred from the observational index (e.g., decoupling process inferred from LWP skewness), which needs further microphysical-process-based in-situ observations as well as model simulations. Finally, uncertainties of retrievals have been discussed in Data and Methods, which provides further context for the limitations of this study.

## 5 Conclusion

This study reveals the diurnal variations of LWP adjustments and the possible mechanisms contributing to these variations in two specific regions with significant differences in cloud regimes, environmental conditions, and aerosol loadings. Important findings from this investigation are as follows:

(1) In the AUW region, the overall negative LWP adjustments decrease from −0.27 to −0.41 before 1300 LT and then increase to −0.34. The diurnal variations of LWP adjustments are insensitive to seasonality. Cloud thickness in the AUW region serves as a confounder to separate the effects of meteorological covariations. The diurnal pattern is primarily associated with cloud thinning induced by decoupling process of MBL quantified by LWP skewness before 1300 LT and the weakening of entrainment induced by the intensification of large-scale subsidence after 1300 LT.

(2) In the ECS region, LWP increases at high $N_d$ (> ~300 cm$^{-3}$), leading to a V shape pattern of $N_d$-LWP relationship. Our results demonstrate a distinct transition in environmental conditions across the turning point of the V shape, indicating the V shape pattern is the result of meteorological covariations. Specifically, the aerosol-rich, relatively cold and dry air from continent reduces the stability of the sub-cloud layer, triggering the release of water vapor into

the boundary layer and subsequently promoting cloud droplet activation and development of thicker clouds. These processes collectively lead to an increase in both $N_d$ and LWP, resulting in a positive LWP adjustment at high $N_d$. The diurnal variations of LWP adjustments exhibit seasonal differences. Samples from winter and spring dominate the overall variations (accounting for 75% of the total samples). The diurnal LWP adjustment pattern is determined by the combined diurnal variations of the ascending and descending branches of the V shape, which is likely attributed to the diurnal variation of $N_d$ induced by entrainment.

(3) The results indicate an underestimation of the cloud albedo sensitivity to aerosol perturbations by up to 89% in the AUW region, while in the ECS region, the bias ranges from a 24% overestimation at 0700 LT to a 40% underestimation at 1600 LT. Furthermore, our results quantify the regional impact of boundary layer dynamic conditions on LWP adjustments. For example, the diurnal decoupling process in the AUW region results in a 219% variation of LWP adjustments within the daytime relative to the daily mean (the diurnal variation range divided by the daily mean), assuming other conditions remain relatively unchanged.

Our research provides a detailed discussion for the diurnal variations of LWP adjustments and how they are influenced by existed boundary layer mechanisms. We underscore the importance of fully considering the covariations with environmental conditions, indicating different potential influencing factors on cloud brightening and radiative forcing in terms of the regional and diurnal daytime scale. It is a highly time-dependent variable lacking quantification and should be taken into consideration of future research in aerosol indirect effects on climate.

**Data availability**

The datasets that support this study are all available to the public. The SatCORPS Himawari-8 product is available at https://asdc.larc.nasa.gov/project/CERES. The MERRA-2 product is available at https://disc.gsfc.nasa.gov/datasets/M2T1NXAER_5.12.4/summary?keywords=merra2. The GPM_3IMERGHHV07 is available at https://disc.gsfc.nasa.gov/datasets/GPM_3IMERGHH_07/summary?keywords=gpm%20imerg. ERA5 reanalysis data are available at https://cds.climate.copernicus.eu/. All data are available in the main text or the supporting information.

**Author contributions**

JiaL and YaW performed the analysis and organized the original manuscript. JimL and YaW conceptualized the study and reviewed the manuscript. WZ assisted in data analysis and validation. LZ and YuW assisted in the investigation and the final review and editing of the manuscript.

**Competing interests**

The contact author has declared that none of the authors has any competing interests.

**Acknowledgments**

We would like to acknowledge ChatGPT for its role in polishing the language for the text. We would like to acknowledge freepik.com for supporting icons used in our schematics ([www.freepik.com](www.freepik.com)).

**Financial support**

This work is supported by the following funding: Key Program of the National Natural Science Foundation of China (42430601), Major Program of the National Natural Science Foundation of China (42090030), National Natural Science Foundation of China (42175087), Science and Technology Project of Gansu Province (Outstanding Youth Fund, 24JRRA386).

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
