# Peer review of "Strong aerosol indirect radiative effect from dynamic-driven diurnal variations of cloud water adjustments"

_EGUsphere, 2024_

## Author Response (AR1)

**Updated Response to Reviewer #1's Comments:**

Jiayi Li et al. (Author)

**We would like to express our sincere gratitude to Reviewer #1 for the insightful and professional comments. We provide the following response to Reviewer #1's comments regarding the data and methods. Additionally, we have incorporated Reviewer #2's suggestions and further refined the results. To ensure consistent responses, we decide to reply to reviewer #1's comments again.**

**In the following text, the reviewers' comments are listed in black, our response is in blue, and changes to the text are highlighted in red. A version of the revised manuscript with tracked changes is also provided.**

**Important revision includes:**

1. According to Reviewer #1's comments, only the threshold of GPM = 0 mm/hr was used for the exclusion of precipitation. The corresponding result section has been modified. The $N_d$-LWP relationship no longer exhibits the initial increasing trend in both regions. The quantification of the impact of diurnal variations on radiative effects has been revised.

2. We have clarified the explanation of the mechanisms based on direct conclusions. This allows us to distinguish between these directly supported conclusions and inferences that rely on previous research with supporting evidence.

**Specific Responses:**

1. The retrievals of cloud properties:

The native resolution of the AHI is 2 km. Why use a microphysical product of 4 km? The low resolution renders the retrieval very sensitive to errors due to partial pixel feeling in most cases, except for the fully cloudy scenes. Therefore, the effects of cloud cover are confounded with those on LWP.

**Response:** Thanks for your great comment. Although the native resolution of AHI is higher, the official AHI dataset only provides the effective radius ($r_e$) Level-2 product for the 2.3 μm channel. In contrast, the SatCOPRS CERES Geostationary Satellite Edition 4 Himawari-8 product we used (CER_GEO_ED4_HIM08_NH_V01.2, CER_GEO_ED4_HIM08_SH_V01.2) has a coarser resolution but provides $r_e$ Level-2 product for the 3.9 μm channel using the Langley Research Center (LARC)s

SatCORPS algorithms in support of CERES project. This product with the 3.9 µm channel is considered more accurate for cloud droplet number concentration ($N_d$) retrievals because this channel better represents the cloud-top information, introducing less bias to the retrieval of $N_d$ (25%~38% for 2.3µm, less than 20% for 3.7 µm according to Grosvenor et al., 2018).

Additionally, the 3.9 µm $r_e$ from CER_GEO_ED4_HIM08 shows good consistency with the 3.7 µm $r_e$ from MODIS (Figure R1), supporting us in obtaining more accurate $N_d$. While the coarser resolution may impact the retrieval to some extent as the reviewer said, SatCOPRS CERES Geostationary Satellite cloud product only uses cloudy pixels based on the CERES Ed4 cloud mask (Trepte et al., 2019; Yost et al., 2021), thus largely avoiding the situation mentioned by the reviewer.

Considering the overall quality of the product and the existing precedent of using SatCOPRS CERES Geostationary Satellite products in studies for LWP adjustments (Qiu et al., 2024), we finally selected this dataset in our study.

[Figure]

**Figure R1. Comparison of the 3.7 µm *r_e* from MODIS and the 3.9 µm *r_e* from CER_GEO_ED4_HIM08_SH_V01.2.**

Why sub-sampling the data? Why is it 8 km at the NH and 6 km at the SH?

**Response:** Thank you for raising this question. The description of the data resolution in the manuscript was based on the data introduction available on the NASA EARTHDATA SEARCH (https://search.earthdata.nasa.gov/search/granules/collection-details?p=C1584977037-LARC_ASDC&pg[0][v]=f&pg[0][gsk]=-start_date&fpj=CERES&lat=0.0703125&long=-0.0703125). However, after reaching out to technical staff through the Earthdata forum for clarification, we learned that the information on the website is incorrect. The observation resolution of CERES_GOES_HIM08 is 2 km at nadir, and has been sub-sampled to 6 km for both the Northern and Southern Hemispheres. The sub-sampled resolution meets the needs of the CERES project without having a data implosion.

We provide the link to our post and the related response from the technical staff for the reviewer's

reference (https://forum.earthdata.nasa.gov/viewtopic.php?t=6315). We have corrected this issue in the revised version of the manuscript.

2. There is no justification for the threshold of $r_e$ < 14 μm. While larger $r_e$ allows more water loss by precipitation, it may be more than balanced by less water loss due to less evaporation of the larger cloud drops.

3. Furthermore, $r_e$ increases with cloud geometrical depth (CGT) and LWP increases with $CGT^2$. Therefore, excluding scenes by their $r_e$ values is incurring bias, rendering the whole study questionable

4. Line 131: The positive trend of LWP with $N_d$ was previously documented to occur only at $N_d$<30 $cm^{-3}$ (Figure 2 of Gyspeerdt et al., 2019). The clouds have to be very shallow with respectively small LWP for $r_e$<14 in clouds with $N_d$<30 $cm^{-3}$.

In fact, the condition of $r_e$<14 um imposes an artifact of more LWP with larger $N_d$, because with larger $N_d$ the cloud needs to grow deeper and have larger LWP for reaching $r_e$=14 μm at the cloud top !!!

1. Indeed, this study's maximum LWP is shifted from 30 (Figure 2 of Gyspeerdt et al., 2019) to nearly 100 $cm^{-3}$. This is evident in Fig1 left panels, especially in the convective regime (AUW), where cloud thickness and, hence, LWP consistently increase with $N_d$. This artifact dominates the results of this study.

**Response:** We agree with this great point, and we sincerely appreciate the reviewer's professional comments. Since these comments all relate to the threshold of $r_e$ < 14 μm, we will address this point below.

This issue was discussed earlier in the study. The reason for choosing the threshold of $r_e$ < 14 μm is that the invalidation of adiabatic assumptions for $N_d$ retrievals under precipitation conditions can introduce bias. For example, Grosvenor et al. (2018) suggests in their review paper on $N_d$ retrieval that "As a precautionary measure, it may be prudent to attempt to filter out situations with precipitation before performing $N_d$ retrievals". Kang et al. (2021), using aircraft observations, also pointed out that removing precipitation enhances the retrieval accuracy of $r_e$ in SatCORPS Himawari-8 product, which is an important variable affecting $N_d$ retrieval. Therefore, in order to obtain more accurate $N_d$ values and focus on the microphysical processes within non-precipitating clouds, we firstly used GPM IMERG hourly precipitation product to exclude precipitation scenes. However, considering GPM's limited ability to detect light drizzle, we additionally applied $r_e$ < 14 μm threshold based on suggestions according to

Rosenfeld et al. (2012), which has been widely used to distinguish between precipitating and non-precipitating clouds (Possner et al., 2020; Rosenfeld et al., 2019).

We acknowledge the reviewer's concern on using the threshold of $r_e$ < 14 μm. The sensitivity analysis on whether to exclude precipitation has been conducted. Figure R2 shows the $N_d$-LWP relationship and the diurnal variations LWP adjustments under different precipitation criteria. In both regions, the threshold application primarily removes cloud samples with small $N_d$ and large LWP, located in the upper left of the $N_d$-LWP space. Specifically, in AUW region, the diurnal pattern shows little change before noon, and without the threshold of $r_e$ <14 μm, the afternoon variation becomes smaller, indicating that the samples we excluded primarily affected the afternoon results. This may be because clouds with larger $r_e$ and larger LWP primarily occur in the afternoon. In the morning, $r_e$ is relatively small, so adding the threshold of $r_e$ < 14 μm does not significantly affect the dominant samples. However, in the afternoon, as $r_e$ increases, the inclusion of samples with smaller $N_d$ and larger LWP causes the LWP adjustments to become more negative. The results using only GPM criterion are similar to those without any precipitation restriction, indicating the diurnal pattern is dominated by non-precipitating samples. In the ECS region, adding the threshold of $r_e$ < 14 μm has little impact on the diurnal pattern of LWP adjustments, mainly affecting the values. This is likely because the ECS region is characterized by smaller $r_e$ values with heavily influence by anthropogenic aerosol pollution. Most precipitation samples are excluded by GPM.

[Figure]

**Figure R2. Comparison of the $N_d$-LWP relationship and diurnal variations of LWP adjustments under different precipitation screening criteria in two typical regions (the west of Australia, AUW) and the east China sea, ECS). Blue dots are all sample within the $N_d$-LWP space at 1400 LT. Black dots represent median LWP in each $N_d$ bin.**

Based on the results of the above sensitivity analysis, and considering the aim to minimize the significant uncertainties that heavy precipitation may introduce to the retrieval of $N_d$. We have decided not to use the $r_e < 14$ μm threshold following the reviewer's suggestion, but we still retain the exclusion of heavy precipitation scenes with the criteria of GPM = 0.

Combining feedbacks from both reviewers, the updated results are provided in the revised manuscript. Overall, the primary conclusions have been revised to (see Lines 510-538): "This study reveals the diurnal variations of LWP adjustments in two specific regions within the sight of Himawari-8, along with the possible mechanisms contributing to these variations. The studied regions have significant differences in environmental conditions and aerosol loadings. Although some conclusions are similar to the previous studies, we have also discovered some new phenomena. The observational studies demonstrate LWP adjustments in two regions are determined by the dominant microphysical-dynamical processes in different $N_d$ stages (entrainment feedbacks and warm invigoration), while their diurnal variations depend on the dynamical conditions of the boundary layer. Important findings from this investigation are as follows:

[revised manuscript text omitted]

Yost, C. R., Minnis, P., Sun-Mack, S., Chen, Y., and Smith, W. L.: CERES MODIS Cloud Product Retrievals for Edition 4—Part II: Comparisons to CloudSat and CALIPSO, IEEE Transactions on Geoscience and Remote Sensing, 59, 3695–3724, https://doi.org/10.1109/TGRS.2020.3015155, 2021.

Jiayi Li et al. (Author)

**We sincerely appreciate Reviewer #2's insightful suggestions, which prompt us to clarify several points and strengthen the arguments. Overall, we have clarified the explanation of the mechanisms with more supporting evidences. Many detailed discussions as suggested for the conclusion have been provided in the revised manuscript. Furthermore, we have improved the language for greater accuracy and readability.**

**In the following text, the reviewers' comments are listed in black, our response is in blue, and changes to the text are highlighted in red. A version of the revised manuscript with tracked changes is also provided.**

**Important revision includes:**

1. According to Reviewer #1's comments, only the threshold of GPM = 0 mm/hr was used for the exclusion of precipitation. The corresponding result section has been modified. The $N_d$-LWP relationship no longer exhibits the initial increasing trend in both regions. The quantification of the impact of diurnal variations on radiative effects has been revised.

2. We have clarified the explanation of the mechanisms based on direct conclusions. This allows us to distinguish between these directly supported conclusions and inferences that rely on previous research with supporting evidence.

**Major comments:**

1. The authors analyze data spanning three entire years, rather than focusing on specific seasons. However, this approach carries some risk because it is well-established that cloud properties and environmental conditions can vary significantly across different seasons. I recommend that the authors examine potential seasonal differences and assess how these variations might influence their results.

**Response:** We sincerely appreciate the reviewer for highlighting this important consideration regarding the potential seasonal differences. We have conducted further analyses on seasonal sensitivities as suggested (Figures R2-5 compared to Figure R1). Overall, in AUW region the diurnal pattern of LWP adjustments is not sensitive to seasonal changes, while the ECS region exhibits seasonal differences.

Among the total samples (173181), spring, summer, autumn, and winter account for 31%, 3%, 22%, and 44%, respectively. Due to the limited summer samples (3%), their results are statistical insignificance (p > 0.05), especially after eliminating the samples with precipitation by applying the threshold (GPM = 0 mm hr$^{-1}$). The LWP adjustments in other seasons exhibit similar diurnal patterns and magnitudes, peaking at noon (black lines in Figures R2F, R4-5F). This similarity may be due to the weak seasonal variations in the diurnal patterns of LWP and $N_d$ (Figure R6). The diurnal patterns of warm invigoration in spring and winter are similar to the overall results (blue lines in Figures R2F and R5F compared to Figure R1F). The minimum $N_d$ at 1100 LT coincides with the weakest warm invigoration (i.e., minimal LWP enhancement). Autumn exhibits the lowest $N_d$ among seasons (Figure R4F), corresponding to the weakest warm invigoration (~50%/~31% lower than spring/winter) and the largest diurnal fluctuations. The diurnal pattern of entrainment feedbacks in spring differs from other seasons, possibly due to its distinct entrainment rate diurnal variation, which can be illustrated by the variation of cloud-top height (CLTH). Here, based on the relationship between CLTH, $w_s$ (always negative) and entrainment rate ($w_e$) ($\frac{dCLTH}{dt} = w_s + w_e$) (Painemal et al., 2013), the diurnal variations of $w_e$ (entrainment rate) can be qualitatively analyzed with the diurnal variations of CLTH and large-scale subsidence ($w_s$) (Figure R7). Before 1400 LT, the variation of large-scale subsidence is unrelated to CLTH, thus the change in CLTH can only be attributed to entrainment rate. It weakens before 1200 LT, possibly due to the decreasing cloud-top longwave cooling after sunrise. It then increases until 1400 LT which may be caused by the enhanced longwave cooling. After 1400 LT, the decrease in CLTH is caused by the enhancement of large-scale subsidence.

We have added a relevant content of seasonal sensitivity in the revised manuscript (see Lines 420-442): "Given that the samples include four seasons over four years, we conduct a sensitivity analysis regarding seasonal influences as cloud properties and environmental conditions can vary significantly across different seasons. Overall, the diurnal pattern of LWP adjustments is not sensitive to seasonal changes in AUW region (black lines in Figures S9F-S12F compared to Figure 1). Since the AUW region is a persistent stratocumulus area, the diurnal variations of cloud thickness remain consistent across all seasons, with the thickest clouds in the morning and the thinnest in the early afternoon, followed by a slow increase. This implies that the decoupling process in the persistent Sc region is not affected by seasonality, resulting in the similar patterns of LWP adjustments. ECS region exhibits seasonal

differences (Figures S9-S12). Among the total samples (173181), spring, summer, autumn, and winter account for 31%, 3%, 22%, and 44%, respectively. Due to the limited summer samples (3%), their results are statistical insignificance (p > 0.05), especially after eliminating the samples with precipitation by applying the threshold (GPM = 0 mm hr$^{-1}$). The LWP adjustments in other seasons exhibit similar diurnal patterns and magnitudes, peaking at noon (black lines in Figures S9F, S11-S12F). This similarity may be due to the weak seasonal variations in the diurnal patterns of LWP and $N_d$ (not shown). The diurnal patterns of warm invigoration in spring and winter are similar to the overall results (blue lines in Figures S9F and S12F compared to Figure 1F). The minimum $N_d$ at 1100 LT coincides with the weakest warm invigoration (i.e., minimal LWP enhancement). Autumn exhibits the lowest $N_d$ among seasons (Figure S11F), corresponding to the weakest warm invigoration (~50%/31% lower than spring/winter) and the largest diurnal fluctuations. The diurnal pattern of entrainment feedbacks in spring differs from other seasons, possibly due to its distinct entrainment rate diurnal variation, which can be illustrated by the variation of cloud-top height (CLTH). Here, based on the relationship between CLTH, $w_s$ and $w_e$ ($\frac{dCLTH}{dt} = w_s + w_e$) (Painemal et al., 2013), the diurnal variations of $w_e$ (entrainment rate) can be qualitatively analyzed with the diurnal variations of CLTH and large-scale subsidence ($w_s$) (Figure S13). Before 1400 LT, the variation of large-scale subsidence is unrelated to CLTH, thus the change in CLTH can only be attributed to entrainment rate. It weakens before 1200 LT, possibly due to the decreasing cloud-top longwave cooling after sunrise. It then increases until 1400 LT which may be caused by the enhanced longwave cooling. After 1400 LT, the decrease in CLTH is caused by the enhancement of large-scale subsidence.".

[Figure]

**Figure R1. LWP adjustments in log-log spaces and their diurnal patterns in two typical regions (the west of Australia, AUW and the east China sea, ECS).** Non-precipitation cloud samples are scattered in $N_d$-LWP log space at 1400 LT in (A) AUW and (D) ECS region. Colored dots are samples in different cloud thickness (H) bins (unit: m). Black dots represent the median LWP in each $N_d$ bin. The colored lines are the fits of black dots at different stages in ECS region. Diurnal variations of LWP adjustments binned by H in (B) AUW and (E) ECS regions are shown. Colored lines in (F) are diurnal variations of different stages in (D), while black lines in (C) and (F) are the overall diurnal variations of LWP adjustments in two regions, respectively. Blue line in (C) represents the diurnal variation of H. Dashed lines represent the average LWP adjustments considering diurnal variations, –0.31 for AUW (C) and 0.02 for ECS (F). Same as Figure 1 in the revised manuscript.

[Figure]

**Figure R2. LWP adjustments in log-log spaces and their diurnal patterns in two typical regions (the west of Australia, AUW and the east China sea, ECS) for spring.** Same as Figure S9 in the revised Supplementary Materials.

[Figure]

**Figure R3. Same as Figure R2 but for summer. The total sample size is 187646 for the AUW region and 5062 for the ECS region. Note that insufficient sample size in the ECS region made LWP adjustment in different H bins impractical, resulting in an empty Figure E.** Same as Figure S10 in the revised Supplementary Materials.

[Figure]

**Figure R4. Same as Figure R2 but for autumn. The total sample size was 94311 for the AUW region and 37387 for the ECS region.** Same as Figure S11 in the revised Supplementary Materials.

[Figure]

**Figure R5. Same as Figure R2 but for winter. The total sample size was 62664 for the AUW region and 76623 for the ECS region.** Same as Figure S12 in the revised Supplementary Materials.

[Figure]

**Figure R6. Diurnal variations of $N_d$ and LWP for four seasons in ECS region.**

[Figure]

**Figure R7. Diurnal variations of cloud-top height (CLTH) and vertical velocity on 700 hPa**

**(omega700, positive values indicate downdraft) for spring in ECS region.** Same as Figure S13 in the revised Supplementary Materials.

2. I am aware of another recent study that examines the diurnal cycle of LWP adjustments over the Eastern North Atlantic (Qiu et al., 2024). The methodology and conceptual framework of that paper are quite similar to the current study, though applied to a different region. Although the authors have cited this work, they have not provided a thorough discussion of it. Given these similarities, I suggest the authors include a discussion comparing their conclusions with those of Qiu et al., 2024, highlighting any key differences or regional contrasts.

**Response:** Thank you very much for the valuable advice! Our observed diurnal LWP adjustment pattern in the AUW region is consistent with Qiu et al. (2024)'s findings in the eastern North Atlantic, where thick-thin cloud transitions dominated daytime variability. However, while they linked this to regional cloud internal evolution, we identify boundary-layer dynamics (e.g., decoupling and entrainment) as the primary driver. Additionally, the method we used is quite different compared with Qiu et al. (2024). They calculated LWP adjustment within each 1° grid box, assuming constant meteorological conditions, whereas we calculated LWP adjustments with all samples over four years, preserving the influence of meteorological covariations at each moment. By stratifying analyses by cloud thickness according to Rosenfeld et al. (2019), we disentangle meteorological covariations from cloud internal feedbacks. Additionally, cloud thickness remains relatively stable after 1300 LT in our results. The weakening of negative LWP adjustments is primarily due to the weakening of entrainment induced by the strengthening of large-scale subsidence. Therefore, our results tend to emphasize the diurnal variations of LWP adjustments induced by time-dependent meteorological covariations primarily stemming from boundary layer dynamical mechanisms, while Qiu et al. (2024) focused more on the evolution of clouds, suggesting that clouds retain the memory from previous state.

Furthermore, we conduct the same analyses in ECS region with completely different environmental background, and obtain entirely different results. In humid and unstable environments, aerosol-induced warm invigoration is more likely to occur. In this condition, cloud thickness is no longer suitable for distinguishing meteorological conditions as a mediator of $N_d$-LWP relationship. The cloud thinning mechanism is also insufficient to explain the diurnal variations of LWP adjustments. This demonstrates that the significant regional differences in the diurnal variations of LWP adjustments, depending on

aerosol loadings, cloud regimes and meteorological conditions.

In the revised manuscript, we have included discussion comparing our results with Qiu et al. (2024). The details have been added on Lines 475-482: "Our observed diurnal LWP adjustment pattern in the AUW region is consistent with Qiu et al. (2024)'s findings in the eastern North Atlantic, where thick-thin cloud transitions dominated daytime variability. However, unlike Qiu et al. (2024)'s method, which focused on regional cloud internal evolution and calculated LWP adjustment within each 1° grid box without considering meteorological covariations, this investigation preserves the influence of meteorological covariations at each moment. By stratifying analyses by cloud thickness according to Rosenfeld et al. (2019), we disentangle meteorological covariations from cloud internal feedbacks. Additionally, cloud thickness remains relatively stable after 1300 LT in our results. The weakening of negative LWP adjustments is primarily due to the weakening of entrainment induced by the strengthening of large-scale subsidence.".

3. Many of the statements in the manuscript appear to be interpretations or inferences drawn from previous studies, rather than conclusions directly supported by the current analysis. However, the authors present these statements as if they are firmly established by their own results. I recommend clarifying which findings are directly derived from the current analysis and distinguishing them from interpretations based on prior work.

**Response:** Thanks for your suggestion! We apologize for some statements that may mislead the reviewer. We acknowledge that some conclusions could only be inferred based on earlier studies due to the observational limitations, despite our best efforts to use proxies to represent physical processes (such as using LWP skewness to characterize the degree of decoupling). In the revised version, we have further separated the discussion and conclusion sections. Within the conclusion section, the key findings have been further condensed. Please refer to Major comments 5. Throughout the manuscript, any statement about mechanisms derived from previous studies has been carefully rephrased to distinguish the direct conclusions and inferences. For example, some of the revisions on the important parts related to the conclusions are as follows:

[revised manuscript text omitted]

4.  There are grammatical errors throughout the paper, which occasionally hinder clarity and understanding. I have noted some examples in the minor comments section, but I recommend that

the authors thoroughly proofread the manuscript to address these issues.

**Response:** We sincerely appreciate the reviewer's feedback about the language in the manuscript. We have carefully revised the manuscript to address these issues and ensure that the language is clear and accurate.

5.  While the authors provide a very detailed discussion in the results section, the conclusions at the end of the paper are quite general and lack specificity. Some of these conclusions are too broad to be useful. Instead of simply stating that LWP adjustments depend on microphysical-dynamical processes and meteorological conditions, I recommend that the authors summarize how each condition specifically influences LWP adjustments and what we can learn from them. Presenting the key findings in bullet points would greatly improve clarity and provide a concise summary of the main results.

**Response:** Thanks for your insightful comments! We agree that the conclusion section is too general. We have separated the discussion and conclusion into two sections, rephrasing a more detailed summary as suggested (see Lines 510-538): "This study reveals the diurnal variations of LWP adjustments in two specific regions within the sight of Himawari-8, along with the possible mechanisms contributing to these variations. The studied regions have significant differences in environmental conditions and aerosol loadings. Although some conclusions are similar to the previous studies, we have also discovered some new phenomena. The observational studies demonstrate LWP adjustments in two regions are determined by the dominant microphysical-dynamical processes in different $N_d$ stages (entrainment feedbacks and warm invigoration), while their diurnal variations depend on the dynamical conditions of the boundary layer. Important findings from this investigation are as follows:

(1) In AUW region, the diurnal variations of LWP adjustments are insensitive to seasonality. The overall negative LWP adjustments decrease from −0.27 to −0.41 before 1300 LT and then increase to −0.34. Cloud thickness in AUW region can serve as a confounder to separate the effects of meteorological covariations. The diurnal pattern is primarily associated with cloud thinning induced by decoupling process of MBL quantified by LWP skewness before 1300 LT and the weakening of entrainment induced by intensification of large-scale subsidence after 1300 LT.

(2) In ECS region, diurnal variations of LWP adjustment exhibit seasonal differences. Samples from winter and spring dominate the overall variations (accounting for 75% of the total

samples). For the overall results, LWP increases and then decreases with $N_d$, suggesting possible competition between entrainment feedbacks and warm invigoration. The diurnal pattern of LWP adjustments is determined by the combined diurnal variations of these two mechanisms. Warm invigoration is related to the diurnal variation of the $N_d$ at the turning points of the two processes. Lower $N_d$ in the ECS region implies a weaker warm invigoration.

(3) We indicate an overall underestimation of the cooling effect by LWP adjustment up to 89% (14%), with a further 20% (15%) offset of the Twomey effect when neglecting the diurnal variations of LWP adjustments in AUW (ECS) region. Furthermore, our results quantify the regional impact of boundary layer dynamic conditions on LWP adjustments. For example, diurnal decoupling process in AUW region results in a 219% variation of LWP adjustments within the daytime relative to the daily mean (the diurnal variation range divided by the daily mean), assuming other conditions remain relatively unchanged.

Our research provides a detailed discussion for the diurnal variations of LWP adjustments and how they are influenced by existed boundary layer mechanisms. We underscore the importance of fully considering the covariation with environmental conditions, indicating different potential influencing factors on cloud brightening and radiative forcing in terms of the regional and diurnal daytime scale. It is a highly time-dependent variable lacking quantification and should be taken into consideration of future research in aerosol indirect effects on climate.".

**Minor comments:**

1. Line 62: Please check the grammar

**Response:** Thanks for your careful checks.. We have revised the sentence as (see Lines 60-62): "Our analysis focuses on $1° \times 1°$ non-precipitation marine low-level cloud samples aggregated from filtered pixel-level satellite data. We aim to avoid the impact of precipitation on retrieval of $N_d$ and focus only on the development of clouds in response to aerosol loading associated with microphysical-dynamical conditions over two selected regions.".

2. Line 73: Why are the retrievals sub-sampled to different resolutions in the Northern and Southern Hemispheres?

**Response:** Thank you for raising this question! In fact, after consulting the technical staff, we found that the incorrect information on the website had led us to provide inaccurate information in the manuscript

and mislead the reader. We have helped them correct the errors on the website. The observation resolution of CERES_GOES_HIM08 is 2 km at nadir, and has been sub-sampled to 6 km for both the Northern and Southern Hemispheres. The sub-sampled resolution meets the needs of the CERES project without having a data implosion. We have corrected this issue in the revised manuscript on Lines 72-73: "The retrievals are at 2-km resolution (at nadir) and are sub-sampled to 6 km. The sub-sampled resolution meets the needs of the CERES project without having a data implosion.".

3.  Line 75: Could you briefly explain how CLTH, CLBH, and H are retrieved?

**Response:** Thanks for your comment! CERES_GOES_HIM08 product provides cloud-top height (CLTH), cloud base height (CLBH), and cloud thickness (H) information. They are retrieved based on the CERES Edition 4 (Ed4) cloud property retrieval algorithm system. We have briefly introduced the algorithm of the three parameters on Lines 77-81: "Briefly, CLTH is estimated as the altitude where the cloud-top temperature (CLTT) occurs in the temperature profile. The temperature profile is provided by CERES Meteorology, Ozone, and Aerosol (CERES MOA) dataset. CLTT is derived from an empirical parameterization of cloud-top emissivity at channel 4 and cloud effective temperature. H is computed using empirical formulas with $\tau$: $H = 0.39 \ln \tau - 0.01$ for liquid clouds. CLBH is directly obtained by subtracting H from CLTH." For specific information, please refer to Minnis et al. (2011, 2021).

4.  Lines 80-86: You assume fad = 0.8, which indicates a sub-adiabatic condition. Therefore, the retrieval of Nd is not strictly under the adiabatic assumption (which requires fad = 1). However, LWP appears to be derived assuming adiabatic conditions. This introduces some inconsistency between your Nd and LWP retrievals. Please clarify and address this inconsistency.

**Response:** Thanks for raising a great point! $LWP = \frac{5}{9} \rho_w \tau r_e$ only highlights its availability for profiles of linearly increasing LWC but does not require the adiabatic assumption (i.e., sub-adiabatic also follow linearly increasing LWC in cloud profile) (Wood and Hartmann, 2006). The relationship is modified by a factor of 0.83 from $\frac{2}{3} \rho_w \tau r_e$ which assumes a vertical homogenous cloud (vertically constant LWC). According to Bennartz (2007), the retrieval method with a factor of 5/9 shows better agreement with microwave observations. Additionally, Lu et al. (2023) have used this method as the actual LWP to estimate the cloud adiabatic fraction with the adiabatic $LWP = \frac{1}{2} c_w H^2$. Therefore, the retrievals of both $N_d$ and LWP are not performed under strictly adiabatic conditions. The combination of these two retrieval methods of $N_d$ and LWP has been widely used in the satellite investigation of LWP adjustment (Fons et

al., 2023; Gryspeerdt et al., 2019; Qiu et al., 2023; Smalley et al., 2024).

We apologize for the misleading description, which has been corrected on Lines 83-90: "$N_d$ can be estimated as (Bennartz, 2007).:

$$N_d = \frac{\sqrt{5}}{2\pi k} \left( \frac{f_{ad} c_\omega \tau}{Q \rho_w r_e{}^5} \right)^{\frac{1}{2}} \tag{1}$$

where $\tau$ represents cloud optical depth and $\rho_w$ is liquid water density. The extinction efficiency $Q \approx 2$, as $Q$ relies less on the size parameter in near-infrared. $k$, related to droplet size distribution, is set as 0.8 for maritime cloud (Martin et al., 1994; Painemal and Zuidema, 2011). $c_w$ represents the condensation rate determined by temperature in cloud (here is cloud-top temperature from SatCORPS). A constant adiabatic value ($f_{ad}$) of 0.8 is used to represent the deviation from the adiabatic profile (Bennartz, 2007).",

and Lines 94-97: "In this study, the LWP from SatCORPS is calculated as $\frac{5}{9} \rho_w \tau r_e$ in sub-adiabatic conditions, following the method by Wood and Hartmann (2006). The combination of these two retrieval methods of $N_d$ and LWP has been widely used in the satellite investigations of LWP adjustment (Fons et al., 2023; Gryspeerdt et al., 2019; Qiu et al., 2023; Smalley et al., 2024).".

5.  Line 93: Why did you choose 268K as the threshold for cloud-top temperature, rather than the more commonly used 273K?

**Response:** Thank you for your question! The threshold of 273 K is stricter, while 268 K allows for the presence of some supercooled phase, where it is ubiquitous over the Southern Ocean (Hu et al., 2010). Actually, the threshold of 268 K also has been used in previous studies for the threshold of liquid water cloud-top temperature (Bennartz and Rausch (2017), Gryspeerdt et al. (2022) and Li et al. (2018)). Our study follows the above studies, and we focus on liquid water clouds, which have typical cloud top temperatures (CLTT) between 268 and 300 K (Bennartz and Rausch, 2017).

[Figure]

**Figure R10.** Histogram of cloud-top temperature (CLTT) in AUW (left) and ECS (right) regions.

In addition, we have checked whether this threshold is appropriate. The statistical distributions of CLTT for the samples in this study are presented in Figure R10. Overall, 96% (97%) of the samples are larger than 273 K in AUW (ECS). Therefore, the threshold has a negligible impact on the overall results.

The relevant description has been rephrased to be clearer on Lines 100-103: "To maintain consistency with previous studies (Bennartz and Rausch, 2017; Li et al., 2018), we adopted 268 K as the threshold of CLTT for liquid clouds, rather than 273 K. In fact, 96% (97%) of the samples exhibited CLTT above 273 K in AUW (ECS) region. Therefore, the threshold has a negligible impact on the overall results.".

6. Line 96: Please check the grammar in this sentence. It should be "Each grid containing at least 30 pixels is considered as a cloud sample." Additionally, how many pixels are there in total in each 1°x1° scene?

**Response:** Thanks for your careful reading. We have corrected the sentence in the revised manuscript. The histograms of samples in 1°x1° scene are showed in Figure R11 for reviewer's reference. In total, we collect 480189 1°×1° scenes in AUW and 173181 1°×1° scenes in ECS. Each scene contains 83 (87) pixels on average in AUW (ECS). We have added the information on Lines 104-105: "Each grid containing at least 30 pixels and is considered as a cloud sample. On average, each grid contains 83 (87) pixels in AUW (ECS) region.".

[Figure]

**Figure R11.** Histograms of pixels in each 1°x1° scene in AUW (left) and ECS (right) regions.

7. Line 137: Using AOD as a proxy for aerosols introduces uncertainties, as AOD represents column-integrated aerosol loading without indicating aerosol vertical distribution. If aerosols are located above the boundary layer, they may not interact with clouds. However, Figure 8 shows that Nd and AOD are well-correlated, which helps justify using AOD as a proxy. I suggest moving Figure 8 earlier in the manuscript to support the use of AOD.

**Response:** Thanks for your suggestion. We agree that column AOD is not an adequate proxy. To remain

consistent with earlier studies, we selected AOD as a proxy for aerosols. We have moved the Figure 8 earlier in Figure 5 as suggested. The relevant content has been rephrased to justify the availability of AOD as a proxy (see Lines 253-258): "Note that we select the column AOD as an aerosol proxy to remain consistent with the above studies. Although AOD may not represent aerosol concentrations in some conditions, Figure 5 shows significant correlations observed between the 4-year long-term variations of AOD and $N_d$ at 1200 LT in both regions, particularly in ECS with a correlation of 0.81. Meanwhile, both regions show the similar distribution patterns, with higher $N_d$ and smaller $r_e$ near the continental coastal area, aligning with the average AOD spatial distribution (spatial correlation coefficients of 0.84 in AUW and 0.91 in ECS) (Figure S1), suggesting the availability of AOD as an aerosol proxy.".

[Figure]

**Figure R12. 4-year long-term variations of $N_d$ and total aerosol optical depth (AOD) from MERRA-2 at 1200 LT in AUW (A) and ECS (B) region.** The correlation coefficients (corr) between $N_d$ and AOD are 0.35 and 0.83 (significant at the 95% confidence level), respectively. Same as Figure 5 in the revised manuscript.

[Figure]

**Figure R13. Distributions of cloud properties in two typical regions (the east China sea (20º-30ºN, 120º-130ºE, ECS) and the west of Australia (25º-35ºS, 95º-105ºE, AUW).** (A) Geographical distribution of the view zenith angle of Satellite Cloud and Radiation Property retrieval System (SatCORPS) Himawari-8 data. The selected regions are marked by red boxes. Spatial distributions of cloud droplet number concentration ($N_d$) (B, E), effective radius ($r_e$) (C, F) and total column aerosol optical depth (AOD) (D, G) from MERRA-2 data are presented. The numbers in the lower right corner represent regional averages being weighted by the cosine of latitude. Same as Figure S1 in the revised Supplementary Materials.

8.  Line 195. The sentence "These moist and unstable conditions lead..." could benefit from further elaboration. Please clarify how these conditions influence cloud properties.

**Response:** Thank you for pointing this out. LTS reflects the stability of the boundary layer, influencing the convective property of clouds. Relative humidity (RH) reflects the humidity of free atmosphere, impacting the mixing process at cloud margin and aerosol activations. We have revised the results section and removed the sentence in response to Reviewer #1's comments. However, the descriptions regarding moist and unstable environments remain. And we have provided a detailed account in the revised manuscript according to Reviewer #2's suggestions (Lines 210-217): "Although the microphysical-dynamical processes are challenging to observe directly, environmental conditions can be considered as proxies and provide further support for the invigoration effect. The cloud deepening in ECS region is mainly attributed to increasing CLTH (Figure 2D). Unstable boundary layers (low LTS) favor the formation of more convective clouds (Manshausen et al., 2022), while high RH provides moisture for cloud vertical development. The unstable and moist atmosphere in ECS provides such conditions with a

mean lower-tropospheric stability (LTS) of 15.94 K and a peak in relative humidity on 700 hPa (RH700) of 70% (Figure 3). Gryspeerdt et al. (2019) also reported this rising behavior at high $N_d$, especially in moist conditions, consistent with our results noted here. Christensen and Stephens (2011) found elevated cloud-top height from open cell clouds in response to ship pollution in relatively unstable and moist conditions.".

[Figure]

**Figure R14. Comparisons between $N_d$-LWP relationship and $N_d$-Thickness relationship in two regions.** Relationship between $N_d$ and (A) LWP, (B) cloud thickness in AUW region. Relationship between $N_d$ and (C) LWP, (D) cloud thickness in ECS region. The orange solid and dashed lines show the change of cloud top height (CLTH) and cloud base height (CLBH) with $N_d$. Same as Figure 2 in the revised manuscript.

[Figure]

**Figure R15. 4-year meteorological conditions of non-precipitation clouds in AUW and ECS regions from 2016 to 2019.** Histograms of meteorological factors are presented here. The mean values are labeled in the top-left corner. Data are directly or indirectly derived from ERA5. For vertical velocities on 800 hPa (omega800), positive (negative) values indicate downdraft (updraft). Same as Figure 3 in the revised manuscript.

9.  Line 210: The statement "The increasing of LWP at high Nd..." seems incorrect. It is well-established that increased surface area of cloud droplets can enhance entrainment and evaporation, potentially reducing LWP. Please revise or clarify this point.

**Response:** Thank you for raising this question! Increased surface area of cloud droplets induced by the increased aerosol loading leads to two opposing effects according to previous studies (Altaratz et al., 2014). On the one hand, they indicated that more droplets delay the collision-coalescence and provide more surface area for condensation, releasing latent heat and promoting cloud vertical development, thus increasing LWP. On the other hand, more small droplets can be more likely to evaporate due to enhanced entrainment, leading to a decreased LWP, as the reviewer mentioned. These two effects determine the final sensitivity of cloud properties (such as cloud depth, LWP, cloud top height) to aerosols in a competitive way, depending on environmental conditions and cloud characteristics (Dagan et al., 2015).

Here in ECS region, our results show that the increasing behavior of LWP at high $N_d$ is consistent with deepening of cloud depth due to increased cloud height. Therefore, we can reasonably infer that this phenomenon represents warm cloud invigoration given the favorable environment conditions in ECS

region. Briefly, high humidity and instability in ECS favor the accumulation of cloud water through condensation. Also, previous studies have reported the occurrence of warm invigoration associated with higher aerosol loading and high proportion of convective clouds(Kaufman et al., 2005; Yuan et al., 2011; Zhang et al., 2021).

We have rephrased the relevant content to be clearer (Lines 189-248): "However, LWP begins to rise at high $N_d$ in ECS (blue line in Figure 1D), which is the primary reason causing the overall positive LWP adjustments in this region. Positive sensitivity over ECS has been reported but not fully understood (Bender et al., 2019; Gryspeerdt et al., 2019; Zhang et al., 2021). Michibata et al. (2016) attributed the positive LWP response in non-precipitation clouds over East Asia to the cloud lifetime effect(Albrecht, 1989). Here in ECS region, clouds are heavily affected by anthropogenic aerosols, showing LWP increases with $N_d$ at high $N_d$ (>300 cm$^{-3}$). This behavior is related to the deepening of cloud depth with aerosols (Figure 2, C and D), indicating warm invigoration by aerosols (Koren et al., 2014).

The above opposite responses of LWP (either enhanced or decreased) to increasing aerosol loading depends on the environmental conditions and cloud characteristics (Altaratz et al., 2014). On the one hand, they indicated that more droplets delay the collision-coalescence and provide more surface area for condensation, releasing latent heat and promoting cloud vertical development, thus increasing LWP (warm invigoration). On the other hand, more small droplets can be more likely to evaporate due to enhanced entrainment, leading to a decreased LWP (entrainment feedbacks). According to Dagan et al. (2015), the competition between these two processes determines the response of cloud macrophysical properties to aerosols. The $N_d$-LWP relationship in ECS indicates that warm invigoration takes over after around 300 cm$^{-3}$ leading to cloud deepening. Here, we will demonstrate that ECS region is favorable for warm invigoration to occur from three aspects: environmental conditions, cloud regimes and aerosols.

Although the microphysical-dynamical processes are challenging to observe directly, environmental conditions can be considered as proxies and provide further support for the invigoration effect. The cloud deepening in ECS region is mainly attributed to increasing CLTH (Figure 2D). Unstable boundary layers favor the formation of more convective clouds(Manshausen et al., 2022), while high RH provides moisture for cloud vertical development. The unstable and moist atmosphere in ECS provides such conditions with a mean lower-tropospheric stability (LTS) of 15.94 K and a peak in relative humidity on 700 hPa (RH700) of 70% (Figure 3).Gryspeerdt et al. (2019) also reported this rising behavior at high $N_d$, especially in moist conditions, consistent with our results noted here. Christensen and Stephens (2011)

found elevated cloud-top height from open cell clouds in response to ship pollution in relatively unstable and moist conditions.

Secondly, the more prevalent convective clouds in the ECS region would be another favorable condition for warm invigoration. Zhang et al. (2021) also attributed the positive LWP adjustments to warm invigoration with the widespread low-level convective clouds (Sc and Cu) in ECS. According to the division from Rosenfeld et al. (2019), we categorize the clouds into three regimes, i.e., Sc (LTS > 18 K), Sc to Cu transition (14 K ≤ LTS ≤ 18 K), and Cu (LTS < 14 K). (Figure 4, G, H and I). We show that clouds in ECS region are dominated by the Sc to Cu transition regime. The formation of this transition regime is associated with increasing sea surface temperature (SST) due to "deepening-warming decoupling" (Albrecht et al., 1995; Bretherton and Wyant, 1997). Sc presents over the relatively shallow and stable boundary layer with cooler sea surface along the coast (Figure 4, A and B) and most of Sc may be advected from the southeast Chinese plain (Klein and Hartmann, 1993). According to the cloud advection scheme by Miller et al. (2018), cloud advection can be approximated as a translation of the cloud field with the wind field. The advection height is assumed to correspond to the height of the cloud top. Therefore, we can simply deduce from the wind field on 700 hPa (Figure 4A) that clouds in ECS have the possibility of advection from the Chinese plain in the west. As air moves offshore, MBL deepens and cloud layer decouples with the surface mixed layer over the warmer sea surface. Cu forms in the moist and unstable subcloud layer and rises to the upper cloud layer, resulting in a local cumulus-coupled MBL.

Finally, at high aerosol-loading conditions, warm invigoration has been found in numerous studies. For instance, Kaufman et al. (2005) reported larger LWP in higher aerosol loading conditions over Atlantic warm clouds (a mix of stratus and trade cumulus) using MODIS observations. Yuan et al. (2011) found increased cloud amount and higher cloud top heights associated with volcanic aerosols in trade cumulus near Hawaii with A-Train satellites. In contrast to the model results of Koren et al. (2014), who suggested that warm invigoration saturates at higher aerosol loading (AOD ~ 0.3), our findings indicate a higher AOD of 0.41 (Figure 2), which is reasonable because the saturation value of AOD exhibits regional variability. For example, Kaufman et al. (2005) reported a maximum AOD of 0.46, while Zhang et al. (2021) found that the AOD in the ECS region is approximately 0.4. To summarize, these evidences all confirm the plausibility of warm invigoration in the ECS region, causing the positive LWP adjustments at high $N_d$.".

10. Line 214: Please define the "invigoration effect" clearly when first mentioned.

**Response:** Thanks! We have added a detailed explanation of the invigoration effect in the revised manuscript on Lines 195-199:"The above opposite responses of LWP (either enhanced or decreased) to increasing aerosol loading depends on the environmental conditions and cloud characteristics (Altaratz et al., 2014). On the one hand, they indicated that more droplets delay the collision-coalescence and provide more surface area for condensation, releasing latent heat and promoting cloud vertical development, thus increasing LWP (warm invigoration). On the other hand, more small droplets can be more likely to evaporate due to enhanced entrainment, leading to a decreased LWP (entrainment feedbacks).".

11. Line 244: Replace makes with make.

**Response:** Thank you for your careful review. The mistake has been corrected in the revised version.

12. Line 251: The statement "we find that LWP adjustments become negative" is incomplete. Please specify whether this refers to high or low cloud thickness (H).

**Response:** Sorry for the misleading. This sentence means that when we calculated LWP adjustments using mixing samples with different H, LWP adjustments present both positive and negative conditions. However, when we calculated in specific H bins, it becomes all negative. We have rephrased it in the revised version to be clearer on Lines 278-279: "However, we find that LWP adjustments become negative after constraining H in the intervals of Figure 1 (B and E), indicating the dominant effect of entrainment processes.".

13. Fig. 4. I recommend adding the wind field to one of the panels in Figure 4 instead of showing it separately in Figure S6.

**Response:** Thanks for your suggestion! As shown in Figure R16, we have added the wind field in Figure 4 in the revised manuscript.

[Figure]

**Figure R16**. Modified Figure 4 for the revised manuscript.

14. Line 262. "This indicates that clouds of different H respond differently to entrainment". This is a known fact, but the authors write it in a way that makes it seem like a new finding.

**Response:** We sincerely appreciate the reviewer's suggestion. This statement is indeed misleading. We have deleted the statement in the revised manuscript to ensure clarity and accuracy.

15. Lines 272-277. This summary is too general. Instead, please explicitly summarize how LWP adjustments are influenced by specific environmental factors such as LTS, cloud thickness (H), etc.

**Response:** We are grateful for your suggestion. We have added more investigations of impact of the environmental factors on LWP adjustment in the revised manuscript on Lines 265-273. And we have rephrased the summary section to be more specific on Lines 293-298.

Lines 265-273: "We further analyze the influence of meteorological conditions (i.e., LTS and RH) on LWP adjustments in the two regions (Figure S7). Overall, LWP adjustments cannot be explained by a single meteorological factor. For example, in ECS region, despite the similarity in diurnal patterns of LWP adjustment within different LTS bins, the magnitudes exhibit significant differences due to different aerosol loadings. Samples with LTS > 18 K are concentrated in coastal areas with higher aerosol loadings. Warm invigoration is stronger for these samples, thus the overall LWP adjustment is positive. In contrast, samples with LTS < 18 K have a larger proportion of smaller aerosol loadings. The effect of

entrainment feedback is more pronounced. This further highlights the importance of aerosol loadings in regulating LWP adjustments in ECS region. Meanwhile, the intricate interplay among meteorological factors, clouds, and aerosols makes it difficult to exclude the influences from meteorological factors (Chen et al., 2014; Engström and Ekman, 2010; Zhang and Feingold, 2023).".

Lines 293-298: "In summary, the above results indicate that LWP adjustments depend on entrainment feedbacks with increasing $N_d$ in AUW region. While in ECS region, LWP adjustments are results of the competition between entrainment feedbacks and warm invigoration. Given that LWP adjustment is influenced by a complex interaction of meteorological factors, we think that cloud conditions provide more reliable indications. Specifically, cloud thickness is important in AUW region, whereas aerosol loading (represented by $N_d$) is a better indicator in the ECS region. Therefore, the diurnal variations of these factors can provide important indications for us to investigate the potential mechanisms driving diurnal variations of LWP adjustments.".

[Figure]

**Figure R17. Diurnal patterns of LWP adjustments within different bins of meteorological factors (lower-tropospheric stability (LTS), relative humidity on 700 hPa and 1000 hPa (RH700 and RH1000)).** The top row shows results for AUW region, and the bottom row shows results for ECS region. Due to sample size limitations, LWP adjustment at each point is the regression slope of $N_d$ and LWP in log-log space for all samples at that condition. Same as Figure S7 in the revised Supplementary Materials.

16. Lines 314-315. The phrase "illustrating the decoupling process …" needs further explanation. Please expand.

**Response:** Thanks for your comments. We use LWP skewness as the quantification of decoupling degree. As the formation of Cu beneath Sc causes positive skewness of probability density function (PDF) of LWP of Sc decks during cold advection (Zheng et al., 2018).We have added further explanation for the LWP skewness in the revised manuscript (Lines 333-334): "Positive skewness indicates more data tends to be distributed to the right, and vice versa. Larger LWP skewness indicates a larger decoupling degree.".

17. Line 330. You mention ws (large-scale subsidence), but I don't believe this quantity is directly observable from your dataset. Please clarify.

**Response:** Thanks for your comment! Here we try to speculate the change of entrainment rate according to the change of CLTH based on Painemal et al. (2013). After 1200 LT, CLTH decreases, which means $\frac{dCLTH}{dt}$ becomes negative. According to the formula $\frac{dCLTH}{dt} = w_s + w_e$, negative $\frac{dCLTH}{dt}$ is subject to decreasing $w_s$ and/or decreasing $w_e$ (entrainment rate). $w_s$ is always negative in Sc region and decreasing $w_s$ refers to enhancing subsidence which will in turn decrease $w_e$. Therefore, negative $\frac{dCLTH}{dt}$ can always lead to decrease of $w_e$.

To ensure the completeness of the inference, we have supplemented the diurnal variation of large-scale vertical velocity on 700 hPa based on ERA5 reanalysis in Figure R8 and Figure 6 in the revised manuscript. After 1200 LT, enhancement of large-scale subsidence is presented (positive values indicate downdraft, unit Pa/s). We have rephrased the relevant description to further confirm the credibility of our result (Lines 353-358): "Furthermore, according to the relationship between CLTH, $w_s$ (always negative) and entrainment rate ($w_e$) ($\frac{dCLTH}{dt} = w_s + w_e$) in the mixed-layer model framework (Painemal et al., 2013), we explain the variations after 1200 LT. CLTH begins to decrease after 1200 LT, suggesting an intensification of large-scale subsidence ($w_s$, always negative in Sc region) and/or a weakening of entrainment rate ($w_e$). Large-scale subsidence on 700 hPa from ERA5 reanalysis becomes stronger (gray line in Figure 6A). It may enhance the temperature-inversion jump, which will in turn decrease the entrainment rate (Painemal et al., 2013).".

18. Line 336. You mention "the diurnal pattern of LWP." Could you indicate clearly where this is shown (e.g., in Figure 1)?

**Response:** Thanks for your comment! In Figure 1, we primarily discuss the diurnal variations of the LWP adjustments. Please refer to Figure 6 for the diurnal variation of LWP.

19. Line 341. The sentence "cloud-top entrainment weakens, as discussed earlier" is difficult to follow.

I recommend reorganizing this section so that the evidence and conclusion are more closely connected and easy to trace.

**Response:** Thanks for your valuable feedback! We apologize for any confusion it may have caused. According to Reviewer #1's comment, we have revised the section about the mechanisms governing the diurnal variation of LWP adjustments in AUW region. We also ensured that the description of our conclusions is easy to follow. This section has been accordingly rephrased as follows (Lines 362-370): "Based on the diurnal mechanisms of MBL discussed above, the diurnal pattern of LWP adjustments is primarily a consequence of the influence of these diurnal-related mechanisms on the relationship between $N_d$ and LWP across different microphysical-dynamical conditions. In AUW, the diurnal variations of the overall LWP adjustments (black line in Figure 1C) and cloud thickness (blue line in Figure 1C) demonstrate a strong consistency with a turning point at 1300 LT. The variation of LWP adjustment here is mainly attributed to the gradual thinning of clouds, which reflects the differential LWP responses to $N_d$ with varying H. LWP adjustment becomes more negative with the thinning of cloud, which is consistent with the results in Figure 1B. After 1300 LT, cloud thickness remains almost unchanged. The variation in LWP adjustments is mainly governed the weakening of entrainment due to the intensification of large-scale subsidence (Figure 6A). During this time, the weakening of the entrainment process leads to a weakening of the negative LWP adjustments over AUW region.".

20. Line 367. You state, "the strongest stability of the transition layer occurs at 1300 LT." How is this determined from your data? Please clarify.

**Response:** Thanks for your comment! Since the transition layer is a thin layer above the subcloud layer, it is challenging to observe it through satellite or to capture it with reanalysis data. In fact, this mechanism is mentioned to explain the diurnal variation of LWP skewness in ECS region. Clouds in ECS region is governed by transition regime. We used LWP skewness to indicate cumulus activity in our manuscript, which exhibits a significant diurnal variation. Miller et al. (1998) and Rogers et al. (1995) indicated that the cumulus activity in the transition region is determined by the stability of the transition layer. So we applied the theory of cumulus activity with respect to the transition layer stability proposed by to explain the diurnal variation of LWP skewness.

We have revised this section to clarify the descriptions, ensuring that speculation is clearly distinguished from the main conclusions of the paper (Lines 389-395): "In the ECS region, the weakest cumulus activity occurs at 1300 LT (the lowest LWP skewness in Figure 7B), which may be attributed

[revised manuscript text omitted]

---

## Referee Report (RR1)

I am satisfied with the authors' responses and the revised manuscript. I have only a couple of remaining comments:

1. The authors state that warm invigoration at ECS occurs at high Nd, which contrasts with existing literature suggesting that collision–coalescence processes are typically delayed at low Nd. It would be helpful if the authors could elaborate further on this apparent discrepancy.
2. Line 497: Mesoscale organization has been shown to cause mesoscale variability in Nd (Zhou and Feingold, 2023), which partially explains the observed inconsistency between Nd and aerosol.

*Zhou, X., & Feingold, G. (2023). Impacts of mesoscale cloud organization on aerosol-induced cloud water adjustment and cloud brightness. Geophysical Research Letters, 50(13), e2023GL103417.*

---

## Author Response (AR2)

**Response to Reviewer #1's Comments:**

Jiayi Li et al. (Author)

I am satisfied with the authors' responses and the revised manuscript. I have only a couple of remaining comments:

1. The authors state that warm invigoration at ECS occurs at high Nd, which contrasts with existing literature suggesting that collision–coalescence processes are typically delayed at low Nd. It would be helpful if the authors could elaborate further on this apparent discrepancy.

**Response:** Thanks for pointing this out! We acknowledge that the results in the ECS region are different from those in previous studies and attributing the positive LWP adjustments at high $N_d$ solely to warm invigoration is insufficient. As suggested by the second reviewer's feedback, we performed a more detailed analysis of the increase in LWP at high $N_d$ observed in the ECS region.

Our results indicate that this increase is likely attributed to the influence of cold air advection caused by prevailing northerly winds, especially for the period of spring and winter in the ECS region (Liu et al., 2016), which collectively account for ~75% of the samples. Coastal samples west of 125°E exhibit both high $N_d$ and LWP under this continental flow (Fig. R1, A and G), because the advection of dry, cold, aerosol-rich air over warm SST enhances surface sensible and latent heat fluxes (Fig. R1, F, H, J and K) (Long et al., 2020), raising saturation vapor pressure and activating cloud droplets. This could be confirmed by a pronounced drop in $\Delta\theta_{950\text{-surf}}$ (potential temperature difference between 950 hPa and 2 m above the sea surface), revealing strong sub-cloud destabilization (Fig. R1I). This leads to an increase in saturated water vapor pressure then promoting the cloud droplet activation. Concurrently, high LTS along the coast (Fig. R1L) suppresses vertical mixing at cloud top (Scott et al., 2020), allowing activated droplets to accumulate more liquid water with thicker clouds (Fig. R1B) and higher CF (Fig. R1C). These meteorological co-variabilities jointly elevate $N_d$ and LWP (and hence the $N_d$-LWP relationship) via cloud microphysical processes (Feingold et al., 2025).

[revised manuscript text omitted]

2. Line 497: Mesoscale organization has been shown to cause mesoscale variability in Nd (Zhou and Feingold, 2023), which partially explains the observed inconsistency between Nd and aerosol.

Zhou, X., & Feingold, G. (2023). Impacts of mesoscale cloud organization on aerosol- induced cloud water adjustment and cloud brightness. Geophysical Research Letters, 50(13), e2023GL103417

**Response:** We sincerely appreciate the reviewer's insightful reference to Zhou and Feingold (2023)and fully agree that mesoscale cloud organization can introduce variability in $N_d$ independent of aerosol loading. As highlighted by Zhou and Feingold (2023), mesoscale cellular convections (MCCs) introduce spatial variability in $N_d$ through scale-dependent processes. In small-scale MCCs, thicker cloud cores tend to have higher $N_d$ due to more homogeneous depletion. In large-scale MCCs, the thicker cloud cores have lower $N_d$ where precipitation scavenging dominates, while the non-precipitating cloud edges exhibit higher values.

We have added some discussions in the revised manuscript (see Lines 518-523): "Note that the correlations between AOD and $N_d$ at certain fixed times are not statistically significant (not shown). This may be due to the relatively insignificant impact of aerosol effects at these moments, while other processes may exert a more pronounced influence. For example, strong boundary layer decoupling inhibits cloud droplet activations (Zeider et al., 2025). Mesoscale cloud organization can also introduce spatial heterogeneity in $N_d$ independent of aerosol loading (Zhou and Feingold, 2023). Future research should broaden its scope to investigate the effects of other influencing factors on $N_d$ at specific times, in addition to the role of aerosols.".

**Reference**

Albrecht, B. A., Bretherton, C. S., Johnson, D., Scubert, W. H., and Frisch, A. S.: The Atlantic Stratocumulus Transition Experiment—ASTEX, Bulletin of the American Meteorological Society, 76, 889–904, https://doi.org/10.1175/1520-0477(1995)076<0889:TASTE>2.0.CO;2, 1995.

Bender, F. A.-M., Frey, L., McCoy, D. T., Grosvenor, D. P., and Mohrmann, J. K.: Assessment of aerosol–cloud–radiation correlations in satellite observations, climate models and reanalysis, Clim Dyn, 52, 4371–4392, https://doi.org/10.1007/s00382-018-4384-z, 2019.

Feingold, G., Glassmeier, F., Zhang, J., and Hoffmann, F.: Opinion: Inferring Process from Snapshots of Cloud Systems, EGUsphere, 1–28, https://doi.org/10.5194/egusphere-2025-1869, 2025.

Gryspeerdt, E., Goren, T., Sourdeval, O., Quaas, J., Mülmenstädt, J., Dipu, S., Unglaub, C., Gettelman, A., and Christensen, M.: Constraining the aerosol influence on cloud liquid water path, Atmos. Chem. Phys., 19, 5331–5347, https://doi.org/10.5194/acp-19-5331-2019, 2019.

Liu, J.-W., Xie, S.-P., Yang, S., and Zhang, S.-P.: Low-Cloud Transitions across the Kuroshio Front

in the East China Sea, Journal of Climate, 29, 4429–4443, https://doi.org/10.1175/JCLI-D-15-0589.1, 2016.

Long, J., Wang, Y., Zhang, S., and Liu, J.: Transition of Low Clouds in the East China Sea and Kuroshio Region in Winter: A Regional Atmospheric Model Study, Journal of Geophysical Research: Atmospheres, 125, e2020JD032509, https://doi.org/10.1029/2020JD032509, 2020.

Michibata, T., Suzuki, K., Sato, Y., and Takemura, T.: The source of discrepancies in aerosol–cloud–precipitation interactions between GCM and A-Train retrievals, Atmos. Chem. Phys., 16, 15413–15424, https://doi.org/10.5194/acp-16-15413-2016, 2016.

Rosenfeld, D., Zhu, Y., Wang, M., Zheng, Y., Goren, T., and Yu, S.: Aerosol-driven droplet concentrations dominate coverage and water of oceanic low-level clouds, Science, 363, eaav0566, https://doi.org/10.1126/science.aav0566, 2019.

Scott, R. C., Myers, T. A., Norris, J. R., Zelinka, M. D., Klein, S. A., Sun, M., and Doelling, D. R.: Observed Sensitivity of Low-Cloud Radiative Effects to Meteorological Perturbations over the Global Oceans, Journal of Climate, 33, 7717–7734, https://doi.org/10.1175/JCLI-D-19-1028.1, 2020.

Zeider, K., McCauley, K., Dmitrovic, S., Siu, L. W., Choi, Y., Crosbie, E. C., DiGangi, J. P., Diskin, G. S., Kirschler, S., Nowak, J. B., Shook, M. A., Thornhill, K. L., Voigt, C., Winstead, E. L., Ziemba, L. D., Zuidema, P., and Sorooshian, A.: Sensitivity of aerosol and cloud properties to coupling strength of marine boundary layer clouds over the northwest Atlantic, Atmospheric Chemistry and Physics, 25, 2407–2422, https://doi.org/10.5194/acp-25-2407-2025, 2025.

Zhang, X., Wang, H., Che, H.-Z., Tan, S.-C., Yao, X.-P., Peng, Y., and Shi, G.-Y.: Radiative forcing of the aerosol-cloud interaction in seriously polluted East China and East China Sea, Atmospheric Research, 252, 105405, https://doi.org/10.1016/j.atmosres.2020.105405, 2021.

Zhou, X. and Feingold, G.: Impacts of Mesoscale Cloud Organization on Aerosol-Induced Cloud Water Adjustment and Cloud Brightness, Geophysical Research Letters, 50, e2023GL103417, https://doi.org/10.1029/2023GL103417, 2023.

Jiayi Li et al. (Author)

**We highly appreciate the reviewer #2 for the valuable comments and constructive critiques, which have helped us improve the quality of our manuscript. Overall, we conducted further analysis of the V shape of LWP adjustments observed in the ECS region, including an investigation into the validity and potential causes of the results. Certain sections of the manuscript were reorganized to improve logical flow. Additionally, the language has been polished for clarity and readability.**

**In the following text, the reviewer's comments are listed in black, our response is in blue, and changes to the text are highlighted in red. A version of the revised manuscript with tracked changes is also provided.**

1. The paper investigates the relationship between LWP and Nd for 2 regions over the Himawari domain. It is not clear why the authors select these regions, nor the motivation to carry out a regional analysis instead of investigating the full Himawari domain. Also, the English and the writing needs substantial work beyond the suggestions I provide here.

**Response:** We sincerely appreciate the reviewer's comment. The West Australia (AUW: 25°–35°S, 95°–105°E) and East China Sea (ECS: 20°–30°N, 120°–130°E) were selected due to their contrasting cloud regimes and environmental conditions. The AUW region represents relatively pristine subtropical Sc region and the ECS region represents polluted Sc-Cu transition clouds affected by high anthropogenic aerosols. These regions offer contrasting aerosol and meteorological regimes, allowing us to isolate and quantify how different environmental conditions modulate the $N_d$-LWP relationship.

The full Himawari domain encompasses a mixture of deep convection, frontal systems, and mid-/high-level clouds whose macro- and micro-physical properties differ markedly from warm Sc. Including such diverse cloud types would introduce high uncertainty of LWP and $N_d$ retrievals (e.g., clouds with a thick optical thickness relative to stratocumulus clouds), large environmental heterogeneity, dilute the signal from the LWP adjustment mechanism we seek to investigate, and complicate the attribution of observed changes to aerosol versus meteorology. Nevertheless, even within the two chosen regions, the number of warm-cloud samples used in this study exceeds $10^5$, ensuring robust statistics. Further studies with specific considerations as discussed above could be carried out for domain-wide data.

In the revised manuscript, we have clarified the scientific motivation for regional selection (see Lines 63-68): "Within the sight of Himawari-8, we selected two cloud regions with significantly different environmental backgrounds (see Fig. S1 in Supplementary Materials). One is a remote stratocumulus region located in the west of Australia (AUW: 25º-35ºS, 95º-105ºE) (Klein and Hartmann, 1993). The other is in the East China Sea (ECS: 20º-30ºN, 120º-130ºE), which is significantly impacted by anthropogenic aerosols and characterized by Sc to Cu transition (Long et al., 2020). The comparison between the two regions allows us to explore the regional differences of LWP adjustments and their potential driving mechanisms.

[Figure]

**Figure S1. Distributions of cloud properties in two typical regions: the East China Sea (ECS: 20º-30ºN, 120º-130ºE) and the west of Australia (AUW: 25º-35ºS, 95º-105ºE).** (A) Geographical distribution of the view zenith angle of Satellite Cloud and Radiation Property retrieval System (SatCORPS) Himawari-8 data. The selected regions are marked by red boxes. Spatial distributions of cloud droplet number concentration ($N_d$) (B, E), effective radius ($r_e$) (C, F) and total column aerosol optical depth (AOD) (D, G) from MERRA-2 data are presented. The numbers in the lower right corner represent regional averages being weighted by the cosine of latitude.".

We are also grateful for your feedback regarding the language and writing of our manuscript. We have carefully revised the manuscript to address these issues and ensure that the language is clear and accurate.

2. It is relevant to note that the precipitation filtering based on GPM observations will not remove light precipitation nor drizzle, which is quite persistent in these clouds. This is because GPM can detect rather large hydrometeors more typical of convective systems. In other words, this study is not limited to non-precipitating clouds, so precipitation must be considered when interpreting the results.

**Response:** Thanks for pointing this out! We agree with the reviewer that GPM cannot identify light precipitation nor drizzle. In fact, the presence of precipitation is recognized as a significant source of uncertainty in $N_d$ retrievals (Grosvenor et al., 2018), thereby influencing the assessment of LWP adjustments. From the perspective of $N_d$ retrieval accuracy, we thus filtered out precipitation events

to minimize the uncertainty. Although the majority of heavy precipitation has been screened out, a considerable amount of light precipitation remains that cannot be effectively detected by GPM. Here, we also have adopted a critical effective radius ($r_e$) threshold of 14 μm (Rosenfeld et al., 2012) to distinguish between drizzle-like clouds and non-drizzle clouds (see the black lines in Fig. R1), which is another widely used precipitation-identification method. The result does show a substantial proportion of precipitating samples remain in the dataset.

In the revised manuscript, we have explicitly recognized this limitation, and precipitation is therefore fully considered throughout our discussion and results. For example, the drizzle-like samples in both regions exhibit a weaker negative LWP adjustment, which may be attributed to precipitation suppression driven by increased $N_d$ that weakens the entrainment-feedbacks (Fig. R1).

[Figure]

**Figure R1. Joint histograms of $N_d$ and LWP in log-log space in the AUW and ECS regions.** The column of each $N_d$ bin is normalized. The black lines are fitted based on the bins in the joint histogram with the effective radius ($r_e$) closest to 14 μm. The gray lines represent the contour of 5% occurrence. Orange dots represent the median LWP in each $N_d$ bins with a sample size greater than 50. The green and blue lines are regression slopes for the orange points with $r_e$ above and below 14 μm, respectively. **The figures are the same as Figure 1 in the revised manuscript.**

As suggested, the discussion of the precipitation filtering has been rephrased on Lines 126-131: "To minimize the influence of precipitation on $N_d$ and LWP retrievals, GPM IMERG Final Precipitation L3 Half Hourly 0.1 degree x 0.1 degree V07 (GPM_3IMERGHH) was used (Huffman et al., 2020). Cloud samples were included in the analysis only if the GPM_3IMERGHH precipitation rate equals 0 mm/hr in a 1° × 1° grid. To align these two satellite products, SatCORPS cloud pixels within each 0.1° grid of GPM_3IMERGHH are assigned the same precipitation value. Considering the limited ability of GPM to detect light precipitation and drizzle, we additionally applied a $r_e$ = 14 μm threshold to distinguish between drizzle scenes and non-drizzle scenes (black lines in Fig. 1).".

In addition, we have expanded the discussion to more explicitly address the influence of precipitation on our findings on Lines 192-196: "For instance, samples with $r_e$ > 14 μm—conditions more likely to contain drizzle (Rosenfeld et al., 2012)—still exhibit a weaker negative LWP

adjustment than those with $r_e < 14$ μm (Fig. 1, –0.22 vs. –0.47 in the AUW and -0.13 vs. -0.23 in the ECS), consistent with the results of Zhou and Feingold (2023) in the northwestern Atlantic. It suggests that in drizzle-like samples, the precipitation suppression partially offsets the dominant LWP reduction caused by the entrainment effect, resulting in a weak decrease in LWP with increasing $N_d$ compared to non-drizzle samples.".

3. The most interesting result of this study is the positive correlation between LWP and Nd for high values of Nd. The authors conducted a partial assessment of the relationship, but it is still a somewhat superficial analysis. A critical question is whether the finding is physical. To that end, it is recommended to analyze variability of Nd and LWP with cloud fraction because biases in broken scenes could well explain the relationship. Also the result might be dependent on the binning methods. Is the binning based on percentiles, that is, bins with the same number of samples?

**Response:** We appreciate the reviewer's insightful comments. As suggested, we have confirmed the robustness of the positive $N_d$-LWP relationship at high $N_d$ through further analysis.

(1) **Influence of broken scenes:** The impact of broken scenes on our results is extremely limited. Figure R2 presents the distribution of CF in $N_d$-LWP space and the sensitivity analysis of the results to different CF groups. The increase in LWP at high $N_d$ aligns with an increase in CF (Fig. R2A). Following Cao et al. (2023), a CF of 80% was used to distinguish between cloudy and broken scenes. For both cloudy scenes (CF > 80%) and broken scenes (CF < 80%), the increase in LWP at high $N_d$ consistently exists (Fig. R2, B and C). In addition, the average CF for samples of positive correlation between LWP and $N_d$ (> ~300 cm⁻³) is 86% (not the broken scenes). Therefore, the observed increase in LWP at high $N_d$ cannot be attributed to broken scenes.

[Figure]

**Figure R2. (A) The median cloud fraction (CF) in $N_d$-LWP log-log space in the ECS region.** Only the bins with at least 5% occurrence are shown, bounded by the gray line. Panels (B) and (C) show the normalized joint histograms of $N_d$ and LWP in log-log space in the ECS region, with CF greater than 80% and less than 80%, respectively. Black and orange dots represent the median LWP in each $N_d$ bins with a sample size greater than 50. **The figures are same as Figure S3 in the revised Supplementary Materials.**

(2) **Influence of binning methods.** This study employs equal-width binning based on equal $\log(N_d)$

intervals. To minimize the noise of sparse samples, only the $N_d$ bins containing more than 50 samples were used to calculate the LWP adjustment. The main reason for choosing equal-width binning was to preserve the original physical scale of the samples, avoiding the excessive smoothing of samples with diverse meteorological conditions gathered in a single bin using equal-sample binning (Towers, 2014). Figure R3 compares results using two binning methods, both consistently showing an LWP increase at high $N_d$. Thus, the positive $N_d$-LWP correlation at high $N_d$ is not artificially induced by the binning methodology.

[Figure]

**Figure R3. Comparison of the $N_d$-LWP relationship using two binning methods.**

Based on the above results, we have added detailed discussions on the impact of the broken scenes and the binning method in the revised manuscript:

Lines 178-182: "To investigate whether the positive $N_d$-LWP relationship is influenced by broken scenes, we assessed the sensitivity of our results to CF. As shown in Fig. S3, the rise in LWP at high $N_d$ coincides with an increase in CF. The average CF for samples with $N_d > 300$ cm$^{-3}$ is 86%. Additionally, the positive $N_d$-LWP relationship persists in both overcast (CF > 80%) and broken (CF < 80%) cloud scenes. This consistency indicates that the observed LWP increase at high $N_d$ is unlikely to be an artifact of broken-cloud scenes.".

Lines 141-147: "LWP adjustment at any given moment is the result of all available data at that moment. The regression slope of $N_d$ and LWP in log-log space ($\frac{\partial \ln LWP}{\partial \ln N_d}$) is calculated on 1° grid scale. We employed equal-width binning, using the median LWP within each $N_d$ bin to regress the slope. To reduce noise from sparse samples, only bins with more than 50 samples were used to calculate LWP adjustments. Additionally, we tested the equal-sample binning method. The patterns of the $N_d$-LWP relationship and diurnal variations of LWP adjustments remained robust across different binning methods. The main reason for choosing equal-width binning was to preserve the original physical scale of the samples, avoiding the excessive smoothing of samples with diverse meteorological conditions gathered in a single bin using equal-sample binning (Towers, 2014).".

For a detailed explanation of the positive correlation between LWP and $N_d$ at high $N_d$, please refer to the response to Comment 18.

4. Abstract, "Their reflectivity to solar radiation is highly sensitive to atmospheric aerosol concentrations because aerosols can serve as the cloud condensation nuclei (CCN) to modify the mediated variables (e.g. droplet number concentrations, Nd; effective radius, re) of aerosol-cloud interactions (ACI)". The sentence does not make sense grammatically.

…while holding cloud liquid water content (the Twomey effect) (Twomey, 1977). What does it mean?

**Response:** Thank you for your careful review. We acknowledge that the original sentences were not expressed clearly and have rephrased the sentences accordingly (see Lines 24-29): "Cloud reflectivity to solar radiation is highly sensitive to atmospheric aerosol concentrations. Because aerosols can serve as the cloud condensation nuclei (CCN), which modify key microphysical variables such as cloud droplet number concentrations ($N_d$) and droplet effective radius ($r_e$). For a given cloud liquid water content, aerosol-induced increases in CCN can enhance $N_d$ and hence reduce $r_e$, boosting cloud albedo (Twomey, 1977), which is known as cloud albedo effect, being an important component of aerosol-cloud interactions (ACI).".

5. Line 60, it should be "non-precipitating"

**Response:** Thank you for your careful review. The mistake has been corrected.

6. Line 83: "Nd can be estimated"….be more specific, did you use Eq. (1)?, If the answer is YES, then say it: "Nd IS estimated"

**Response:** Thanks for your comment! We confirm that $N_d$ was calculated by Eq. (1). We have revised the sentence as the reviewer suggested.

7. Line 77: "Briefly, CLTH is estimated as the altitude where the cloud-top temperature (CLTT) occurs in the temperature profile." This assertion is incorrect, SatCORPS does not directly use the profile and the retrieved temperature for deriving the cloud height. Instead, the method is based on a lapse rate approach. See Sun-Mack et al., 2014:

Sun-Mack, S., P. Minnis, Y. Chen, S. Kato, Y. Yi, S. C. Gibson, P. W. Heck, and D. M. Winker, 2014: Regional Apparent Boundary Layer Lapse Rates Determined from CALIPSO and MODIS Data for Cloud-Height Determination. J. Appl. Meteor. Climatol., 53, 990–1011, https://doi.org/10.1175/JAMC-D-13-081.1.

**Response:** We appreciate the reviewer's comment and apologize for the incorrect descriptions. We

have revised the sentence (see Lines 84-86): "Briefly, for boundary layer clouds, CLTH is retrieved using a lapse rate method: $\Gamma_b = (CET - T_0)/(CLTH - Z_0)$ (Sun-Mack et al., 2014). Cloud effective temperature (CET) was estimated from the Infrared Window (IRW) channel. $Z_0$ denotes the surface elevation and $T_0$ is the sea surface temperature.".

8. Line 74: "retrieve Nd" do you mean "calculate Nd"?

**Response:** Thanks for your comment! We have replaced 'retrieve' with 'calculate' to ensure an accurate description.

9. Line 76-77 The SatCORPS is based on the CERES Ed4 cloud retrieval algorithm, providing more accurate CLTH and H parameterizations (Minnis et al., 2011, 2021)." More accurate than what?

**Response:** Thanks for your comment! We have clarified the sentence in the revised manuscript(see Lines 82-83): "The SatCORPS product is based on the CERES Ed4 cloud retrieval algorithm (Minnis et al., 2021), which provides more accurate parameterizations of CLTH and H than the CERES Edition 2 retrieval algorithm (Minnis et al., 2011).".

10. Line 79: "CLTT is derived from an empirical parameterization of cloud-top emissivity at channel 4 and cloud effective temperature." This correction primarily applies to thin cirrus clouds. For boundary layer clouds, effective and cloud top temperature are identical. Remove the sentence.

**Response:** We are grateful for your comment! We have removed the sentence as suggested.

11. Line 82: "The SatCORPS retrievals provide cloud effective radius (re) in the 3.9 μm near-infrared band". Please, be accurate with your description. It is more clear to say: "SatCORPS cloud droplet effective radius (re) is primarily estimated from the 3.9um near-infrared band"

**Response:** We sincerely apologize for the inaccurate description. We have revised the sentence as suggested.

12. Line 86: "as Q relies less on the size parameter in near-infrared." The sentence does not make sense. Do you mean that Q varies little with particle size?

**Response:** Thank you for raising this question. We acknowledge that the description here is inaccurate. Q is the extinction efficiency factor. The size parameter is the ratio of the particle size to the wavelength ($\chi = \frac{2\pi r}{\lambda}$)is the particle size ($r$) and wavelength ($\lambda$). For liquid cloud droplets (~10 μm), the geometric optics limit is almost reached because $r \gg \lambda$ in the visible wavelength range (0.65–0.86 μm), and thus Q is approximately equal to its asymptotic value of 2 (Grosvenor et al., 2018). We have rephrased the sentence to be clear in the revised manuscript (see Line 92): "The extinction efficiency factor Q $\approx$ 2.".

13. Lines 106: "We followed the previous methods to filter cloud pixels. But this classification...",
Why classification? Please revise the sentence.

**Response:** Thank you for your comment. We have rephrased the sentence to be clearer (see Lines 111-112): "We followed the above methods to filter cloud pixels, which only limit cloud top properties and cloud phase, inevitably including different cloud regimes, such as low-level cumulus clouds.".

14. Line 106-108, Why do you say that departures from adiabaticity are different between cumulus and stratocumulus clouds? This is a speculative statement.

**Response:** Thank you for pointing this out. We agree that this is a speculative statement. However, it is grounded in multiple lines of published evidence rather than pure conjecture. Below shows the reasons.

Cumulus clouds are subject to stronger lateral entrainment driven by their larger vertical velocity shear and smaller aspect ratio (Heus et al., 2008). Stratocumulus, capped by strong inversions and often driven by cloud-top radiative cooling, entrains primarily at the top (Mellado, 2017); this ventilation affects the upper portion but leaves the lower portion closer to adiabatic. The difference in entrainment processes significantly impacts their adiabatic profiles. Small et al. (2013) found that the adiabatic fraction ($f_{ad}$, defined as the ratio of actual LWC to adiabatic LWC) of cumulus clouds showed no significant variation with height. However, Wood (2005) observed that the adiabaticity in stratocumulus clouds decreased from cloud base to cloud top.

Vertical soundings and aircraft penetrations repeatedly show that shallow continental cumuli often exhibit strong dilution and large negative buoyancy within a few hundred meters above cloud base (e.g. Drueke et al., 2020). In contrast, stratocumulus layers sampled during DYCOMS-II, VOCALS-REx and SOCRATES retain near-adiabatic cores over the upper 30-50% of the cloud layer (Stevens et al., 2003; Wang et al., 2021; Wood et al., 2011). The difference is quantified by the $f_{ad}$, whose median values are 0.25-0.40 for cumulus and 0.7-0.9 for stratocumulus. Except for in-situ aircraft data, satellite observations (e.g., MODIS/CloudSat) collocations show that the slope of LWP versus cloud depth is markedly lower in cumulus scenes than in stratocumulus (Michibata et al., 2021), implying larger departures from the adiabatic LWP profile.

Therefore, to ensure the accuracy of this statement, we have added the relevant references to support this speculative statement in the revised version (see Line 112-118): "This might introduce uncertainties as cumulus clouds and stratocumulus clouds have different adiabatic properties, but we have set $f_{ad}$ as a constant value in $N_d$ calculations. Small et al. (2013) found that the $f_{ad}$ of cumulus clouds showed no significant variation with height, whereas Wood (2005) observed that the

adiabaticity in stratocumulus clouds decreased from cloud base to cloud top. The difference in departures from adiabaticity between cumulus and stratocumulus stems from their different entrainment processes. Stratocumulus clouds are primarily influenced by the entrainment of dry air at cloud top (Mellado, 2017). In contrast, cumulus clouds are dominated by lateral entrainment (Heus et al., 2008).".

15. Line 115: "Consequently, the above uncertainties will not greatly affect the conclusions of this paper..." We do not really know. For instance, if retrievals uncertainties are a function of solar zenith angle, then uncertainties will change with local time... Also, hourly changes in entrainment rate can modify the Fad parameter.

**Response:** Thank you for your careful reading. We acknowledge that the above phrasing is not precise. To date, there is no robust method to improve the uncertainties discussed in our paper (namely, the impact of constant $f_{ad}$ and $k$ values). Consequently, $f_{ad}$ and $k$ are generally assumed to be constant in almost all studies investigating the diurnal variation of LWP adjustments based on geostationary satellite (Fons et al., 2023; Qiu et al., 2024; Smalley et al., 2024). This simplification is not arbitrary: in-cloud aircraft data show that $k$ is nearly invariant above the lower half of warm clouds (Brenguier et al., 2011; Martin et al., 1994), and satellite-based $f_{ad}$ climatology indicates standard deviations < 15% for extensive stratocumulus decks (Painemal and Zuidema, 2011). Nevertheless, $k$ and $f_{ad}$ do exhibit systematic vertical and cloud-type dependencies, and ignoring these variations can bias retrieved $N_d$ by up to 30 % (Grosvenor et al., 2018). In the absence of global, high-resolution $k$ and $f_{ad}$ products, however, constant values remain the best practical compromise. Dedicated airborne campaigns that simultaneously measure cloud microphysics, vertical structure, and radiation are therefore essential to reduce these residual uncertainties for future studies.

We have revised the relevant descriptions (see Lines 119-125): "As acquiring hourly $f_{ad}$ on a global scale is rather difficult, to date, studies investigating diurnal variations of LWP adjustments based on geostationary satellites continue to employ a constant $f_{ad}$ value (Fons et al., 2023; Qiu et al., 2024; Smalley et al., 2024). Also, the choices of a constant $k$ might introduce bias into the retrieval of $N_d$ (Grosvenor et al., 2018). Studies have found that $k$ parameter varied with the height within cloud and cloud types (Brenguier et al., 2011; Martin et al., 1994; Painemal and Zuidema, 2011). This indicates that the presence of diurnal variations in $k$ and $f_{ad}$ (e.g., hourly changes in entrainment rate can modify $f_{ad}$) introduces further bias. The resulting uncertainties warrant further in situ observation to improve the accuracy.".

16. Line 125: I disagree. I direct way to calculate the effects of aerosols on LWP is to actually use aerosol concentration rather Nd. True, we don't have aerosol retrievals for doing these….Please,

change the "direct way" sentence….perhaps you could say: "is the standard way to quantify LWP sensitivity to aerosol from satellite data".

**Response:** Thank you for your comment! We have revised the sentences as suggested (see Lines 133-136): "The logarithmic relationship between $N_d$ and LWP $(\frac{\partial \ln LWP}{\partial \ln N_d})$ is the standard way to quantify LWP sensitivity to aerosol from satellite data, where $N_d$ is considered a proxy of CCN. Another way of describing the changes of cloud water due to aerosols $(-\frac{\Delta \ln \tau}{\Delta \ln r_e})$ is deduced from the contributions of changes in LWP and $r_e$ to the changes in cloud optical depth $(\frac{\Delta \tau}{\tau} = \frac{\Delta LWP}{LWP} - \frac{\Delta r_e}{r_e})$.".

17. Line 144: "obtained or calculated by ERA5 reanalysis"….What does it mean? Just say that the atmospheric parameters are obtained from ERA5.

**Response:** Thanks for your suggestion. We have rephrased the sentence as suggested.

18. Line 176. "For non-precipitation clouds, both positive and negative LWP adjustments have been reported" This is correct but all these studies report an inverted-V shape, whereas this study show a V shape. This is a substantial difference, which needs to be further explored. Finding a V-shape is significant departure from previous studies and, thus, the authors should spend more time trying to understand these results and double test check the signature is real.

**Response:** We thank the reviewer for raising valuable suggestions and great questions! We conducted a more comprehensive assessment of the V-shape and attribute the difference between our results and the established inverted-V pattern primarily to two factors:

(1) **Limited pristine conditions in our study regions.** The inverted-V shape is associated with positive LWP adjustments at low $N_d$, which have been linked to precipitation suppression (Albrecht, 1989; Glassmiere et al., 2021). That is, as increasing $N_d$, the reduced $r_e$ may enhance the stability against coalescence and suppress the precipitation and loss of LWP (Albrecht, 1989; Glassmeier et al., 2021). The positive slopes at low $N_d$ are typically observed in very pristine conditions especially when $N_d$ is below approximately 10 cm$^{-3}$ (Fons et al., 2023; Goren et al., 2025). However, in this study, 98% of the AUW samples have $N_d$ above 15 cm$^{-3}$, and 99% of the ECS samples have $N_d$ above 30 cm$^{-3}$. This explains the absence of a significant positive slope at low $N_d$. However, this LWP increase resulting from precipitation suppression is still detectable in our study. In drizzle-like clouds ($r_e > 14$ μm), we observe a less negative slope than in non-drizzle clouds (Fig. R1, -0.22 vs. -0.47 in the AUW and -0.13 vs. -0.23 in the ECS), which aligns with results of Zhou and Feingold (2023) in the Northwest Atlantic. It suggests that in drizzle-like samples, the precipitation suppression partially offsets the dominant LWP

reduction caused by entrainment effect, resulting in a weak decrease in LWP with increasing $N_d$ compared to non-drizzle samples.

**(2) Strong transition of meteorological conditions across the turning point of V-shape.** Figure R4 shows the distribution of meteorological conditions in $N_d$-LWP log-log space. In the ECS region, where the V-shape is prominent, the samples exhibit strong geographic dependence. Coastal samples west of 125°E exhibit both high $N_d$ and LWP under northerly continental flow (Fig. R4, A and G),especially for the period of spring and winter in the ECS region (Liu et al., 2016), which accounts for 75% of the samples. Northerly advection of dry, cold, aerosol-rich air contrasts sharply with the warm ocean surface and enhances surface heat fluxes (SSHF) and surface latent fluxes (SLHF) from ocean to the atmosphere (Fig. R4, F, H, J and K) (Long et al., 2020), raising saturation vapor pressure and activating cloud droplets. This could be confirmed by a pronounced drop in $\Delta\theta_{950\text{-surf}}$ (potential temperature difference between 950 hPa and 2 m above the sea surface), revealing strong sub-cloud destabilization (Fig. R4I). Additionally, high LTS along the coast (Fig. R4L) suppresses vertical mixing at cloud top (Scott et al., 2020), allowing activated droplets to accumulate more liquid water with thicker clouds (Fig. R4B) and higher CF (Fig. R4C). These meteorological co-variabilities jointly elevate $N_d$ and LWP (and hence $N_d$-LWP relationship) via cloud microphysical processes (Feingold et al., 2025).

[revised manuscript text omitted]

19. Line 344 why is unexpected that the AOD diurnal cycle is flat? This is absolutely expected, especially over oceanic regions. Variations in AOD should be minimal, especially away from the aerosol sources.

**Response:** We agree with the reviewer's comment. We have removed the inappropriate description and revised the sentences (see Line 357-360): "$N_d$ continually declines from 0700 LT to 1600 LT and $r_e$ does not change significantly before 1200 LT and then rises. In contrast, there is no evident diurnal variation of AOD in the AUW, which is reasonable in the remote ocean area but insufficient to explain the diurnal variations of $N_d$ and $r_e$. This suggests that other factors rather than aerosols may be responsible for the diurnal variations of $N_d$ and $r_e$ over the AUW region.".

20. Line 437. The reference should be Painemal et al. 2017.

**Response:** Thanks for raising this question. We have corrected the reference.

21. Section 3.3. Without considering diurnal variations in Nd and albedo, the whole sensitivity calculation has no validity.

**Response:** We agree with the reviewer's point and acknowledge our oversight in the manuscript. To address this issue, we calculated cloud albedo ($A_c$) from cloud optical depth ($\tau$) based on a general expression for the two-stream approximation solution (Glenn et al., 2020):

$$A_c = \frac{\tau}{13.33 + \tau} \tag{1}$$

The susceptibility of $A_c$ to $N_d$ (S) is estimated as (Bellouin et al. 2020):

$$S = \frac{dA_C}{d \ln N_d} = \frac{A_C(1 - A_C)}{3}\left(1 + \frac{5}{2}\frac{d \ln LWP}{d \ln N_d}\right) \tag{2}$$

Based on the above formula, we calculated S using MODIS-averaged value (Terra at 1030 LT and Aqua at 1330 LT) for either $A_c$ or LWP adjustments while retaining the diurnal variation for the

other (Fig. R5). Given the minimal diurnal fluctuation in $\frac{A_C(1-A_C)}{3}$, the diurnal variations of S are mainly controlled by LWP adjustments. Therefore, considering the diurnal variability of $A_c$ would not significantly affect our results.

[Figure]

**Figure R5. The susceptibility (S) of $A_c$ to $N_d$ calculated by Eq. (2) under three input combinations in the (A) AUW and (B) ECS regions:** (1) diurnally varying $A_c$ and dlnLWP/dln$N_d$, (2) MODIS-averaged $A_c$ (Terra at 1030 LT and Aqua at 1330 LT) and diurnally varying dlnLWP/dln$N_d$, and (3) MODIS-averaged dlnLWP/dln$N_d$ and diurnally varying $A_c$. **The figures are the same as Figure S18 in the revised Supplementary Materials.**

In the revised manuscript, Section 3.3 has been rephrased and incorporated into Discussion section for better flow (see Lines 464-486):

"As discussed above, regional geostationary observations reveal the significant impact of regional diurnal dynamic processes on LWP adjustments, ranging from –0.41 to –0.27 in the AUW and from –0.11 to 0.21 in the ECS. Assuming a constant LWP adjustment based on polar-orbiting snapshots, rather than considering its diurnal variations will ultimately affect the estimation of the aerosol indirect effect. The cloud albedo ($A_c$) susceptibility to aerosols perturbations is estimated as (Bellouin et al. 2020):

$$S = \frac{dA_C}{d\ln N_d} = \frac{A_C(1-A_C)}{3}\left(1 + \frac{5}{2}\frac{d\ln LWP}{d\ln N_d}\right) \tag{6}$$

where S is the sensitivity of cloud albedo to $N_d$. $A_c$ is calculated from $\tau$ based on a general expression for two-stream approximation solution (Glenn et al., 2020):

$$A_c = \frac{\tau}{13.33 + \tau} \tag{7}$$

The first term of Eq. (6) refers to the changes in albedo due to the changes in $N_d$, while holding the LWP (i.e. Twomey effect). The second term, which accounts for LWP adjustment, can regulate the Twomey effect. The Twomey effect is completely offset when $\frac{d\ln LWP}{d\ln N_d}$ equals –2/5. Figure 7 shows the diurnal variations of S, calculated with Eq. (6) using the diurnal variations of both $A_c$ and LWP

adjustments. To isolate their individual influence, S was calculated using MODIS-averaged value for either $A_c$ or LWP adjustments while retaining the diurnal variation for the other (Fig. S18). Given the minimal diurnal fluctuation in $\frac{A_c(1-A_c)}{3}$, the diurnal variations of S are mainly controlled by LWP adjustments. According to Fig. 7, if S is evaluated only at fixed moments (e.g. the average value during MODIS overpasses for Terra at 1030 LT and Aqua at 1330 LT), the cooling effect of S is consistently underestimated before 1100 LT, with a maximum bias of 89% at 0800 LT. At 1300 LT, S even turns negative, suggesting that albedo decreases with increasing $N_d$, which has been reported in previous studies (Zhang et al., 2022). The negative S is possibly linked to strong decoupling over the AUW region at 1300 LT as discussed in Section 3.2. In the ECS region, the associated bias spans from a 24% overestimation at 0800 LT to a 40% underestimation at 1600 LT. The results highlight the critical need to account for diurnal variations of LWP adjustments when assessing the aerosol indirect effect. Future studies should incorporate geostationary observations or high-resolution simulations to better constrain the diurnal effects of LWP adjustments.

[Figure]

**Figure 7. The diurnal variations of S calculated by Eq. (6) in the (A) AUW and (B) ECS (blue lines).** The red dashed lines represent the average values during MODIS Terra (1030 LT) and Aqua (1330 LT) overpasses.".

22. Discussion section: The criticism of Qiu et al. (2024) does not make sense. I would argue that the methodology in Qiu et al. was more carefully crafted than the one in Li et al. 2025.Also the stratification by cloud thickness is flawed because both LWP and thickness are a function of optical depth, so both parameters are NOT independent.

**Response:** Thank you for your comments! As suggested by the previous reviewer, we added a comparison rather than criticism with Qiu et al. (2024) in the discussion section about similarities and differences in our conclusions. Both studies recognize that changes in cloud thickness drive the diurnal variations of LWP adjustments, yet the underlying physical interpretations are entirely different.

Qiu et al. (2024) calculated LWP adjustments within 1°×1° grid, which offers the advantage of

assuming negligible meteorological influence. In contrast, our approach calculates LWP adjustments using all 1° scenes. Furthermore, Qiu et al. (2024) focused on the evolution of cloud thickness at fixed grids, whereas in our study, thickness serves as an indicator to isolate meteorological covariations following Rosenfeld et al., (2019). Thus, we emphasize how varying meteorological and dynamical conditions at different times drive LWP adjustments.

We have rephrased the discussion about Qiu et al. (2024) to be clearer (see Lines 491-498): "Our observed diurnal LWP adjustment pattern in the AUW region is consistent with Qiu et al. (2024)'s findings in the eastern North Atlantic, where thick-thin cloud transitions dominated daytime variability. However, the main drivers emphasized in the two studies are different. Qiu et al. (2024) calculated LWP adjustment within each 1° grid box to minimize the meteorological covariations and highlighted cloud-intrinsic evolution, whereas we retain these covariations and then disentangle their influence by cloud thickness stratification analyses following Rosenfeld et al. (2019). Consequently, we attribute the diurnal variations in LWP adjustments mainly to temporal changes in meteorological and dynamical conditions. Additionally, after 1300 LT, cloud thickness remains relatively stable; the weakening of negative LWP adjustments is linked to reduced entrainment as large-scale subsidence strengthens (Fig. 4A).

[Figure]

**Figure 4. Diurnal patterns in the AUW region.** (A) Cloud liquid water path (LWP), cloud-top height (CLTH), cloud base height (CLBH), and vertical velocity on 700 hPa (omega700, positive values indicate downdraft) from ERA5 reanalysis. (B) LWP skewness and decoupling index in the AUW region. (C) Cloud droplet number concentration ($N_d$) and effective radius ($r_e$). (D) Aerosol optical depth (AOD).".

We agree that LWP and cloud thickness are not independent, but the stratification by cloud thickness remains physically meaningful. Cloud thickness typically serves as a mediator for large-scale meteorology (such as cold air advection, boundary layer decoupling, and surface heat fluxes) to influence LWP. These processes are particularly evident in the ECS region, where the increase in LWP at high $N_d$ corresponds with an increase in cloud thickness (Fig. R4B). Additionally, cloud thickness alters LWP by influencing cloud microphysical processes, such as promoting condensation growth. Fons et al. (2023) suggested cloud thickness is an important confounder in the $N_d$-LWP relationship that should be conditioned on. Constraining cloud thickness effectively limits the vertical development of clouds, which restricts the changes in LWP driven by vertical cloud evolution, thereby suppressing thickness-driven variability.

Once H is fixed within a narrow bin, the range of allowable LWP driven solely by vertical growth is capped. What remains is LWP variability arising from microphysics (droplet size, number), exactly the signal we want to isolate. Therefore, the stratification of cloud thickness can isolate a significant portion of covariations, highlighting the impact of $N_d$ on LWP. According to Rosenfeld et al. (2019), for a given cloud thickness, $N_d$ accounts for nearly half of the observed LWP variations. Therefore, although LWP, H, and optical depth are algebraically related, fixing H effectively decouples the dynamic envelope from the microphysics, making the $N_d$-LWP relationship conditionally independent and scientifically meaningful.

We have added discussions of using H for interpreting the $N_d$-LWP relationship (see Lines 261-293): "The above results suggest that the impact of large-scale meteorology on cloud microphysical processes ultimately determines the pattern of LWP adjustment. Previous studies employed various methods to exclude environmental confounding factors, such as opportunistic experiments from ship-track or volcano eruptions (Chen et al., 2022; Toll et al., 2019), where an overall weak LWP adjustment was observed. For satellite studies, Rosenfeld et al. (2019) pointed out that cloud thickness (H) constrained most of the meteorological impacts, and $N_d$ explained nearly half of the LWP variability for a given H. They demonstrated an overall positive LWP adjustment when separating H. However, we find that LWP adjustments become negative after constraining H in the intervals of Fig. 3 (B and E), indicating the dominant effect of entrainment-feedbacks. The discrepancy may arise from their focus on samples in convective cores (top 10% of cloud optical thickness), which are closer to adiabatic, whereas our samples suggest more exchange with the free atmosphere.

Here, our results indicate the physical significance of constraining H. In the AUW region, negative LWP adjustments become weaker as H increases (Fig. 3B). H alters LWP adjustments by influencing cloud microphysical processes, such as promoting condensation growth (Fons et al.,

2023). Thicker clouds with higher cloud-top $r_e$ are less sensitive to entrainment-feedbacks with increasing $N_d$ compared to thinner clouds. In other words, LWP in different H intervals responds differently to $N_d$, so it is necessary to restrict H to exclude the effects of covariations. However, in the ECS region, negative LWP adjustments for clouds with H < 900 m become stronger with increasing H, while for clouds with H > 900 m, quite the contrary: it weakens with increasing H (Fig. 3E). The bidirectional sensitivity of LWP adjustments to H is likely attributed to distinct mixing characteristics among different cloud regimes in the ECS region. Constraining H in the ECS region restricts a majority of mechanisms influencing cloud vertical development. Cloud thickness typically serves as a mediator for large-scale meteorology (such as cold air advection, LTS, and surface heat fluxes) to influence LWP. These processes are particularly evident in the ECS region, where the increase in LWP at high $N_d$ corresponds with an increase in cloud thickness (Fig. 2B). Therefore, the stratification of cloud thickness can isolate a significant portion of covariations, highlighting the impact of $N_d$ on LWP.

[Figure]

**Figure 3. LWP adjustments in log-log spaces and their diurnal patterns in two typical regions (the west of Australia, AUW and the East China Sea, ECS).** Cloud samples are scattered in $N_d$-LWP log space at 1400 LT in the (A) AUW and (D) ECS region. The complete pictures of all available daytime are presented in Fig. S11. Colored dots are samples in different cloud thickness (H) bins (unit: m). Black dots represent the median LWP in each $N_d$ bin. The colored lines are the fits of black dots at different stages in the ECS region. Diurnal variations of LWP adjustments binned by H in the (B) AUW and (E) ECS regions are shown. Colored lines in (F) are diurnal variations of different stages in (D), while black lines in (C) and (F) are the overall diurnal variations of LWP adjustments in two regions, respectively. The blue line in (C) represents the diurnal variation of H. Red dashed lines represent the average LWP adjustments during MODIS Terra (1030 LT) and Aqua (1330 LT) overpasses, –0.39 for the AUW region (C) and –0.03 for the ECS region (F).".

23. What is the physical process that would explain the convective invigoration? Why is this the only plausible explanation. For example, postfrontal conditions and cold-air outbreaks could explain both an increase in both LWP and Nd. In my opinion, the convective invigoration explanation is not proved by the analysis. The use of LTS is not optimal because LTS explains a modest fraction of cloud variance, so not finding a relationship between cloud microphysics and LTS does not mean much.

**Response:** We highly appreciate these constructive comments raised by the reviewer. We agree that attributing the positive LWP adjustments at high $N_d$ solely to warm invigoration is insufficient. As described above in response to Comment 18, our revised analysis confirms that large-scale meteorology, especially the cold air advection, is the main driver of the observed LWP increase at high $N_d$ (> ~300 cm$^{-3}$) in the ECS region. This is supported by the significant meteorological transitions across the turning point of the V shape (Fig. R4). The aerosol-rich, relatively cold and dry air from continent reduces the stability of the sub-cloud layer, triggering the release of water vapor into the boundary layer and subsequently promoting cloud droplet activation and development of thicker clouds. Additionally, we agree that LTS explains only a modest fraction of cloud variance, while the increase in LWP at high $N_d$ results from the combined influence of meteorological conditions. Thus, we incorporated additional variables in Fig. R4 to disentangle the impact of meteorological factors on cloud microphysical processes.

We acknowledge cold air outbreaks (CAOs) could explain the increase in both $N_d$ and LWP, but CAOs are more likely to be a strong form of cold air advection. Our analysis reveals the seasonal dependence of CAOs. Following Papritz et al. (2015), the CAO Index (CAOI) was calculated as the difference in potential temperature between the surface skin and 850 hPa. CAO events are identified when the CAO Index is greater than zero. CAOs are most pronounced in autumn and winter, with no significant occurrence in spring (Fig. R6). Results of summer are statistically insignificant due to the limited samples (3%), especially after eliminating the samples with strong precipitation by applying the threshold (GPM = 0 mm hr$^{-1}$). The seasonal variations are consistent with the East Asian monsoon, where strong northerly winds prevail in winter but weaken in spring (Liu et al., 2016), leading to reduced CAOI. However, the impacts of cold air advection are prevalent throughout the seasons (Fig. R7), making it a more plausible reason for the observed sub-cloud destabilization and subsequent increases in $N_d$ and LWP.

We have added the discussion of CAOs in the revised manuscript (see Lines 244-253): "While cold air outbreaks (CAOs) also contribute to the observed increases in both $N_d$ and LWP, our analysis suggests that cold air advection is a more consistent and seasonally pervasive driver and CAOs represent a strong form of cold air advection. Following Papritz et al. (2015), the Cold Air Outbreak

Index (CAOI) was calculated as the difference in potential temperature between the surface skin and 850 hPa. CAO events are identified when CAOI > 0. Our results indicate that CAOs are most pronounced in autumn and winter, with no significant occurrence in spring (Fig. S5). Results of summer are statistically insignificant due to the limited samples (3%), particularly after excluding cases with strong precipitation (GPM = 0 mm hr⁻¹). The seasonal variations are consistent with the East Asian monsoon, where strong northerly winds prevail in winter but weaken in spring (Liu et al., 2016), leading to reduced CAOI. In contrast, the impacts of cold air advection are prevalent throughout the seasons (Fig. S6), making it a more plausible reason for the observed sub-cloud destabilization and subsequent increases in $N_d$ and LWP.".

[Figure]

**Figure R6. The median cold-air outbreak index (CAOI) in $N_d$-LWP log-log space for spring, autumn, and winter in the ECS region.** Among the total samples (173181), spring, summer, autumn, and winter account for 31%, 3%, 22%, and 44%, respectively. Due to the limited samples (3%), results of summer are statistical insignificance, especially after eliminating the samples with strong precipitation by applying the threshold (GPM = 0 mm hr⁻¹). **The figures are the same as Figure S5 in the revised manuscript.**

[Figure]

**Figure R7. The median horizontal temperature advection at the surface (SST$_{adv}$) in $N_d$-LWP log-log space for spring, autumn, and winter in the ECS region. The figures are the same as Figure S6 in the revised manuscript.**

---

## Author Response (AR3)

**Response to Reviewer #1's Comments:**

Jiayi Li et al. (Author)

The manuscript has been substantially improved. My major concerns has been addressed. However, with the inclusion of new material, there are a number of lingering suggestions the authors should consider before their manuscript is accepted for publication. My main concern is on the interpretation of the V shape. Based on the new material, even though the V shape is clear in the analysis, this is not the manifestation of physical processes, instead, the artifact of spatial variability included in the analysis. In other words, in an ideal observational system, where it is possible to compute statistics at domains smaller than 1deg, the V shape disappears.

**Response:** We sincerely appreciate the reviewer's constructive comments, which have helped us improve the manuscript and clarify the interpretation of the V shape. In general, we agree with the reviewer that the V shape is an artifact of spatial variability in the ECS region rather than a robust physical signal.

As detailed in our responses to Comments #8 to #10, we have: (a) explicitly discussed the potential influence of spatial sampling inhomogeneity, (b) revised the relevant sections to avoid over-interpretation of the V shape, and (c) toned down any claims that could imply it is a genuine physical feature.

We believe these revisions make the interpretation more cautious and consistent with the limitations of the current observational configuration. Further investigations with sub-degree spatial sampling would indeed be valuable and are now mentioned as a future research direction.

**Below, we provide a point-by-point response in blue. The changes to the text are highlighted in red. A version of the revised manuscript with tracked changes is also provided.**

1. Line 11: "geostationary observations were conducted"? you mean "Himawari-8 retrievals were used to investigate ACI in 2 regions"?

**Response:** The reviewer's interpretation is correct. We have rephrased the sentence as suggested.

2. Line 15: "…is driven primarily by diurnal-related boundary layer decoupling and cloud top entrainment" Since the entrainment was not directly estimated, the statement is speculative and should be modified accordingly.

**Response:** Thank you for your careful review. We have rephrased the sentence to be more accurate (see Lines 16-17): "Furthermore, the diurnal variation of LWP adjustments is likely driven by cloud-top entrainment in the ECS region, but is primarily associated with diurnal-related boundary layer

decoupling in the AUW region.".

3. Line 32: The Albrecht 1989 citation is incorrect in the context of the sentence.

**Response:** We thank the reviewer for this comment. The Albrecht 1989 citation is for the precipitation suppression due to life time effect. We have rephrased the sentence to be more accurate (see Lines 31-32): "For example, it has been documented that liquid water path (LWP) can either increase (positive LWP adjustments) due to precipitation suppression (Albrecht, 1989).".

4. Line 34: "…as the least understood…" the phrase is not needed.

**Response:** Thank you for your comment. We have removed the phrase.

5. Line54-61. Qiu et al. (2024) needs to be described here because the paragraph is misleading.

**Response:** We appreciate the reviewer for this insightful comment. We have rephrased the paragraph as suggested (see Lines 54-66): "To date, a majority of studies have relied on observations from polar-orbiting satellites to investigate LWP adjustments and cloud microphysical properties (Bennartz and Rausch, 2017; Gryspeerdt et al., 2019; Li et al., 2018; McCoy et al., 2018; Rosenfeld et al., 2019), which are insufficient to depict the time-dependent nature of LWP adjustments. Recent studies began to emphasize the diurnal evolutions of LWP adjustments with geostationary satellites. For example, Rahu et al. (2022) detected polluted cloud tracks over Eastern Europe and revealed afternoon LWP increases in some cases. Qiu et al. (2024) identified a distinct "U-shaped" diurnal pattern of LWP adjustments over the Eastern North Atlantic. However, the understanding of diurnal LWP adjustments, particularly their interplay with varying meteorological conditions and boundary layer dynamic mechanisms, remains limited. Based on Himawari-8 geostationary satellite, the diurnal variations of cloud microphysical properties and LWP adjustments in two typical regions, and the associated influencing factors and mechanisms are presented in this study. Our research aims to expand our understanding of the influence of meteorological factors, initial aerosol states (especially $N_d$), and the covariance between meteorology and aerosols on cloud LWP, gaining a comprehensive understanding of the diurnal variations in LWP adjustments, which is a highly time-dependent variable lacking quantification, in conjunction with shifts in regional meteorological conditions.".

6. Line 130 where is the re=14 um threshold coming from?

**Response:** Thanks for pointing this out. The $r_e = 14\,\mu m$ threshold is based on Rosenfeld et al. (2012). Their results show that the precipitation intensified quickly when $r_e$ exceeds 14 μm (> 2.5 mm d$^{-1}$). The threshold has been widely used in satellite-based studies to distinguish between drizzle scenes and non-drizzle scenes (e.g., Gryspeerdt et al., 2019; Rosenfeld et al., 2019; Zhou and Feingold,

2023).

We have rephrased the sentence to be more accurate (Lines 132-134): "Considering the limited ability of GPM to detect light precipitation and drizzle, we additionally applied a $r_e = 14$ µm threshold to distinguish between drizzle scenes and non-drizzle scenes (Rosenfeld et al., 2012)".

7. Line 145-148. This justification doesn't make sense. Could you clarify it please?

**Response:** Thank you for raising this question. We acknowledge that the description is inaccurate. Both equal-sample bins and equal-width bins have been employed in previous studies with comparable results (Fons et al., 2023; Goren et al., 2025; Gryspeerdt et al., 2019; Qiu et al., 2024; Rosenfeld et al., 2019; Zhou and Feingold, 2023). Given our substantial sample size (480189 cloud samples in the AUW and 173181 cloud samples in the ECS) and to ensure consistency with the subsequent analysis of joint histograms of $N_d$ and LWP in log-log space, we employed equal-width binning. Specifically, the median LWP within each $\log(N_d)$ bin was used to regress the slope.

We apologize for the misleading and have rephrased the sentence to be more accurate (see Lines 145-149): "Given our substantial sample size and to ensure consistency with the subsequent analysis of the joint histograms of $N_d$ and LWP in log-log space, we employed equal-width binning, using the median LWP within each $\log(N_d)$ bin to regress the slope. To reduce noise from sparse samples, only bins with more than 50 samples were used to calculate LWP adjustments. Additionally, we tested the equal-sample binning method. The patterns of the $N_d$-LWP relationship and diurnal variations of LWP adjustments remained robust across different binning methods.".

8. Page 7, first paragraph. The results for ECS do not contradict Grsypeerdt et al. (2019). Note that the Nd range in Gryspeerdt does not exceed 300 /cc. In contrast Fig 1b depicts datapoints > 300 /cc. And the increase in Nd with LWP in SatCORPS NASA retrievals occur for Nd>300. My interpretation is that the East China Sea region and its range of variability is not represented in previous studies, but, again, I don't see necessarily a discrepancy. The question is whether this increase in Nd with LWP is also observed in regions over land, where one should expect high Nd values.

**Response:** We thank the reviewer for this insightful comment. We agree that our results over the ECS region do not contradict previous studies, and the difference is largely due to the sampling range. To elucidate the influence of the $N_d$ range, we selected a polluted continental region in the west of the ECS region, where 46% of samples exhibit $N_d > 300$ cm⁻³, and a pristine oceanic region, with only 5% of high $N_d$ samples (Fig. R1A). The continental region exhibits a general increase in LWP with $N_d$ (Fig. R1B). In contrast, the oceanic region shows a general decrease in LWP with $N_d$ (Fig. R1C). The result demonstrates that the V shape in the ECS region results from the mixing of

continental and oceanic air masses within the coastal zone.

[Figure]

**Figure R1: Normalized joint histograms of $N_d$ and LWP for all samples in log-log space (B) in the west land and (C) in the east ocean of the East China Sea.** Panel (A) shows the geographical location of the selected regions (ECS: 20°-30°N, 120°-130°E, LAND: 25°-30°N, 113°-118°E, OCEAN: 20°-25°N, 132°-137°E). Orange dots in panels (B) and (C) represent the median LWP in each $N_d$ bins with a sample size greater than 50. The black lines are fitted based on the bins in the joint histogram with the effective radius ($r_e$) closest to 14 μm. Gray contours indicate 5% occurrence levels. **The figure is the same as Figure S4 in the revised manuscript.**

We have revised the paragraph as suggested (see Lines 198-213): "According to Fig. 1B, LWP begins to rise at high $N_d$ (> ~300 cm$^{-3}$), exhibiting a V shape that dominates the overall positive LWP adjustments in the ECS region. Although a positive sensitivity of LWP to $N_d$ perturbations has been reported (Bender et al., 2019; Gryspeerdt et al., 2019; Michibata et al., 2016; Zhang et al., 2021), its origin remains unclear. To investigate whether the positive $N_d$-LWP relationship is biased by broken scenes, we assessed the sensitivity of our results to CF. As shown in Fig. S3, the rise in LWP at high $N_d$ coincides with an increase in CF. The average CF for samples with $N_d > 300$ cm$^{-3}$ is 86%. The positive $N_d$-LWP relationship persists in both overcast (CF > 80%) and broken (CF < 80%) cloud scenes, but the magnitude of the positive adjustment is markedly larger under high-CF scenes. Thus, the LWP rise at high $N_d$ is unlikely to be an artifact of broken-cloud scenes; rather, overcast environments amplify the positive LWP adjustment.

It is plausible that the unique $N_d$-LWP relationship over the East China Sea is closely related to heavy pollution advected from the continent. Previous studies of LWP adjustments have typically concentrated on the range of $N_d$ below 300 cm$^{-3}$. However, 18% of the samples exhibited $N_d$ values exceeding 300 cm$^{-3}$ in the ECS region, where clouds are downwind of the major emission sources of China. We further compare the $N_d$-LWP relationships over the continental region (46% of

samples have $N_d > 300$ cm$^{-3}$) in the west of ECS and the oceanic region (5% of samples have $N_d > 300$ cm$^{-3}$) in the east of ECS, respectively (Fig. S4). Results demonstrate that LWP generally rose with increasing $N_d$ on the heavily polluted continent, while LWP declined with $N_d$ over the ocean. The opposing $N_d$-LWP relationships correspond to the ascending and descending branches of the V shape, which indicates that V shape results from mixing samples of continental and oceanic air masses.

[Figure]

**Figure 1: Joint histograms of $N_d$ and LWP in log-log space in the AUW and ECS regions (the west of Australia, AUW and the East China Sea, ECS).** The column of each $N_d$ bin is normalized. The black lines are fitted based on the bins in the joint histogram with the effective radius ($r_e$) closest to 14 μm. The gray lines represent the contour of 5% occurrence. Orange dots represent the median LWP in each $N_d$ bins with a sample size greater than 50. The green and blue lines are regression slopes for the orange points with $r_e$ above and below 14 μm, respectively.

[Figure]

**Figure S3: (A) The median cloud fraction (CF) in $N_d$-LWP log-log space in the ECS region.** Only the bins with at least 5% occurrence are shown, bounded by the gray line. Panels (B) and (C) show the normalized joint histograms of $N_d$ and LWP in log-log space in the ECS region, with CF greater than 80% and less than 80%, respectively. Black and orange dots represent the median LWP in each $N_d$ bins with a sample size greater than 50.".

9. Page 7 second paragraph. Insisting in the same topic of Nd range; it appears plausible that the unique behavior over the East China Sea is contributed by heavy pollution advected from the continent. In other words, meteorology is not the single factors that can modulate the relationship.

**Response:** Thank you for your comment! As described above in response to Comment #8, we have revised the manuscript to emphasize the impact of $N_d$ range.

10. Figure 2: This is an interesting analysis. It is also relevant because it explains 2 things: 1) The v-shape is the manifestation of spatial variability. In other words, the behavior is the consequence of aggregating data from different locations (Fig. 2a). If the analysis was constrained to a narrower region (e.g. the region west of 124˚W), the so called v-shape disappears, and would be replaced by a single regime: a linear increase of Nd with LWP. Second, the linear increase of Nd comes from winter observations (cold SST). All these aspects should be discussed in the paper. Lastly, I would like to emphasize that the v-shape is an artifact, as 2 different sub-domains contribute, individually, to this shape, but in reality, it does not represent the cloud behavior. Having said that, it is still interesting that Nd increases with LWP in winter for coastal clouds. Figure 2 warrants changes to the abstract and the discussion/conclusion section. The importance of this artificial v-shape, attributed to spatial variability, should be deemphasized, while the rather unique increase of Nd with LWP over the East China Sea should be highlighted.

**Response:** Thank you very much for the constructive comments on our manuscript. We agree with the reviewer that V shape is a result of the spatial variability in the ECS region. Fig. R1 has demonstrated the spatial heterogeneity. Samples in the ECS region are a mixture of continental and oceanic samples. We further performed a sub-domain analysis in the coast and offshore. As shown in Fig. R2, LWP adjustments demonstrate a clear spatial gradient, transitioning from positive values near the coast to negative values offshore. The spatial distribution of LWP adjustments is consistent with $N_d$. For coastal grids with high $N_d$ values, LWP increases with $N_d$, particularly after ~300 cm$^{-3}$ (Fig. R2B), while for offshore grids where $N_d$ values are concentrated below ~300 cm$^{-3}$, LWP decreases with $N_d$ (Fig. R2C). Therefore, the V shape is the manifestation of spatial variability in the ECS region. We have clarified this in the revised manuscript and, as suggested, provided a more detailed discussion of Figure 2, including the spatial variability and seasonality of the samples (see Lines 224-243): "To further investigate the spatial variability of the ECS region and the potential reasons for the rising behavior of LWP at high $N_d$, we followed the method of Goren et al. (2025) to present the distribution of environmental conditions in $N_d$-LWP space (Fig. 2). The cloud samples in the ascending branch are concentrated west of 125°E (Fig. 2A). We performed a sub-domain analysis in the coast and offshore (Fig. S5). LWP adjustments demonstrate a clear spatial gradient, transitioning from positive values near the coast to negative values offshore. The spatial distribution of LWP adjustments is consistent with $N_d$. For coastal grids with high $N_d$ values, LWP increases with $N_d$, particularly after ~300 cm$^{-3}$, while for offshore grids where $N_d$ values are concentrated below ~300 cm$^{-3}$, LWP decreases with $N_d$. Therefore, the V shape is the manifestation of spatial variability in the ECS region. The observed increase in LWP at high $N_d$ is attributed to samples from the coastal area. These coastal samples are characterized by cold SST (Fig. 2E), since 75% of the samples are from spring and winter when the Kuroshio Current produces a sharp SST gradient in

the ECS region (Fig. S6) (Liu et al., 2016). Results of summer are statistically insignificant due to the limited samples (3%), particularly after excluding cases with strong precipitation (GPM = 0 mm hr$^{-1}$). Such seasonal patterns indicate that the samples are strongly influenced by the northerly cold-air advection at the surface that destabilizes the air-sea interface (Fig. 2, F and G). The potential temperature difference between 950 hPa and 2 m above the sea surface ($\Delta\theta_{950\text{-surf}}$) is calculated as an indicator of sub-cloud layer stability, revealing an extremely unstable sub-cloud layer in the ascending branch (Fig. 2I). Northerly winds transport relatively dry, cold, aerosol-rich air across the warm ocean (Fig. 2, F, G, and H). This destabilizes the sub-cloud layer and intensifies the upward fluxes of sensible and latent heat from sea surface into the atmosphere (Fig. 2, I, J and K) (Long et al., 2020), raising saturation water vapor pressure and facilitating cloud droplet activation. Additionally, high LTS along the coast (Fig. 2L) suppresses the entrainment drying at cloud top (Scott et al., 2020), allowing activated droplets to accumulate more liquid water with thicker clouds (Fig. 2B) and higher CF (Fig. 2C). These conditions jointly elevate both $N_d$ and LWP, resulting in the observed increasing LWP at high $N_d$.

[Figure]

**Figure 2: Distributions of meteorological conditions in $N_d$-LWP log-log space in the ECS region.** The color scale represents the median values in each bin. Only bins with an occurrence of at least 5% are shown, bounded by the gray lines. (A) Longitude. (B) Cloud thickness. (C) Cloud fraction (CF). (D) Cloud effective radius ($r_e$). (E) Sea surface temperature (SST). (F) Horizontal temperature advection at the surface (SST$_{adv}$). (G) Wind direction on 1000 hPa. 0° indicates a northerly wind. (H) Relative humidity on 1000 hPa (RH1000). (I) The potential temperature difference between 950 hPa and 2 m above the sea surface ($\Delta\theta_{950\text{-surf}}$), a proxy of the sub-cloud layer stability. (J) Surface sensible heat flux (SSHF). (K) Surface latent heat flux (SLHF). For the vertical fluxes, the negative is upwards. (L) Lower-tropospheric stability (LTS). Black dots represent the median LWP in each $N_d$ bins with a sample size greater than 50.

[Figure]

**Figure S6: Distributions of meteorological factors and different cloud regimes in the ECS region.** (A) Sea Surface Temperature (SST), the composite wind field (arrows) on 700 hPa. The numbers in the lower right corner represent regional averages being weighted by the cosine of latitude. Distributions of the proportion of cloud regimes for (B) Stratocumulus (Sc, LTS > 18 K), (C) Cumulus (Cu, LTS < 14 K), (D) Sc to Cu transition regime (Trans, 14 K <= LTS <= 18 K) are shown.".

[Figure]

**Figure R2:** (A) Spatial distribution of LWP adjustments over the ECS region. White asterisks mark the coast and offshore areas for which normalized joint histograms of $N_d$ and LWP in log-log space are presented in panels (B) and (C), respectively. Orange dots in (B) and (C) represent the median LWP in each $N_d$ bins with a sample size greater than 20. Gray contours indicate 5% occurrence levels. **The figure is the same as Figure S5 in the revised manuscript.**

Additionally, we have revised the Abstract and Conclusion sections to deemphasize the V shape and instead highlight the unique positive $N_d$-LWP relationship found in the coastal region as the key discovery:

Abstract (see Lines 8-21): "Aerosol-cloud interaction (ACI) remains a key uncertainty in climate projections. A major challenge is that the sign and magnitude of cloud liquid water path (LWP)

response to aerosol perturbations (represented by cloud droplet number concentration, $N_d$) at different temporal and spatial scales are highly variable, but potential microphysical-dynamical mechanisms are still unclear, especially at a diurnal scale. Here, Himawari-8 retrievals were used to investigate LWP adjustments in two distinct cloud regions: the stratocumulus region off the western Australia (AUW) and clouds over the East China Sea (ECS) characterized by a transition from stratocumulus to cumulus under strong anthropogenic influences. In the ECS region, LWP exhibits a unique pronounced rising (positive LWP adjustments) at high $N_d$. Results indicate that this pattern is driven by northerly cold-air advection during the cold seasons, which enhances surface fluxes and subsequently leads to increases in both LWP and $N_d$. Furthermore, the diurnal variation of LWP adjustments is likely driven by cloud-top entrainment in the ECS region, but is primarily associated with diurnal-related boundary layer decoupling in the AUW region. The results indicate that neglecting diurnal variations of LWP adjustments leads to an underestimation (up to 89%) of the cooling effect induced by changes in cloud albedo due to aerosol perturbations in the AUW. The bias spans from a 32% overestimation to a 37% underestimation in the ECS. Our findings highlight the key role of diurnal variations of ACI in reducing the uncertainty in climate projections.".

Conclusion (see Lines 519-528): "In the ECS region, LWP increases at high $N_d$ (> ~300 cm$^{-3}$), leading to a V shape pattern of $N_d$-LWP relationship. Our results demonstrate the V shape is the manifestation of spatial variability in the ECS region. Specifically, the ECS serves as a coastal transition zone, experiencing the combined effects of continental and oceanic air masses. Coastal samples show a pronounced LWP increase with $N_d$, particularly for $N_d$ > ~300 cm$^{-3}$, while offshore samples exhibit a negative $N_d$-LWP relationship, with most $N_d$ values below ~300 cm$^{-3}$. The unique behaviour of LWP increasing at high $N_d$ is attributed to the influence of northerly cold-air advection from the continent during the cold seasons (75% of samples are from spring and winter). The aerosol-rich, relatively cold and dry air from continent reduces the stability of the sub-cloud layer, triggering the release of water vapor into the boundary layer and subsequently promoting cloud droplet activation and development of thicker clouds. These processes collectively lead to an increase in both $N_d$ and LWP, resulting in a positive LWP adjustment at high $N_d$.".

11. Page 9, line 244-246. CAO is an extreme manifestation of cold advection, but the idea remains the same, that is, surface fluxes in the cold season are strong, attributed to changes between air-sea temperature differences, which strongly modulate cloud variability.

**Response:** We appreciate the reviewer's comment. We have removed the discussion of CAOI but emphasized the seasonal pattern of the samples. Specifically, samples from the cold seasons are subject to cold advection, resulting in both increase in LWP and $N_d$ (see Lines 231-236): "The observed increase in LWP at high $N_d$ is attributed to samples from the coastal area. These coastal

samples are characterized by cold SST (Fig. 2E), since 75% of the samples are from spring and winter when the Kuroshio Current produces a sharp SST gradient in the ECS region (Fig. S6) (Liu et al., 2016). Results of summer are statistically insignificant due to the limited samples (3%), particularly after excluding cases with strong precipitation (GPM = 0 mm hr⁻¹). Such seasonal patterns indicate that the samples are strongly influenced by the northerly cold-air advection at the surface that destabilizes the air-sea interface (Fig. 2, F and G).".

12. I still find it unnecessary to use cloud thickness to constrain the analysis of LWP. First, as I said in my first review, both LWP and physical depth are functions of optical depth and, therefore, they are not independent variables. Second, there is no physical algorithm for deriving cloud physical depth and, thus, any method is based on empirical relationships, which are highly uncertain. The fact that a remote sensing group provides a product it does not mean that this is suitable for atmospheric research.

**Response:** Thank you for pointing this out! We acknowledge that cloud thickness (H) should not be used to explain the $N_d$-LWP relationship, and we have and removed the section using H to constrain the analysis of LWP adjustments.

However, using H to explain the diurnal variation of LWP adjustment is still physically meaningful, especially in the AUW region. H is the result of the boundary layer dynamical condition. Therefore, we emphasize that the different physical processes under differing H conditions determine the diurnal variations of LWP adjustments. Specifically, in the morning thicker clouds occur in a well-mixed boundary layer with stronger cloud-top entrainment and sufficient upward moisture transport (Lu et al., 2023; Zheng et al., 2018b). LWP is less sensitive to entrainment-feedbacks with increasing $N_d$ in thicker clouds. LWP adjustments become more negative with the gradual thinning of clouds. This can be attributed to the weaker moisture transport and cloud-top entrainment in the decoupling boundary layer. As a result, the entrainment feedback with increasing $N_d$ has a greater influence on LWP in thinner clouds, corresponding to more negative LWP adjustments.

Considering H from the passive retrievals has uncertainties, we further perform a validation using the 2B-GEOPROF-LIDAR product from CloudSat-CALIPSO, where H is directly calculated by the cloud base height (CBH) and cloud top height (CTH) of single-layer cloud samples. H from the active sensor was regarded as a reference value of cloud geometrical thickness in previous studies (Zhang et al., 2025) and independent from the passive LWP retrievals. Fig. R3 presents the LWP adjustments across different H bins for the AUW region (2016-2017), derived from collocated pixels of the active satellite product and the geostationary satellite product. The result demonstrates that LWP adjustment increases with H. This finding is consistent with the pattern observed in the manuscript for the AUW region, confirming the robustness of our conclusions.

[Figure]

**Figure R3: LWP adjustments binned by H for the AUW region (25º-35ºS, 95º-105ºE) during 2016-2017.** H is derived from the CloudSat-CALIPSO 2B-GEOPROF-LIDAR product. LWP adjustment is calculated by the $N_d$ and LWP obtained from the CER_GEO_ED4_HIM08_SH_V01.2 product. **The figure is the same as Figure S20 in the revised manuscript.**

Therefore, we have revised the discussion to better reflect this point only in the section of diurnal variation of LWP adjustments (see Lines 328-350): "In the AUW region, the diurnal variations of the overall LWP adjustments (black line in Fig. 3B) and cloud thickness (blue line in Fig. 3B) demonstrate a strong consistency with a turning point at 1300 LT. In the morning, thicker clouds occur in a well-mixed boundary layer with stronger cloud-top entrainment and sufficient upward moisture transport (Lu et al., 2023; Zheng et al., 2018b). Therefore, LWP is less sensitive to entrainment-feedbacks with increasing $N_d$ in thicker clouds. LWP adjustments become more negative with the gradual thinning of clouds. This can be attributed to the weaker moisture transport and cloud-top entrainment in the decoupling boundary layer. As a result, the entrainment feedback with increasing $N_d$ has a greater influence on LWP in thinner clouds, corresponding to more negative LWP adjustments. Qiu et al. (2024) also discovered the important role of thick-thin cloud transitions in the diurnal variation of LWP adjustments. Rosenfeld et al. (2019) pointed out that cloud thickness (H) constrained most of the meteorological impacts, and $N_d$ explained nearly half of the LWP variability for a given H. We further constrain LWP adjustments in different H intervals (Fig. S19B). Negative LWP adjustments become weaker as H increases (Fig. S19B). Our results suggest that the physical processes under different H conditions appear to determine how LWP changes with $N_d$, rather than H itself.

Rosenfeld et al. (2019) demonstrated an overall positive LWP adjustment when separating H. The discrepancy may arise from their focus on samples in convective cores (top 10% of cloud optical thickness), which are closer to adiabatic, whereas our samples suggest more exchange with the free atmosphere. Considering H from the passive satellite product is based on an empirical relationship, we further validated the conclusion using the 2B-GEOPROF-LIDAR product from CloudSat-CALIPSO, where H is directly calculated by the cloud base height (CBH) and cloud top height

(CTH) of single-layer cloud samples (Fig. S20). The consistent pattern of increasing LWP adjustment with H from both datasets supports the robustness of our conclusion.

In contrast, H cannot explain the diurnal variation of LWP adjustments in the ECS region (Fig. S19E), even when the spatial variability is considered (Fig. S21). This is likely attributed to different cloud regimes and complex meteorological covariations in the ECS region. Therefore, we attempt to explain the diurnal characteristics of LWP adjustments based on the boundary layer mechanisms in the ECS region.

[Figure]

**Figure 3:** Panels (A) and (C) display the normalized joint histograms of $N_d$ and LWP in log-log space at 1400 LT, for the AUW region and the ECS region, respectively. The complete pictures of all available daytime are presented in Fig. S2. The solid black lines in Panels (B) and (D) show the diurnal variations of LWP adjustments. The blue line in (B) represents the diurnal variation of H in the AUW region. Colored lines in (D) are diurnal variations of different stages in (C). Red dashed lines represent the average LWP adjustments during MODIS Terra (1030 LT) and Aqua (1330 LT) overpasses, –0.39 for the AUW region (B) and –0.01 for the ECS region (D).

[Figure]

**Figure S19: LWP adjustments in log-log spaces and their diurnal patterns in two typical regions.** Cloud samples are scattered in $N_d$-LWP log space at 1400 LT in the (A) AUW and (D) ECS region. The complete pictures of all available daytime are presented in Fig. S11. Colored dots are samples in different

cloud thickness (H) bins (unit: m). Black dots represent the median LWP in each $N_d$ bin. The colored lines are the fits of black dots at different stages in the ECS region. Diurnal variations of LWP adjustments binned by H in the (B) AUW and (E) ECS regions are shown. Colored lines in (F) are diurnal variations of different stages in (D), while black lines in (C) and (F) are the overall diurnal variations of LWP adjustments in two regions, respectively. The blue line in (C) represents the diurnal variation of H. Red dashed lines represent the average LWP adjustments during MODIS Terra (1030 LT) and Aqua (1330 LT) overpasses, –0.39 for the AUW region (C) and –0.01 for the ECS region (F).

[Figure]

**Figure S21:** Panels (A) and (C) present the diurnal variations of LWP adjustments binned by H for two different $N_d$ stages of the ECS region in Figure S19D. Panels (B) and (D) display the overall diurnal patterns for the two stages.".

13. Estimation of cloud height: It suffices to say that the estimation follow the method in Sun-Mack et al (2014).

**Response:** Thanks for your comment! We have rephrased the sentence as suggested.

14. Line 329: "indicating that entrainment drying originates from evaporation at the cloud base". This explanation is unphysical. Moreover, in a mixed-layer model, cloud top entrainment is the strongest during the nighttime, because it is driven by turbulence enhancement.

**Response:** Thank you for your feedback. We acknowledge that attributing the lifting of cloud base to entrainment drying was unphysical. According to the model study by Bougeault (1985), the daytime rise in cloud base height of marine stratocumulus is due to the reduced moisture supply from the surface, resulting from boundary layer decoupling.

The sentence has been rephrased to be more accurate (see Lines 293-295): "The decrease of LWP before 1300 LT is primarily attributed to the lifting of the cloud base, indicating that the upward moisture transport is suppressed, which is in line with an early modeling study for typical Sc cloud

regimes (Bougeault, 1985).".

15. Line 369, why should we expect a diurnal cycle in subsidence over the open ocean? It doesn't seem realistic. Even if a diurnal cycle in subsidence was observed, it remains to be demonstrated that is meaningful or physical. I am not aware of any large-scale diurnal cycle in subsidence other than in coastal regions near mountainous terrain (e.g. west coast of South America).

**Response:** We agree with the reviewer's comment. Analysis of ERA5 reanalysis data for the AUW region reveals a weak diurnal variation in the large-scale subsidence (~ 0.047 to ~ 0.054 Pa/s, Fig R4A). Therefore, discussing diurnal variations in this parameter is inappropriate. We have rephrased the sentences in the revised manuscript (see Lines 320-323): "Furthermore, according to the relationship between CLTH, large-scale subsidence ($w_s$, always negative), and entrainment rate ($w_e$) ($\frac{dCLTH}{dt} = w_s + w_e$) in the mixed-layer model framework (Painemal et al., 2013), we explain the variations after 1200 LT. Given the weak diurnal variation of $w_s$ over the open ocean (~ 0.047 to ~ 0.054 Pa/s, gray line in Fig 4A), the observed decrease in CLTH after 1200 LT is likely attributed to a weakening of $w_e$.".

[Figure]

**Figure R4: Diurnal patterns in the AUW region.** (A) Cloud liquid water path (LWP), cloud-top height (CLTH), cloud base height (CLBH), and vertical velocity on 700 hPa (omega700, positive values indicate downdraft) from ERA5 reanalysis. (B) Decoupling index in the AUW region. (C) Cloud droplet number concentration ($N_d$) and effective radius ($r_e$). (D) Aerosol optical depth (AOD). **The figures are the same as Figure 4 in the revised manuscript.**

16. Line 364: It should read "Meanwhile, the decoupling cuts off moisture…"

**Response:** Thanks for your careful review. We have rephrased the sentence as suggested.

17. Line 387-388: "Apparently, LWP skewness is a more appropriate indicator to reflect cumulus coupling in this region", What is the line of evidence for this?, why did you use the word "apparently"?

**Response:** Thanks for your constructive comment! The ECS region is a Sc-Cu transition region due to the "deepening-warming" process. Under this condition, the boundary layer is never fully coupled but exhibits local cumulus coupling. The formation of Cu beneath the Sc causes a positive skewness of the probability density function (PDF) of LWP (Zheng et al., 2018a). Therefore, LWP skewness serves as a measure of the decoupling degree.

We have removed the word "apparently" and rephrased the sentence to be clearer in Lines 362-368: "Under this condition, MBL is never fully coupled but exhibits local cumulus coupling. According to Zheng et al. (2018), the formation of Cu beneath the Sc will render local coupling by feeding moisture into the upper cloud layer, thus causing a positive skewness of the probability density function (PDF) of LWP. Therefore, the skewness of the LWP PDF can be used to estimate the cumulus coupling in this region:

$$skewness = \frac{E(x - u)^3}{\sigma^3} \tag{4}$$

where E is the expected value, μ and σ is the mean and standard deviation of x, respectively. Positive skewness indicates more data tends to be distributed to the right, and vice versa.".

18. Line 402: "hypothesized" instead of "believed"

**Response:** The mistake has been corrected.

19. Line 484-486: This is also the conclusion of Qiu et al. (2024)

**Response:** Thank you for your comments! The sentences have been rephrased in Lines 465-468: "The results highlight the critical need to account for diurnal variations of LWP adjustments when assessing the aerosol indirect effect. This growing consensus among researchers underscores the importance of incorporating geostationary observations or high-resolution simulations to better constrain the diurnal effects of LWP adjustments (Qiu et al., 2024; Rahu et al., 2022; Smalley et al., 2024).".

20. Line 540: "Cloud thickness in the AUW region serves as a confounder to separate the effects of meteorological covariations." You have not demonstrated this.

**Response:** Thanks for your insightful comment! We agree with the reviewer that the statement was not directly demonstrated in our study. We have revised the sentence to maintain accuracy (see Lines 516-517): "Negative LWP adjustments become weaker as cloud thickness increases.".

21. Line 545: I disagree, Figure 2 clearly shows that the V shape is an artifact of confounding spatial variability with microphysical processes. In other words, the relationship

**Response:** Thank you for your comment! We noted that the reviewer's final comment appears to be truncated and not fully visible in our system. However, this comment seems to align with the previously raised concerns regarding the V shape. In response, we have revised the sentence to address the concern (see Lines 520-525): "Our results demonstrate the V shape is the manifestation of spatial variability in the ECS region. Specifically, the ECS serves as a coastal transition zone, experiencing the combined effects of continental and oceanic air masses. Coastal samples show a pronounced LWP increase with $N_d$, particularly for $N_d > \sim300$ cm$^{-3}$, while offshore samples exhibit a negative $N_d$-LWP relationship, with most $N_d$ values below $\sim300$ cm$^{-3}$. The unique behaviour of LWP increasing at high $N_d$ is attributed to the influence of northerly cold-air advection from the continent during the cold seasons (75% of samples are from spring and winter).".